# Flows and Diffusions on the Neural Manifold

## Abstract

Diffusion and flow-based generative models have achieved remarkable success in domains such as image synthesis, video generation, and natural language modeling. In this work, we extend these advances to *weight space learning* by leveraging recent techniques to incorporate structural priors derived from optimization dynamics. Central to our approach is modeling the trajectory induced by gradient descent as a trajectory inference problem. We unify several trajectory inference techniques towards matching a gradient flow, providing a theoretical framework for treating optimization paths as inductive bias. We further explore architectural and algorithmic choices, including reward fine-tuning by adjoint matching, the use of autoencoders for latent weight representation, conditioning on task-specific context data, and adopting informative source distributions such as Kaiming uniform. Experiments demonstrate that our method matches or surpasses baselines in generating in-distribution weights, improves initialization for downstream training, and supports fine-tuning to enhance performance. Finally, we illustrate a practical application in safety-critical systems: detecting harmful covariate shifts, where our method outperforms the closest comparable baseline.

## 1 Introduction

Flow matching (FM) (Albergo and Vanden-Eijnden, 2023; Lipman et al., 2023; Liu et al., 2023) is a prominent fixture in generative modeling tasks from imaging (Lipman et al., 2023; Tong et al., 2024; Esser et al., 2024; Liu et al., 2024a) to language (Gat et al., 2024; Shaul et al., 2024; Campbell et al., 2024). However, its application to neural network weights has not been explored. By leveraging the principled, yet versatile training of FM, we aim to generate task-specific weights on novel tasks.

The natural question is: why generate task-specific weights instead of relying on conventional training methods? One compelling reason is efficiency. If we can train a meta-model to produce classifiers conditioned only on the evaluation dataset, then generating weights reduces to a single inference pass of our flow or diffusion model. This motivation parallels recent work in zero- and few-shot learning (Zhang et al., 2024; Soro et al., 2025), where generalization to new tasks is achieved with minimal or no training. Further on efficiency, conditionally generated weights can also serve as a strong, head-start initialization for downstream fine-tuning, which we later evaluate on corrupted datasets. This approach is especially practical when training a large number of smaller networks, such as in applications involving implicit neural representations (Essakine et al., 2025). Finally, we argue that learning to generate neural network weights opens a new perspective: it allows us to reinterpret diverse problems as questions on weight space. We illustrate this view in Section 5 through an application to detecting harmful covariate shift.

In this paper, we introduce flow matching as a new class of methods for generating neural network weights, designed to incorporate structural priors such as training trajectories and source distributions. Under this framework, we may cast our goal as one of *trajectory inference* (Lavenant et al., 2024), reconstructing the continuous-time dynamics $t \mapsto p_t^*$ given easy sampling from the marginal distributions $(p_{t_k}^*)_{k=0}^K$. In practice, temporal observations are sparse, necessitating methods that can sensibly interpolate between observed timepoints, often leveraging biases in data. Indeed, we further ground our approach in the *manifold hypothesis* (Bengio et al., 2013), which posits that natural data lies on a low-dimensional submanifold of the ambient

---

†Equal contribution.

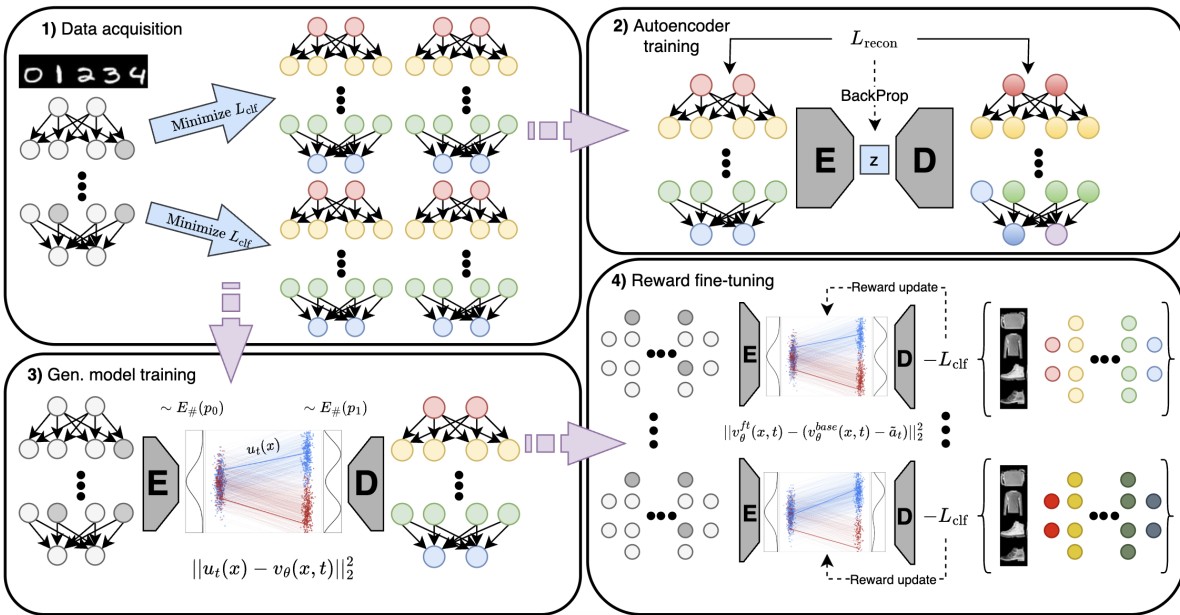

Figure 1: **Example unconditional pipeline. (1)** Base model pre-training, shown here on MNIST, producing checkpoints across epochs. **(2) Optional:** variational autoencoder training with a weight-space reconstruction objective. **(3)** Generative meta-model training; here we illustrate *unconditional* N$\mathcal{M}$-CFM w/ (trained) VAE (our default N$\mathcal{M}$-CFM is on weight space directly) using the weight initialization from **(1)** as $p_0$. **(4) Optional:** reward fine-tuning via adjoint matching where $r(\cdot) = -L_{\mathrm{clf}}(\boldsymbol{X}_{\mathrm{FashionMNIST}}; \cdot)$, steering the *trained* meta-model towards generating FashionMNIST classifiers.

space, representing a bias that may be incorporated as a prior. Drawing on the Lottery Ticket Hypothesis (Frankle and Carbin, 2019; Zhang et al., 2021; Liu et al., 2024b) as well as the body of work on pruning (Cheng et al., 2024), we extend this intuition to weight space: neural networks themselves tend to lie on a low-dimensional structure, which we refer to as the *neural manifold* (N$\mathcal{M}$)[1]. We will make use of these observations to motivate the various experiments conducted in Section 5.

In this work, we make preliminary steps toward understanding flows and diffusions on the neural manifold. Our contributions include: **1)** we unify and prove characterizations of various methods to approximate a gradient descent trajectory, enabling more accurate modeling of our priors; **2)** we incorporate theoretical considerations to design flow- and diffusion-based approaches for generating weights that match or exceed conventionally trained models on in-distribution tasks, provide better initializations for downstream training, and allows for conditioning on context data to retrieve pre-trained weights from a distribution pre-trained on various datasets; **3)** we incorporate a fine-tuning mechanism, grounded in adjoint matching (Domingo-Enrich et al., 2025), to enhance performance, and **4)** we show how this can be used to detect harmful covariate shifts that outperforms the closest comparable baseline, supporting our motivation to reinterpret problems as questions on weight space.

## 2 Preliminaries

**Conditional flow models.** Chen et al. (2019) first introduced continuous normalizing flows as an effective data generation process through modeling dynamics. Simulation-free methods improve on this concept by simplifying the training objective (Albergo and Vanden-Eijnden, 2023; Lipman et al., 2023; Liu et al., 2023). Following the formulation of Lipman et al. (2023), given random variables $\bar{\mathbf{x}}_0 \sim p_0$ and $\bar{\mathbf{x}}_1 \sim p_1$ a

---

[1]We note that modern deep neural network architectures exhibit parameter symmetries (Hecht-Nielsen, 1990; Chen et al., 1993) that may be exploited to reduce the size of weight space. In particular, the neural manifold can be viewed as a *quotient set* where we identify distinct weights perturbed by symmetric transformations. As our focus is on generative techniques, the main paper limits itself to the un-identified space, deferring a discussion and modest experiments to App. G.

data distribution, define a reference flow $\bar{\mathbf{x}} = (\bar{\mathbf{x}}_t)_{t \in [0,1]}$ where $\bar{\mathbf{x}}_t = \beta_t \bar{\mathbf{x}}_0 + \alpha_t \bar{\mathbf{x}}_1$ with the constraint that $\alpha_0 = \beta_1 = 0$ and $\alpha_1 = \beta_0 = 1$. The aim of flow modeling is to learn a path $\mathbf{x} = (\mathbf{x}_t)_{t \in [0,1]}$ which has the same marginal distribution as $\bar{\mathbf{x}}$. To make this a feasible task, we describe this process as an ODE: $d\mathbf{x}_t = v(\mathbf{x}_t, t)dt$ where $\mathbf{x}_0 \sim \mathcal{N}(0, \boldsymbol{I})$. Training proceeds by first parameterizing $v(\mathbf{x}_t, t)$ by a neural network $\theta$ and matching the reference flow velocity, i.e. $u(\mathbf{x}_t, t) := \frac{d}{dt}\bar{\mathbf{x}}_t$. This would, however, be an unfeasible training objective, therefore, we condition on samples from the distribution $\mathbf{x}_1 \sim p_1$ and train

$$\mathcal{L}_{\mathrm{cfm}}(\theta) = \mathbb{E}_{t \sim U[0,1], \mathbf{x}_1 \sim p_1, \mathbf{x}_t \sim p_t(\cdot|\mathbf{x}_1)} ||v_\theta(\mathbf{x}_t, t) - u(\mathbf{x}_t, t \mid \mathbf{x}_1)||. \tag{1}$$

Lipman et al. (2023) proved that this loss produces the same gradients as the marginal loss, thus optimizing it will result in convergence to the reference $u(\mathbf{x}_t, t)$. Moreover, we can always marginalize an independent conditioning variable $\boldsymbol{y}$ on $v_\theta, u$ – this will serve as our context conditioning vector.

**Modeling probability paths.** The straight line paths advocated in the conventional flow matching framework (Lipman et al., 2023) can be contextualized in the broader framework of matching interpolants $(\mathbf{x}_t)_{t \in [0,1]}$ to minimize a given energy function. In the conventional case, the kinetic energy $\mathcal{E}(\mathbf{x}_t, \dot{\mathbf{x}}_t) = \mathbb{E}_{t \sim U[0,1]} ||\dot{\mathbf{x}}_t||^2$ is minimized (Shaul et al., 2023), however, more general energies have been considered (Neklyudov et al., 2023; 2024; Kapusniak et al., 2024; Liu et al., 2024c) to broaden the class of learnable paths. Of interest to us is the concept of finding interpolants that depend on prior knowledge as provided through samples. For instance, Metric Flow Matching (Kapusniak et al., 2024) learns parametric interpolants that minimize

$$\mathbb{E}_{(\mathbf{x}_0, \mathbf{x}_1) \sim \pi} \mathcal{E}_g(\mathbf{x}_t) = \mathbb{E}_{t \sim U[0,1], (\mathbf{x}_0, \mathbf{x}_1) \sim \pi} ||\dot{\mathbf{x}}_t||^2_{g(\mathbf{x}_t)}, \tag{2}$$

where $\pi$ is some distribution on the product, typically $p_0 \otimes p_1$, and $g$ is a data-dependent Riemannian metric on the ambient space $\mathbb{R}^d$. Alternatively, Rohbeck et al. (2025) proposed a more stable option using cubic splines for multi-marginal flow matching. Most related to our setting is the task of modeling distributions evolved via gradient flow: $\frac{d}{dt}p_t = -\nabla \cdot (p_t \nabla s_t)$. For instance, TrajectoryNet (Tong et al., 2020) is an older work that uses continuous normalizing flows to model cellular trajectories under a spatial potential. More recently, JKOnet and JKOnet* (Bunne et al., 2022; Terpin et al., 2024) exploited the connection between diffusion processes and energy-minimizing probability paths (in Wasserstein space) to model a Wasserstein gradient flow by learning the drift potential $V_\theta(\cdot, t)$ in a diffusion process.

**Harmful covariate shift detection.** Covariate shifts refers to changes in the test data distribution $p_{\mathrm{test}}(x)$ as compared to the training distribution $p_{\mathrm{train}}(x)$ while the relation between inputs and outputs remain fixed, i.e. $p_{\mathrm{test}}(y|x) = p_{\mathrm{train}}(y|x)$. Importantly, we do not require labels to determine this shift, thus it is practical to do so in a standard deployment setting. Prior work in this domain include deep kernel MMD (Liu et al., 2020), H-divergences (Zhao et al., 2022), and Detectron (Ginsberg et al., 2023). As Detectron requires minimal tuning and is most performant in low-data regimes ($N < 100$ samples), we emphasize the use of this approach. In particular, Detectron (Ginsberg et al., 2023) builds off of *selective classification*—building classifiers that accept/reject test data depending on closeness to the training distribution—and *PQ-learning* (Goldwasser et al., 2020) that extends the conventional theory of PAC learning to arbitrary test distributions by employing selective classification. The main idea considers the generalization set $\mathcal{R}$ of a classifier $f_\theta$ and samples $\mathcal{Q}$ from an unknown distribution. The strategy is to fine-tune constrained disagreement classifiers (CDCs) to agree with $f_\theta$ on $\mathcal{R}$ but disagree on $\mathcal{Q}$. If $\mathcal{Q} \subset \mathcal{R}$, then it will be difficult to disagree on $\mathcal{Q}$, but if the CDCs behave inconsistently on $\mathcal{Q}$, that suggests a covariate shift. Notably, this method is sample-efficient, agnostic to classifier architecture, and may be used in tests of statistical significance.

## 3 Modeling Weight Trajectories

In this section, we build towards our approach. Proofs are provided in App. A.

### 3.1 The continuity equation on neural network parameters

For the purpose of our analysis, let us restrict our view to neural networks that are optimized by gradient descent (GD) algorithms to minimize a loss $\mathcal{L}(\theta_k) := d(\mathcal{M}_\theta(\boldsymbol{X}) - \boldsymbol{Y})$, where $\mathcal{M}_\theta$ is a neural network

parameterized by trainable weights $\theta \in \mathbb{R}^p$, $\boldsymbol{X} \in \mathbb{R}^{N \times D}$ are inputs, $\boldsymbol{Y} \in \mathbb{R}^{N \times c}$ are labels, and $d$ is some differentiable distance function, such as cross-entropy. To minimize via GD, parameter updates are done by $\theta_{k+1} = \theta_k - \alpha \nabla \mathcal{L}(\theta_k)$ given some learning rate $\alpha > 0$. Taking the learning rate to zero, we can view parameter evolution as a *gradient flow*. For simplicity, we assume that the updates are deterministic (contrary to stochastic gradient descent which randomly selects training batches), and defer to App. D an approach incorporating stochasticity via Schrödinger bridges. For now, the sole source of randomness is the initialization $\theta_0 \sim p_0$. Within this framework, we can write down a continuity equation. In later sections, we show how this result underpins the choice of modeling framework.

---

**Theorem 1** (Informal; follows Ch. 8.3 of Santambrogio (2015)). *Let $\theta_0 \sim p_0$ be initialized network parameters and the loss $\mathcal{L}$ is $C^1$ in the network parameters. If $(\theta_t)_{t \geq 0}$ is the gradient descent curve, we have $p_t = \mathrm{Law}(\theta_t)$ with*

$$\partial_t p_t - \nabla \cdot (p_t \nabla \mathcal{L}) = 0. \tag{3}$$

*Proof.* The idea is to view GD as an iterated minimization scheme on $\mathcal{P}(\Omega)$ with functional $\mathcal{F}(p_t) = \int_\Omega \mathcal{L}(x) \, dp_t(x)$. Alternatively, this may be viewed as the necessary first-order optimality conditions of a JKO scheme (Lanzetti et al., 2024; Terpin et al., 2024). We provide the full proof in App. A. □

---

## 3.2 Approximating Eqn. 3

The problem of learning the continuous dynamics of a system governed by a continuity equation has been studied in many forms in existing literature. In our setting, Theorem 1 establishes a link between the practical dynamics of SGD and the continuity equation (Eqn. 3), which provides a more tractable theoretical framework. Building on this connection, we study methods that realize parameterized solutions to Eqn. 3: $\partial_t p_t + \nabla \cdot (p_t v_t^\Theta) = 0$, thereby providing a common lens of interpretation. Lipman et al. (2023, Theorem 1) showed that CFM may be viewed through this lens, hence motivating its use. However, its training objective assumes affine Gaussian paths, which oversimplifies the non-terminal distributions along a GD path. Therefore, we turn to a natural generalization: multi-marginal flow matching (MMFM), which has been shown (Rohbeck et al., 2025, Proposition 2) to correspond to training multiple CFMs. To add to our selection, Theorem 2 shows a correspondence between MMFM and JKOnet* (Terpin et al., 2024) via the *action gap*; this connection, its formulation towards modeling Eqn. 6, and its efficient scalar potential parameterization motivates its consideration. Below, we expound on the action gap lens, provide some background on MMFM and JKOnet*, and present a generalization to non-affine regression targets.

### 3.2.1 The action gap

Frameworks which match gradients have a learnable function $\Psi_\theta(x, t)$ which is trained to match a regression target. Two representatives of this approach are the works on Action Matching (Neklyudov et al., 2023; 2024) and the JKOnet family (Bunne et al., 2022; Terpin et al., 2024). We found the theoretical framework of the *action gap* from Action Matching to be most suitable as a reference; we recall it here.

**Action matching.** The action matching setup presumes an initial distribution $q_0$, a velocity field $v : [0, 1] \times \Omega \to \mathbb{R}^p$, and a continuity equation which describes the dynamics: $\frac{d}{dt} q_t = -\nabla \cdot (q_t v_t)$. Neklyudov et al. (2023, Theorem 2.1) showed that, under mild conditions on $q_t$, a unique function $s_t^*(x)$ termed the *action* may be defined such that $v_t(x) = \nabla s_t^*(x)$ and the continuity equation $\frac{d}{dt} q_t = -\nabla \cdot (q_t \nabla s_t^*)$ is satisfied. One can readily see the connection with Eqn. 3: $s_t^* \equiv -\mathcal{L}$ up to a constant. Therefore, the *action gap* is

$$AG(s, s^*) = \frac{1}{2} \int_0^1 \mathbb{E}_{q_t(x)} ||\nabla s_t(x) - \nabla s_t^*(x)||^2 \, dt = \frac{1}{2} \int_0^1 \mathbb{E}_{q_t(x)} ||\nabla s_t(x) + \nabla \mathcal{L}(x)||^2 \, dt. \tag{4}$$

This optimization is clearly intractable because of the required access to $\nabla \mathcal{L}$, therefore the authors computed a more tractable variational objective for optimization (Neklyudov et al., 2023, Theorem 2.2.). To close this exposition, we recall a bound on the 2-Wasserstein distance which will be a recurring theme in this section.

**Proposition 1** (Prop. A.1 of Neklyudov et al. (2023)). *Suppose the curl-free vector field $\nabla s_t$ is continuously differentiable in $(t, x)$, and uniformly Lipschitz in $x$ throughout $[0, 1] \times \mathbb{R}^p$ with Lipschitz constant $K$. Let $\hat{q}_t$ denote the density path induced by $\nabla s_t$. Then,*

$$W_2^2(\hat{q}_\tau, q_\tau) \leq e^{(1+2K)\tau} \int_0^\tau \mathbb{E}_{q_t(x)} ||\nabla s_t(x) + \nabla \mathcal{L}(x)||^2 \, dt. \tag{5}$$

### 3.2.2 Approximating Eqn. 3 in practice

Although we made use of the action gap as our theoretical framework, we opt for the simpler objectives of JKOnet and multi-marginal flow matching in practice. In this section, we present some background and show their objectives may be cast as minimizing the action gap in the discretization limit.

**JKOnet.** The JKOnet family considers the problem of modeling the Fokker-Planck equation

$$\partial_t p_t(x) = \nabla \cdot (p_t(x) \nabla V(x)) + \beta \Delta p_t(x), \tag{6}$$

given some potential $V$. The seminal work Jordan et al. (1998) (namesake of the JKO scheme) related such equations to a variational objective in Wasserstein space, namely

$$\mu_{t+1} = \arg\min_{\mu \in \mathcal{P}(\mathbb{R}^p)} J(\mu) + \frac{1}{2\tau} W_2^2(\mu, \mu_t) \quad \text{where} \quad J(\mu) := \int_{\mathbb{R}^p} V(x) p_t(x) \, dx + \beta \int_{\mathbb{R}^p} p_t(x) \log(p_t(x)) \, dx, \tag{7}$$

and step size $\tau > 0$. Focusing on the most recent presentation, termed JKOnet$^*$ (Terpin et al., 2024), consider the Euclidean analog of Eqn. 7 and its first-order optimality condition: $\nabla J(x_{t+1}) + (x_{t+1} - x_t)/\tau = 0$, where $x_t \sim \mu_t$. If we let $\beta = 0$, then by Terpin et al. (2024, Prop. 3.1), we have the minimization objective

$$\int_{\mathbb{R}^p \times \mathbb{R}^p} \left\| \nabla V(x_{t+1}) + \frac{1}{\tau}(x_{t+1} - x_t) \right\|^2 d\pi_t(x_t, x_{t+1}) = 0, \tag{8}$$

where $\pi_t$ is the optimal coupling between $\mu_t$ and $\mu_{t+1}$. It can be readily seen that when $\beta = 0$, Eqn. 6 matches the form of Eqn. 3, and the requisite setup for action matching. The issue with applying the bound Eqn. 5 stems from the time-discretization. To resolve this, we show in Theorem 2 (replacing $u_t$ with $(x_{t+1} - x_t)/\tau$) that the action $\nabla s_t^*$ Eqn. 4 may be approached in the limit. Moreover, the objective Eqn. 8 is an intuitive description of matching the best linear approximation of the gradient $\nabla \mathcal{L}$. Motivated by its simplicity, we opt towards JKOnet$^*$ as the representative approach.

**Multi-marginal flow matching.** The MMFM objective is similar to that of CFM in Eqn. 3. The difference lies in the definition of the regression target. Suppose we wish to generate marginal densities $p_{t_0}^*, p_{t_1}^*, \ldots, p_{t_K}^*$. Instead of sampling $x \sim p_1$, we sample $z = (x_0, \ldots, x_K)$ independently from each of the marginal densities. To align with the CFM objective, the reference path $p_t(x|z)$ must be a piecewise affine Gaussian path, and its mean a linear interpolation between the $K + 1$ samples. Formally, define the interpolant

$$\mu_t(z) = \sum_{k=0}^{K-1} \left( x_k + \frac{t - t_k}{t_{k+1} - t_k}(x_{k+1} - x_k) \right) \cdot \mathbf{1}_{[t_k, t_{k+1})}(t) \tag{9}$$

and the regression target ought to be

$$u_t(x \mid z) = \sum_{k=0}^{K-1} \frac{x_{k+1} - x_k}{t_{k+1} - t_k} \cdot \mathbf{1}_{[t_k, t_{k+1})}(t). \tag{10}$$

With the usual marginalization of $u_t$, i.e. $u_t(x)p_t(x) = \mathbb{E}_{q(z)}[u_t(x \mid z)p_t(x \mid z)]$, we can argue by checking the continuity equation Tong et al. (2024, Theorem 3.1) that $p_t$ is generated by $u_t$.

In addition, it is natural to think that if the timepoints $(t_0, \ldots, t_K)$ were dense enough, then its limit is the true probability path $t \mapsto p_t^*$. Rohbeck et al. (2025, Proposition 2) proved that the MMFM objective is

equivalent to solving $K$ CFM objectives. By analogy, we consider a MMFM setup equivalent to $K$ OT-CFM objectives, and we show that the reference path $p_t$ approaches $p_t^*$ in the sense of the action gap (cf. Prop. 1).

---

**Theorem 2.** *Suppose the true marginals evolve according to $\frac{d}{dt}p_t^* = -\nabla \cdot (p_t^* \nabla s_t^*)$ and $t \mapsto p_t^*$ is an absolutely continuous curve. Define $q(z)$ such that marginalizing $q$ with respect to all variables except $x_k, x_{k+1}$ yields the coupling $p_{t_k} \otimes (T_k^{k+1})_{\#} p_{t_k}$, where $T_k^{k+1}$ is the transport map from $p_{t_k}$ to $p_{t_{k+1}}$. Then,*

$$\lim_{|t_k - t_{k+1}| \to 0} \int_0^1 \mathbb{E}_{p_t(x)} ||u_t(x) - \nabla s_t^*(x)||_2^2 \, dt = 0. \tag{11}$$

*Replacing $u_t$ with $\frac{x_{t+1} - x_t}{\tau}$, this shows that $\nabla V$ (Eqn. 8) regresses to the reference action in the limit.*

---

### 3.2.3 Learned proxy matching

In this section, we generalize the regression target in Eqn. 8 and $u_t(x|z)$ in Eqn. 10 to encompass methods such as Metric Flow Matching by presenting the notion of *proxy curves*. In particular, we define a family of curves that minimize an objective (such as a data-dependent metric or a Lagrangian) and discuss its fitness as an interpolant (cf. $\mu_t$ in Eqn. 9) w.r.t. the action gap. As we only discuss the theory of this approach, we only present an overview here and leave details to App. A.3.

The minimization objective of choice in this section is the Lagrangian $L(x_t, \dot{x}_t, t) = ||\dot{x}_t||_2^2 + V_t(x_t, \dot{x}_t)$. This allows some flexibility in the choice of energy functional $V$, which in practice will be data-dependent such as in Metric Flow Matching. In this setting, we seek to characterize choices of energy $V$ to minimize the $W_2$ distance between the proxy probability path, which evolves by $v_\theta$, and the reference $p_t$ in Eqn. 3. We start by writing down a continuity equation for the proxy path (cf. Theorem 1).

---

**Theorem 3** (Proxy reference path). *Suppose the Lagrangian $L(x_t, \dot{x}_t, t) = ||\dot{x}_t||_2^2 + V_t(x_t, \dot{x}_t)$ is Tonelli and strongly convex in velocity. The Lagrangian optimal transport map $T$ exists between $p_0$ and $p_1$. Moreover, there exists a locally Lipschitz, locally bounded vector field $w$ s.t.*

$$\partial_t \hat{p}_t + \nabla \cdot (\hat{p}_t w_t) = 0 \tag{12}$$

*satisfies $\hat{p}_t = \text{Law}(\gamma_t)$ where $\gamma$ is a random, smooth Lagrangian-minimizing curve and $(\gamma_0, \gamma_1)$ is an optimal coupling of $p_0, p_1$.*

---

Following the action matching discussion, we define a *proxy action gap* $\frac{1}{2} \int_0^1 \mathbb{E}_\gamma ||\nabla \mathcal{L}(\gamma_s) + \dot{\gamma}_s||^2 \, ds$ in terms of the curve $\gamma$. We show in App. A.3 that $W_2^2(p_t, \hat{p}_t)$ may be bound like in Prop. 1. Moreover, we note that the smoothness assumption of $\gamma$ is quite restrictive, but in practice, this may be weakened (App. E). On the flip side, if the gradient descent path is smooth enough, then we can define an energy functional $V$ such that its minimizing curve $\gamma$ stays close in gradient to $-\nabla \mathcal{L}$ (see Theorem 4).

## 4 Methods

### 4.1 Architectural modules

We describe the components of our approach below and leave more details to Appendix F, H. Throughout, we use the $\text{N}\mathcal{M}$- prefix to denote our methods, e.g. $\text{N}\mathcal{M}$-CFM to denote conditional flow matching. Our framework is designed to be modular, with different components that can be instantiated in various ways. In this work, we prioritize simplicity in order to highlight the generative framework and the proposed reward fine-tuning mechanism. Figure 1 provides an overview of how these components connect; changes to the schema are possible, such as the generative model in (3) may take in a conditioning signal, and the reward signal in (4) can differ, as in Meta-Detectron.

**Weight encoder.** Due to the often intractable size of weight space, it is sometimes necessary for modeling to in latent space (see App. F.5 for scaling remarks). We justify this design by appealing to work on the Lottery Ticket Hypothesis (Frankle and Carbin, 2019; Zhang et al., 2021; Liu et al., 2024b) as well as the body of work on pruning (Cheng et al., 2024), which suggests that, like natural data, neural networks live on a low-dimensional manifold within its ambient space. There are a variety of encoders to choose from, such as the variational autoencoder (VAE) (Kingma and Welling, 2022), and specialized encoders for neural network parameters (Kofinas et al., 2024; Putterman et al., 2024; Schürholt et al., 2024). However, to simplify matters as we are mostly focused on the generative aspect, we stick to the VAE as used in Soro et al. (2025).

**Generative meta-model.** The backbone of our meta-learning framework is a conditional FM model following Tong et al. (2024), alongside the multi-marginal variant (MMFM) that matches piecewise-linear interpolants; see App. E for a detailed discussion of interpolants. We also experiment with using the JKOnet$^*$ method in a few tests; see details in App. F. Preliminary experiments were also done with learned interpolants such as Metric Flow Matching, but we found them to be unstable, likely due to the sparsity of data and the large ambient space favoring simpler interpolants. Exploiting the flexibility of FM to use a non-Gaussian prior, we use the Kaiming uniform or normal initializations (He et al., 2015a), as the source $p_0$; see App. B for a brief theoretical remark on the effect on $p_t$. To illustrate the benefits of this choice, see the experiments and discussion in App. G.

**Reward fine-tuning.** FM models lend themselves to the recently proposed reward fine-tuning method, based on the adjoint ODE (Domingo-Enrich et al., 2025), which casts stochastic optimal control as a regression problem. This allows us to tune pre-trained flow meta-models for downstream applications, exemplified in this work by detecting harmful covariate shifts, and improved generative performance (see results in App. H.4). Specifically, this method modifies the base generative distribution $\hat{p}_1^{\text{base}}$ to generate the reward-tilted distribution $p_1^*(x) \propto p_1^{\text{base}}(x) \exp(r(x))$ via the *Adjoint Matching* (AM) algorithm. Naturally, in our setting, we suppose $p_1^{\text{base}}$ is obtained from meta-training and governs classifiers that predict on $\mathcal{D}_1$, but we wish to modify the meta-model to generate base models that predict on $\mathcal{D}_2$. Therefore, we set the reward $r(X_1) := -\mathcal{L}_2(X_1)$ where $\mathcal{L}_2$ is a loss on $\mathcal{D}_2$ such as cross-entropy and proceed with adjoint matching. Unlike naive fine-tuning strategies that require backpropagation through the inference-time solver, adjoint matching reduces the problem to a regression objective, closely resembling standard CFM training (see Algorithm 2). Importantly, it simply renews the drift by learning an additive correction: $v_t^{ft}(x) := v_t^{base}(x) + u(t, x)$ (App. F.4), so the analysis in Sec. 3.2 continues to apply without modification.

## 4.2 Detecting harmful covariate shifts

**Training CDCs.** Continuing the exposition in Section 2, we specify the training regiment of CDCs. Let $g(\cdot)$ represent one CDC and $f(\cdot)$ the base classifier; further, let $\mathbf{P} = \{(x_i, y_i)\}_{i=1}^n$ be samples from the generalization set $\mathcal{R}$ and $\mathbf{Q} = \{\tilde{x}_i\}_{i=1}^m$ from the unknown distribution. Our objective is for $g$ to maintain performance on $\mathbf{P}$, but to disagree with $f$ on $\mathbf{Q}$. Naturally, we use the cross-entropy loss $\ell_{ce}$ on $\mathbf{P}$, but on $\mathbf{Q}$, Ginsberg et al. (2023) introduces the *disagreement-cross-entropy*:

$$\ell_{dce}(\hat{y}, f(x_i)) = \frac{1}{1-N} \sum_{c=1}^N \mathbf{1}_{f(x_i) \neq c} \log(\hat{y}_c),$$

where $N$ denotes the total number of classes. We combine these objectives by minimizing the CDC loss:

Table 1: Best validation accuracy of unconditional N$\mathcal{M}$ CNN3 generation for various datasets. *Original* denotes base models trained conventionally by SGD and *p-diff* those generated with p-diff (Wang et al., 2024). MMFM($k$) and JKO($k$) indicates the number of marginal distributions in addition to $p_0$ and $p_1$.

| | CIFAR100 | CIFAR10 | MNIST | STL10 |
|---|---|---|---|---|
| Original | 25.62 | 63.38 | 98.93 | 53.88 |
| p-diff | 25.99 | 63.37 | 98.93 | 53.86 |
| N$\mathcal{M}$-CFM w/ VAE | 26.01 | 64.32 | 98.91 | 53.50 |
| N$\mathcal{M}$-CFM | 25.31 | 62.52 | 98.52 | 53.49 |
| N$\mathcal{M}$-MMFM(2) | 21.16 | 63.35 | 98.53 | 53.20 |
| N$\mathcal{M}$-MMFM(3) | 24.53 | 63.34 | 98.62 | 53.53 |
| N$\mathcal{M}$-MMFM(4) | 22.89 | 61.33 | 98.39 | 53.08 |
| N$\mathcal{M}$-JKO(3) | 24.84 | 62.60 | 98.59 | 53.44 |
| N$\mathcal{M}$-JKO(4) | 24.97 | 63.35 | 98.57 | 53.21 |

$$\ell_{cdc}(\mathbf{P}, \mathbf{Q}) := \frac{1}{|\mathbf{P} \cup \mathbf{Q}|} \left( \sum_{(x_i, y_i) \in \mathbf{P}} \ell_{ce}(g(x_i), y_i) + \lambda \sum_{\tilde{x}_i \in \mathbf{Q}} \ell_{dce}(g(\tilde{x}_i), f(\tilde{x}_i)) \right) \tag{13}$$

To test for shift, Detectron (Ginsberg et al., 2023) compares $g$ trained with $\mathbf{Q}$ sampled from the unknown distribution ($g_{\mathbf{Q}}$) against $\mathbf{Q} = \mathbf{P}^*$ ($g_{\mathbf{P}}$), i.e. samples from the generalization set. In particular, the disagreement rate or the class entropy for each case is obtained and hypothesis tested. In both cases, the disagreement and entropy are higher if $\mathbf{Q}$ represents a significant shift.

**Motivation.** The problem of detecting covariate shift is not just about the data—a lot of modern neural networks are robust to such changes. The essence of the problem is whether or not *the classifier weights* required to predict on $\mathbf{Q}$ differs from the current classifier, motivating a method that is sensitive to changes in the classifier weights required to predict on a new set. Building on the finding that the support of classifier weights is narrow (see Sec. 5.2) and the fact that the reward-tilted distribution (obtained from reward fine-tuning) has the same support, if the ideal classifier required to predict on a new dataset lies far outside of the original support, then we would expect a noticeable performance difference after reward fine-tuning than if it were close to the original support (see corruption experiments in App. H.4).

**Meta-detectron.** Our approach, termed *meta-detectron*, builds on reward fine-tuning by adjoint matching. We start by meta-training a N$\mathcal{M}$-CFM meta-model to learn classifier distributions on each of the datasets (this is identical to the unconditional generation setup). Next, we reward fine-tune, maintaining the procedure of sampling from the meta-model at each iteration to compute the reward, but now the reward function is $r(X_1) = -\ell_{cdc}(\mathbf{P}, \mathbf{Q}; X_1)$, where $X_1$ serves the role of $g$, and the original meta-model generates the base classifier $f$ in Eqn. 13. As the method requires training $g_{\mathbf{P}}$ and $g_{\mathbf{Q}}$, we likewise fine-tune two different meta-models depending on the disagreement set, and compare the disagreement rate and entropy of the generated $g_{\mathbf{P}}$ and $g_{\mathbf{Q}}$. Returning to our motivation, it is more likely for the support to lie closer to classifiers that disagree on an out-of-distribution set $\mathbf{Q}$ than those disagreeing on $\mathbf{P}^*$. That being said, we expect the disagreement rate to be more conservative overall due to the tightness of the support.

## 5 Experiments

First, we confirm various properties that are to be expected of weight generation models. Next, we explore reward fine-tuning courtesy of adjoint matching, finishing with an application to detect harmful covariate shifts. Further experiments and details may be found in Apps. G and H respectively.

### 5.1 Generative modeling desiderata

**Unconditional generation.** We first evaluate the basic modeling capabilities of the flow meta-model. The target distribution $p_1$ is obtained by training a variety of base models on common datasets: CIFAR10, CIFAR100, STL10, and MNIST, and saving weight checkpoints across 100 epochs of training. For large models, we may choose to generate only a subset of the weights. In our case, we generate the batch norm parameters for ResNet-18 (He et al., 2015b), ViT-base (Dosovitskiy et al., 2021) and ConvNext-tiny (Liu et al., 2022), and the full medium-CNN (Schürholt et al., 2022), which we denote CNN3 corresponding to the number of convolutional layers. The objective is to train a separate meta-model for each dataset and validate its base model reconstruction on classifying its corresponding test set. Focusing on CNN3, Table 1 shows that we are able to match base models trained conventionally and with p-diff (Wang et al., 2024), with extra results in Table 9. The results here suggest that our approach have at least the same modeling capacity as prior work with diffusion models, which is expected. In addition, Figure 2 visualizes how validation loss changes over inference timesteps for the various N$\mathcal{M}$ methods. We observe that methods incorporating trajectory information (MMFM and JKO) observe a faster decrease in validation loss over inference steps. This suggests that gradient descent converges rather quickly to the final validation loss and so learning the intermediate marginal distributions leads to faster inference-time convergence. Finally, due to computational constraints, we restrict further experiments to the best observed MMFM and JKO settings: in our case, MMFM(3) and JKO(4); see App. H for a discussion of computation times.

**Trajectory modeling.** In this experiment, we evaluate the ability of different approaches to model the weight trajectory. As decided above, we used N$\mathcal{M}$-MMFM(3) and N$\mathcal{M}$-JKO(4) as representatives for this method. Moreover, in the interest of fairness, we divide the trajectory into 5 buckets, and so the MMFM

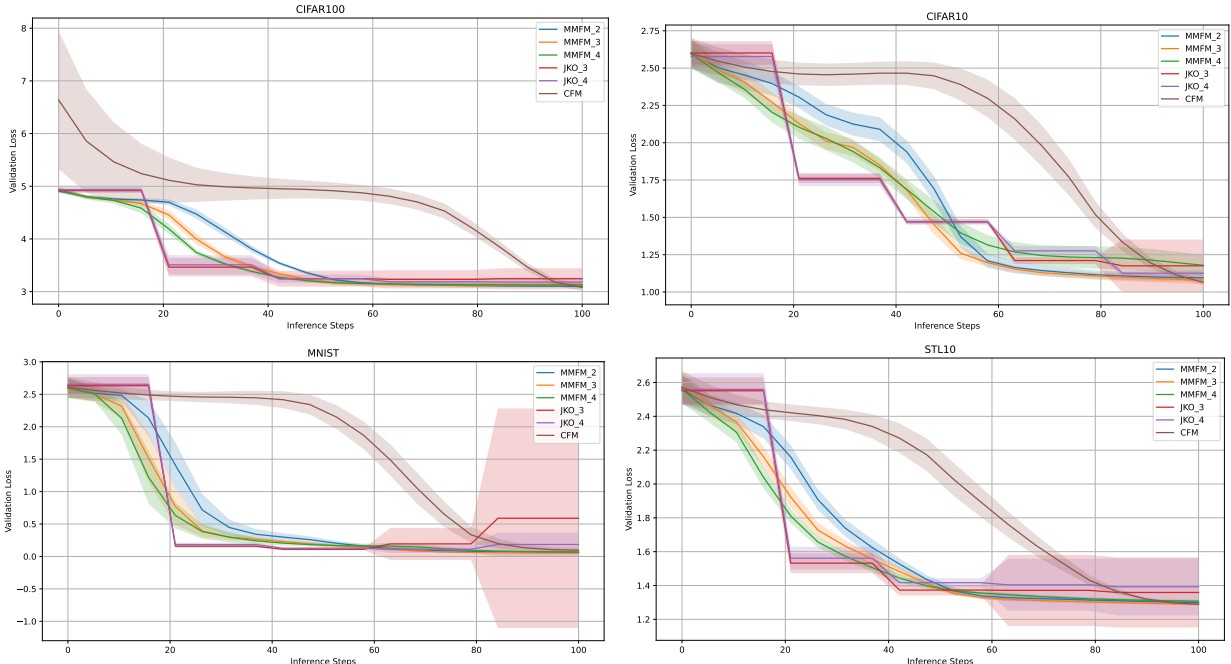

Figure 2: Base model validation loss over the course of inference for various N$\mathcal{M}$ methods. The plots were computed on 20 out of 100 intermediate timepoints for MMFM and CFM, but restricted by design to $k$ timepoints for JKO($k$). In the legend, MMFM_k refers to MMFM with $k$ intermediate marginal distributions (that is, distributions in addition to $p_0$ and $p_1$) and likewise for JKO.

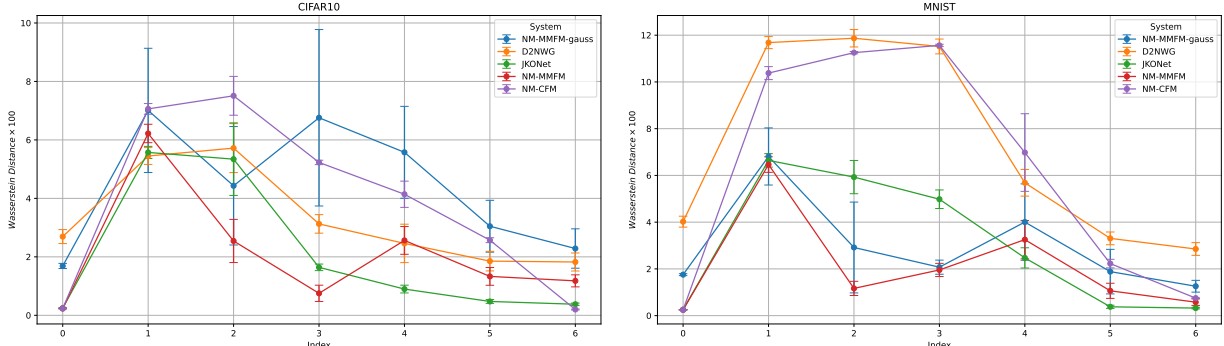

Figure 3: Mean $W_1$-distance ($\times 100$) between reference and generated intermediate marginals over 5 seeds of unconditional generation. The horizontal axis corresponds to increasing indices of the intermediate marginals, i.e. $k$ in $p_{t_k}$ where $t_0 = 0$, $t_6 = 1$. The plots also show the effect of using a Gaussian prior with MMFM (denoted N$\mathcal{M}$-MMFM-gauss); we exclude N$\mathcal{M}$-CFM-gauss due to its large $W_1$ deviation.

and JKO methods would need to interpolate between training distributions. Our baseline is D2NWG (Soro et al., 2025) where the VAE is trained on full trajectory weights at each batch iteration. See Figure 3 for results and App. G for more results, including an investigation of parameter symmetries. Interestingly, we find that the Wasserstein-1 ($W_1$) distance of the generated trajectory to be consistently lower in D2NWG vs. N$\mathcal{M}$-CFM, but both methods lag behind MMFM and JKO which explicitly models the weight trajectory. The over-performance of MMFM and JKO is to be expected, and we suspect the D2NWG performance is due to latent space training on the full trajectory data; that is, even if the interpolated weights do not follow the expected trajectory, it still lands on the data manifold. See Figure 5 for a mock illustration.

**Model retrieval.** Following (Soro et al., 2025), we perform model retrieval to test whether the meta-model can distinguish weights of the base model given conditioning samples from the dataset the base model was trained on. The base model is the same CNN3 model and we obtain weight checkpoints from the trained on MNIST, Fashion-MNIST (FMNIST), CIFAR10, and STL10 across 50 epochs of conventional training. Unlike in the previous test, we will train just a single meta-model conditioned on context samples from their respective training sets passed through a CLIP (Radford et al., 2021) encoder (see Figure 8). During validation, we pass in a random support sample from one of the four datasets and generate the *full* CNN3. In Table 2, we see that most of our mean top-5 validation accuracy matches that of the base models. This shows that our approach has the necessary capacity for *conditional* generation on weight space. There is a noticeable decline in N$\mathcal{M}$-MMFM accuracy; this is due to the higher complexity of the target vector field as opposed to the straight N$\mathcal{M}$-CFM targets. With a larger meta-model, we were able to get higher accuracies, but the results here serve to inform the reader of this downside.

**Downstream initialization.** Next, we repeat the model retrieval test, but instead we obtain weights across 10 epochs of conventional training. We opted for the encoded runs if possible for efficiency. The generated weights are used as initialization before fine-tuning another 20 epochs. As shown in Table 3, our initialization enjoys faster convergence, even for corrupted datasets, highlighting the generalization capability of our meta-model.

## 5.2 Reward fine-tuning

**Adjoint matching.** Continuing from the exposition in Section 4, we note that our noise injection is different from the noise schedule assumed in the original derivations (Domingo-Enrich et al., 2025). As such, we modified the computations slightly as given in App. F.4. However, since the conventional flow matching algorithm injects a very small noise in the Gaussian path, i.e. $p_t(x_t|x_0, x_1) = \mathcal{N}(\mu_t, \sigma_t^2 I)$, where practically $\sigma_t = 10^{-3}$, we found that divisions by $\sigma_t$ in the adjoint matching algorithm will explode quantities in the loss. To resolve this issue, we derived a deterministic version of the adjoint matching algorithm, which we found to work much better with more sensible norms. Further, we derived a multi-marginal variant of the adjoint matching algorithm, however, we focus our resources only on the CFM case. All computations may be found in App. F.4.

Table 2: Mean validation accuracy of top-5 N$\mathcal{M}$ model retrievals. A single meta-model is used for all base datasets, with a conditioning signal obtained from image samples used to distinguish between each set.

| | CIFAR10 | STL10 | MNIST | FMNIST |
|---|---|---|---|---|
| Original | 63.38 | 53.88 | 98.93 | 89.77 |
| N$\mathcal{M}$-CFM | 62.89 | 53.47 | 98.69 | 90.24 |
| N$\mathcal{M}$-CFM w/ VAE | 62.79 | 53.41 | 98.54 | 90.59 |
| N$\mathcal{M}$-MMFM | 53.45 | 49.62 | 92.86 | 76.55 |
| N$\mathcal{M}$-MMFM w/ VAE | 63.87 | 52.86 | 98.36 | 89.73 |
| N$\mathcal{M}$-JKO[1] | 62.04 | 51.16 | 97.44 | 87.75 |

**Support of classifier weights.** Notably, the method of reward fine-tuning *cannot* be applied for arbitrary meta-model fine-tuning. Indeed, this follows from the fact that supp $p_1^* =$ supp $p_1^{\text{base}}$. Empirically, we found that fine-tuning a N$\mathcal{M}$-CFM model trained to generate CIFAR10 classifiers into one that generates STL10 classifiers fails at achieving reasonable accuracies, often just slightly better than a Kaiming uniform initialization. This suggests that the support of classifier weights, trained for different datasets, are mostly

---

[1]JKOnet expects the trajectory to evolve via gradient flow. This is not guaranteed in the latent space, hence we only show the un-encoded variant.

Table 3: Mean validation accuracy over top-5 fine-tuned generated weights post-retrieval. Setup follows Table 2, excepting the target weight data. tilde indicates corrupted datasets, on which the model was not trained.

| Epoch | Method | $\widetilde{\text{CIFAR10}}$ | $\widetilde{\text{STL10}}$ | $\widetilde{\text{MNIST}}$ | $\widetilde{\text{FMNIST}}$ | CIFAR10 | STL10 | MNIST | FMNIST |
|---|---|---|---|---|---|---|---|---|---|
| 0 | RandomInit | $\sim 10\%$ | $\sim 10\%$ | $\sim 10\%$ | $\sim 10\%$ | $\sim 10\%$ | $\sim 10\%$ | $\sim 10\%$ | $\sim 10\%$ |
| | N$\mathcal{M}$-CFM w/ VAE | $39.66 \pm 0.08$ | $36.85 \pm 0.36$ | $90.93 \pm 0.35$ | $72.44 \pm 0.27$ | $44.63 \pm 0.05$ | $40.06 \pm 0.02$ | $95.57 \pm 0.05$ | $83.68 \pm 0.24$ |
| | N$\mathcal{M}$-MMFM w/ VAE | $39.08 \pm 0.22$ | $21.98 \pm 0.89$ | $90.55 \pm 0.23$ | $74.08 \pm 0.08$ | $41.89 \pm 0.37$ | $36.41 \pm 1.54$ | $90.02 \pm 1.09$ | $83.38 \pm 0.29$ |
| | N$\mathcal{M}$-JKO | $40.53 \pm 0.27$ | $18.83 \pm 0.30$ | $90.80 \pm 0.20$ | $74.45 \pm 0.15$ | $45.04 \pm 0.19$ | $41.76 \pm 0.21$ | $95.42 \pm 0.22$ | $83.90 \pm 0.04$ |
| 1 | RandomInit | $34.05 \pm 1.13$ | $16.64 \pm 1.61$ | $83.21 \pm 0.57$ | $66.81 \pm 0.74$ | $36.27 \pm 2.05$ | $22.13 \pm 2.11$ | $96.52 \pm 0.20$ | $77.21 \pm 0.26$ |
| | N$\mathcal{M}$-CFM w/ VAE | $44.68 \pm 0.16$ | $38.18 \pm 0.40$ | $94.02 \pm 0.42$ | $76.43 \pm 0.22$ | $48.28 \pm 0.14$ | $41.66 \pm 0.19$ | $97.38 \pm 0.10$ | $85.27 \pm 0.30$ |
| | N$\mathcal{M}$-MMFM w/ VAE | $43.96 \pm 0.37$ | $29.44 \pm 1.25$ | $94.71 \pm 0.35$ | $79.85 \pm 0.47$ | $47.98 \pm 0.32$ | $39.93 \pm 1.45$ | $97.68 \pm 0.13$ | $85.50 \pm 0.12$ |
| | N$\mathcal{M}$-JKO | $45.04 \pm 0.31$ | $22.26 \pm 0.08$ | $94.94 \pm 0.17$ | $80.63 \pm 0.20$ | $49.61 \pm 0.29$ | $42.41 \pm 0.31$ | $97.39 \pm 0.02$ | $85.48 \pm 0.13$ |
| 5 | RandomInit | $46.55 \pm 0.80$ | $25.08 \pm 1.13$ | $92.53 \pm 0.28$ | $79.08 \pm 0.93$ | $47.74 \pm 1.33$ | $31.55 \pm 2.00$ | $98.24 \pm 0.03$ | $84.87 \pm 0.40$ |
| | N$\mathcal{M}$-CFM w/ VAE | $47.41 \pm 0.13$ | $39.98 \pm 0.29$ | $95.22 \pm 0.09$ | $79.88 \pm 0.61$ | $51.69 \pm 0.14$ | $42.62 \pm 0.08$ | $98.04 \pm 0.05$ | $86.76 \pm 0.15$ |
| | N$\mathcal{M}$-MMFM w/ VAE | $47.06 \pm 0.45$ | $35.24 \pm 0.72$ | $95.92 \pm 0.19$ | $82.08 \pm 0.11$ | $51.70 \pm 0.21$ | $41.35 \pm 0.65$ | $98.51 \pm 0.02$ | $86.64 \pm 0.14$ |
| | N$\mathcal{M}$-JKO | $48.14 \pm 0.08$ | $26.00 \pm 0.28$ | $95.92 \pm 0.11$ | $82.87 \pm 0.19$ | $53.01 \pm 0.11$ | $43.38 \pm 0.29$ | $98.13 \pm 0.05$ | $86.75 \pm 0.04$ |
| 20 | RandomInit | $50.28 \pm 0.43$ | $33.63 \pm 0.99$ | $95.81 \pm 0.18$ | $82.36 \pm 0.42$ | $51.35 \pm 1.21$ | $44.16 \pm 1.28$ | $98.51 \pm 0.05$ | $88.25 \pm 0.69$ |
| | N$\mathcal{M}$-CFM w/ VAE | $52.25 \pm 0.18$ | $41.18 \pm 0.28$ | $96.41 \pm 0.08$ | $83.42 \pm 0.23$ | $55.66 \pm 0.23$ | $44.38 \pm 0.16$ | $98.25 \pm 0.05$ | $88.02 \pm 0.17$ |
| | N$\mathcal{M}$-MMFM w/ VAE | $52.57 \pm 0.73$ | $39.80 \pm 0.48$ | $97.01 \pm 0.21$ | $84.38 \pm 0.08$ | $55.85 \pm 0.76$ | $44.10 \pm 0.26$ | $98.85 \pm 0.03$ | $88.29 \pm 0.03$ |
| | N$\mathcal{M}$-JKO | $52.59 \pm 0.02$ | $34.53 \pm 0.35$ | $96.82 \pm 0.10$ | $84.89 \pm 0.20$ | $56.65 \pm 0.27$ | $45.06 \pm 0.21$ | $98.43 \pm 0.03$ | $88.10 \pm 0.14$ |
| 30 | RandomInit | $52.99 \pm 0.55$ | $37.79 \pm 0.55$ | $96.55 \pm 0.22$ | $84.16 \pm 0.66$ | $56.05 \pm 1.21$ | $45.80 \pm 1.19$ | $98.55 \pm 0.05$ | $88.55 \pm 0.66$ |

disjoint. We stress that this property of the support is a function of both the downstream data *and* the model architecture. Indeed, due to the small size of the CNN3, the parameters that predict on different datasets e.g. CIFAR10 and STL10, will differ considerably, but this may not be the case for larger neural networks which possess a larger generalization set. To summarize our hypothesis: *the support set of CNN3 weights trained for different datasets are narrow and mostly disjoint, thus, small changes in the training data will noticeably affect the support w.r.t. validation accuracy.*

We found experimental evidence to support this hypothesis, and also to suggest that reward fine-tuning goes a long way towards improving validation accuracy on out-of-distribution data. Indeed, we find accuracies to be bounded above, often far below the validation accuracy obtained from SGD fine-tuning for the most corrupted data. We defer results and discussion to App. H.4, specifically Tables 12 and 13. Given this finding, we use it to approach the problem of harmful covariate shifts.

## 5.3 Detecting harmful covariate shifts

Table 4: True positive rate at the 5% significance level (TPR@5) and area under receiver operating characteristic curve (AUROC) for detection of harmful covariate shift on CIFAR10.1 and Camelyon17. We test on both the disagreement rate (DAR) and the entropy, setting $\lambda = \kappa/(|\mathbf{Q}|+1)$. See App. H for details on choosing $\kappa$. The best result for each column and our method are **bolded**.

| **TPR@5** | CIFAR10 | | | Camelyon | | |
|---|---|---|---|---|---|---|
| $|\mathbf{Q}|$ | 10 | 20 | 50 | 10 | 20 | 50 |
| Detectron (DAR), $\kappa = 1$ | 0 | 0 | 0 | 0 | $.10 \pm .10$ | 0 |
| Detectron (DAR), $\kappa$ vary | 0 | 0 | $.10 \pm .10$ | $.10 \pm .10$ | $.20 \pm .13$ | $.50 \pm .17$ |
| **Meta-detectron (DAR), $\kappa$ vary** | $.53 \pm .13$ | $.47 \pm .13$ | $.53 \pm .13$ | $.73 \pm .12$ | $\mathbf{.40 \pm .13}$ | $\mathbf{.68 \pm .10}$ |
| Detectron (Entropy), $\kappa = 1$ | $\mathbf{.60 \pm .17}$ | $.40 \pm .16$ | $.50 \pm .17$ | 0 | 0 | 0 |
| Detectron (Entropy), $\kappa$ vary | $.60 \pm .16$ | $.10 \pm .10$ | $.10 \pm .10$ | 0 | 0 | 0 |
| **Meta-detectron (Entropy), $\kappa$ vary** | $.47 \pm .13$ | $\mathbf{.93 \pm .07}$ | $\mathbf{1.00}$ | $\mathbf{1.00}$ | 0 | $.24 \pm .09$ |
| **AUROC** | CIFAR10 | | | Camelyon | | |
| $|\mathbf{Q}|$ | 10 | 20 | 50 | 10 | 20 | 50 |
| Detectron (DAR), $\kappa = 1$ | 0.515 | 0.595 | 0.485 | 0.59 | 0.595 | 0.795 |
| Detectron (DAR), $\kappa$ vary | 0.480 | 0.495 | 0.665 | 0.665 | 0.750 | 0.875 |
| **Meta-detectron (DAR), $\kappa$ vary** | **0.876** | 0.838 | 0.900 | 0.867 | 0.760 | **0.930** |
| Detectron (Entropy), $\kappa = 1$ | 0.740 | 0.695 | 0.850 | 0.345 | 0.610 | 0.720 |
| Detectron (Entropy), $\kappa$ vary | 0.775 | 0.740 | 0.785 | 0.490 | 0.445 | 0.660 |
| **Meta-detectron (Entropy), $\kappa$ vary** | 0.809 | **0.987** | **1.000** | **1.000** | **0.836** | 0.755 |

Table 5: In-distribution validation accuracy before and after meta-detectron training at various $|\mathbf{Q}|$ sizes.

| | CIFAR10 | | |
|---|---|---|---|
| $|\mathbf{Q}|$ | 10 | 20 | 50 |
| Meta-detectron ($\mathbf{P}^*$), $\kappa$ vary | $60.78 \pm 0.21 \to 61.09 \pm 0.17$ | $60.97 \pm 0.16 \to 60.49 \pm 0.25$ | $60.78 \pm 0.09 \to 60.38 \pm 0.24$ |
| Meta-detectron ($\mathbf{Q}$), $\kappa$ vary | $60.97 \pm 0.15 \to 60.76 \pm 0.26$ | $61.61 \pm 0.30 \to 61.11 \pm 0.13$ | $61.14 \pm 0.15 \to 60.59 \pm 0.26$ |

| | Camelyon | | |
|---|---|---|---|
| $|\mathbf{Q}|$ | 10 | 20 | 50 |
| Meta-detectron ($\mathbf{P}^*$), $\kappa$ vary | $92.10 \pm 0.35 \to 92.12 \pm 0.46$ | $92.09 \pm 0.54 \to 91.95 \pm 0.48$ | $92.31 \pm 0.21 \to 90.43 \pm 0.82$ |
| Meta-detectron ($\mathbf{Q}$), $\kappa$ vary | $92.36 \pm 0.34 \to 91.89 \pm 0.66$ | $90.61 \pm 1.45 \to 91.38 \pm 0.23$ | $92.66 \pm 0.24 \to 91.01 \pm 0.36$ |

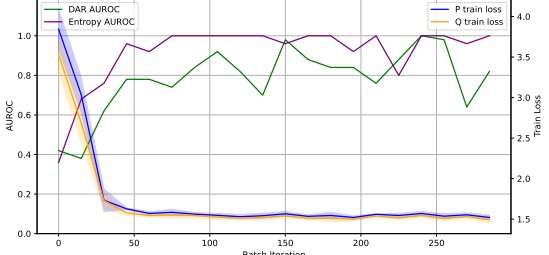 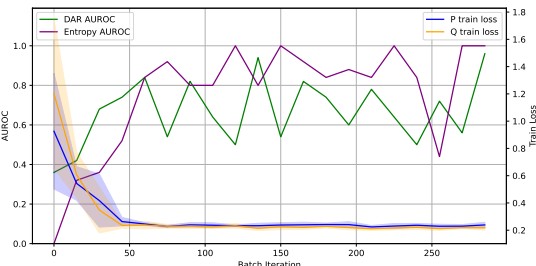

Figure 4: Plots illustrating how AUROC and $\ell_{cdc}$ evolves over meta-detectron training iterations for CIFAR10 and Camelyon17 when $|\mathbf{Q}| = 20$. See App. H.5 for more figures

We evaluate Meta-detectron (Sec. 4.2) on CNN3 with experiments following Ginsberg et al. (2023) on CIFAR10.1 (Recht et al., 2019), where shift comes from the dataset pipeline, and Camelyon17 (Veeling et al., 2018), which consists of histopathological slides from multiple hospitals. Table 4 shows the *True Positive Rate at 5% Significance Level (TPR@5)* and *AUROC* aggregated over 10 randomly chosen seeds for sampling $\mathbf{P}^*$ and $\mathbf{Q}$ of varying sample sizes. In addition, we ablated over the weight $\lambda$; see App. H.5 for details and further results. Compared to the original tests (Ginsberg et al., 2023, Table 1) on ResNet-18, we observe that covariate shift is highly architecture dependent. This is expected as CNN3 underfits CIFAR10 ($\sim 63\%$ validation accuracy). Our approach accounts for this as the base classifiers are generated directly by the fine-tuned meta-models. We also observe–though not shown–lower disagreement rates overall, which pays off in the TPR@5 as the $\mathbf{P}^*$ disagreement rates are close to zero in all cases, and confirms the conservative nature of our method. Importantly, we also observe in Table 5 that the validation accuracy on $\mathbf{P}$ is mostly unchanged. Regarding meta-training behavior, Figure 4 shows that the AUROC increases sharply early in the reward fine-tuning phase, requiring only about 50 batch iterations to reach its peak. This coincides with a marked decrease in $\ell_{cdc}$. However, we also note some instability in the AUROC throughout training, particularly in the Camelyon experiments, where fluctuations are more pronounced.

# 6 Related work

**Flow matching for trajectory inference.** The flow matching framework (Albergo and Vanden-Eijnden, 2023; Lipman et al., 2023; Liu et al., 2023) gives way to a few methods of controlling the trajectory of the inference path, from the simple multi-marginal approach (Rohbeck et al., 2025), to approaches with more sophisticated interpolants (Neklyudov et al., 2023; 2024; Kapusniak et al., 2024; Pooladian et al., 2024; Rohbeck et al., 2025). A traditional application of trajectory inference is single cell RNA-sequencing (Tong et al., 2020; Neklyudov et al., 2024; Kapusniak et al., 2024), however, a similar problem arises in weight generation. For a broad mathematical overview, see (Lavenant et al., 2024).

**Weight generation.** Recent approaches to generating neural network parameters centers around learning a distribution over pre-trained weights and generalizing this ability with the help of conditioning information.

Most similar to our approach in this regard are the various diffusion-based approaches (Soro et al., 2025; Zhang et al., 2024; Wang et al., 2024) which have been used to generate neural network weights with a focus on in-context learning tasks such as zero- and few-shot learning. However, flexibility is limited by its restriction to Gaussian processes and a sluggish inference speed. More broadly, we may categorize this form of learning as meta-learning (Fifty et al., 2024; Hu et al., 2022; Zhmoginov et al., 2022), which aims to learn concepts from a few demonstrations. It is therefore natural that the literature has two evaluation settings: in-distribution tasks and out-of-distribution (OOD) tasks. With enough training and capacity, it's clear *meta-models* (i.e. models trained on multiple data distributions) should excel at in-distribution tasks. However, generalization to novel tasks often presents a challenge to meta-learning and weight generation frameworks (Wang et al., 2024; Schürholt et al., 2024; Soro et al., 2025), including non-diffusion-based approaches Knyazev et al. (2021; 2023). See Appendix C for further related works.

## 7 Conclusion, Limitations, and Future Work

In this work, we have provided a preliminary investigation of the latest dynamical generative models for weight generation with applications to covariate shift detection. Due to the large size of modern neural network architectures, limited resources constrain our study to architectures with $< 10^6$ parameters. To address other concerns such as training dataset diversity and the lack of experiments incorporating stochastic weight evolution, future research directions include: **1)** exploration of equivariant architectures to reduce dimensionality of weight space, **2)** incorporating Schrödinger bridge matching approaches to address stochastic weight evolution. Moreover, the methods here open up a plethora of other applications. For instance, we may experiment with **1)** more traditional meta-learning tests such as zero- and few-shot learning; **2)** model merging by superposition of the inference ODE/SDE (Skreta et al., 2025); or **3)** network constrained problems such as generating binary or Lipschitz neural networks.

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

## Appendix

This appendix consist of details left out in the main text. First, we complete the proofs of the results in the text. Next, we perform a more comprehensive review of the related literature. Following, we fill in a few technical results that were promised in the main text, before closing with architecture and training settings.

## A    Further Theory

### A.1    Proof of Theorem 1

**Theorem 1**:
Let $\theta_0 \sim p_0$ be the initialized network parameters residing on $\Omega \subset \mathbb{R}^p$ a compact set. Suppose that $\mathcal{L} : \Omega \to \mathbb{R}$ is $C^1$ and the gradient descent curve $(\theta_t)_{t \geq 0}$ reside in $\Omega$. Define $p_t = \text{Law}(\theta_t)$ and further assume $p_t > 0$ a.e., then it satisfies the continuity equation Eqn. 3.

*Proof of Theorem 1.* We follow the reasoning of Santambrogio (2015, Ch. 8). First, we provide context as for the discretized formulation of a gradient flow. Let $F : \mathbb{R}^d \to \mathbb{R}$ be lower semi-continuous and is bounded below as $F(x) \geq C_1 - C_2 |x|^2$ for some $C_1, C_2 \geq 0$. Consider the formal problem

$$\begin{cases} x'(t) = -\nabla F(x) \\ x(0) = x_0 \end{cases} \quad .$$

This is understood as a Cauchy problem which happens to be a gradient flow. If we fix a small time step $\tau > 0$, this problem has a discretization

$$x_{k+1}^\tau \in \underset{x \in \mathbb{R}^d}{\arg\min} \, F(x) + \frac{|x - x_k^\tau|^2}{2\tau}$$

We now generalize this discretization scheme and show that its limit is a solution to the gradient flow. Define $J(p) := \int_\Omega \mathcal{L} \, dp$. We now have the scheme

$$p_{k+1}^\tau \in \arg\min_{p \in \mathcal{P}(\Omega)} J(p) + \frac{W_2^2(p, p_k^\tau)}{2\tau}.$$

**Claim:** The minimization above produces a minimizer $p = p_{k+1}^\tau$. If we modify

$$\tilde{J}(p) = \begin{cases} J(p) & p \ll \text{Leb}^p \\ +\infty & \text{otherwise} \end{cases}$$

then this is a unique minimizer.

*Proof.* As $\Omega$ is compact, it's a well-known result that $\mathcal{P}(\Omega)$ is compact in the weak topology (Parthasarathy, 2005). Moreover, Santambrogio (2015, Prop. 7.1) gives lower semi-continuity of $J$, which is enough for existence. Moreover, from Santambrogio (2015, Prop. 7.19), we have: if $p \ll \text{Leb}^p$, then $W_2^2(\cdot, p)$ is strictly convex. Since $J(p)$ is linear in $p$, we have that the minimization objective is strictly convex with the modification to $\tilde{J}$, thus the minimizer is unique. $\square$

**Note:** due to the nice properties of having $p \ll \text{Leb}^p$, and considering that we do not lose much generality, we will use $\tilde{J}$ for the rest of the proof.

**Claim:** The first variation $\frac{\delta J}{\delta p}(p) = \mathcal{L}$ for all $p \in \mathcal{P}(\Omega)$.

*Proof.* We note that since $\Omega$ is compact, $\mathcal{L}$ is presumed continuous, and $p$ a finite measure, we have that $J(p) < \infty$ for all $p \in \mathcal{P}(\Omega)$. This is sufficient to show that $p$ is always regular for $J$. Moreover, the first variation satisfies:

$$\frac{d}{d\epsilon}\bigg|_{\epsilon=0} J(p + \epsilon\chi) = \int \frac{\delta J}{\delta p}(p) \, d\chi.$$

By linearity: $J(p + \epsilon\chi) = J(p) + \epsilon \int \mathcal{L} \, d\chi$, and therefore $\frac{\delta J}{\delta p}(p) = \mathcal{L}$. $\square$

It was remarked Santambrogio (2015, Remark 7.13) that the first variation of the transport cost $\mathcal{T}_c$ for a continuous cost $c$ is exactly the Kantorovich potential $\varphi$ *only if it is unique.* In our case, uniqueness stems from Santambrogio (2015, Prop. 7.18). Given our preparation, we are now ready to state the main Lemma.

**Lemma 1.** *Let $T_{k+1}^\tau$ be the optimal transport map from $p_{k+1}^\tau$ to $p_k^\tau$ (note the reverse), then we have the velocity*

$$v_{k+1}^\tau := \frac{\iota - T_{k+1}^\tau}{\tau} = -\nabla\mathcal{L} \quad a.e. \tag{14}$$

*Proof.* Since the first variation is linear in the functional argument, we have that the first variation

$$\frac{\delta(J + W_2^2(\cdot, p_k^\tau)/2\tau)}{\delta p}(p) = \mathcal{L} + \varphi/\tau.$$

By Santambrogio (2015, Prop. 7.20), we have $\mathcal{L} + \varphi/\tau = C$ a.e. where $\varphi$ is the Kantorovich potential and some constant $C$ (precisely, on all $supp\, p_{k+1}^\tau$, which is assume $> 0$ a.e.). Differentiating, we have

$$\nabla\varphi = \iota - T_{k+1}^\tau = -\tau\nabla\mathcal{L} \quad a.e.$$

$\square$

**Remark.** The reader may notice that the equation above for $\nabla \varphi$ resembles the JKOnet\* objective presented in Sec. 3.2.2. Terpin et al. (2024) uses the first-order necessary conditions for optimality, which bypasses checking that the learned gradient function defines a transport map and the requirement that the scalar function be convex. The original JKOnet (Bunne et al., 2022), which requires a bi-level optimization objective, is more explicit in following the transport map, negatively affecting efficiency and stability.

Before proceeding further, we provide a simple bound on the 2-Wasserstein distance of consecutive iterates: by optimality

$$J(p_{k+1}^\tau) + \frac{W_2^2(p_{k+1}^\tau, p_k^\tau)}{2\tau} \leq J(p_k^\tau),$$

therefore

$$\sum_k \frac{W_2^2(p_{k+1}^\tau, p_k^\tau)}{2\tau} \leq \sum_k 2(J(p_k^\tau) - J(p_{k+1}^\tau)) \leq 2J(p_0) =: C.$$

where we telescoped the sum and used $\inf J \geq 0$ as $\mathcal{L} \geq 0$.

For us to take a limit, we need to fine $p_t^\tau$ for values in $(k\tau, (k+1)\tau)$. Following Santambrogio (2015, Ch. 8), take

$$p_t^\tau = \left( \frac{k\tau - t}{\tau} v_k^\tau + \iota \right)_\# p_k^\tau \quad \text{for } t \in ((k-1)\tau, k\tau).$$

Moreover, $v_t^\tau$ ought to be defined so that it advects $p^\tau$ over time, and following the intuition of interpolating between discrete values, we require

$$||v_t^\tau||_{L^2(p_t^\tau)} = |(p^\tau)'|(t) = \lim_{t' \to 0} \frac{W_2(p_{t+t'}^\tau, p_t^\tau)}{|t'|} = \frac{W_2(p_{k-1}^\tau, p_k^\tau)}{\tau}.$$

In fact,

$$v_t^\tau = \frac{\iota - T_k^\tau}{\tau} \circ ((k\tau - t)v_k^\tau + \iota)^{-1}$$

works. Define the momentum $E^\tau = p^\tau v^\tau$.

**Remark.** Note the similarity of the approximation method to the MMFM intermediate densities in Sec. 3.2.2. For simplicity of our objective, the interpolants $p_t^\tau$ are defined by requiring the conditional $p_t^\tau(x|x_{(k-1)\tau}, x_{k\tau}) = \mathcal{N}(x; (1-\beta_t)x_{(k-1)\tau} + \beta_t x_{k\tau}, \sigma_t I)$ where $\beta_t := \frac{t-(k-1)\tau}{\tau}$. In contrast, the density approximations here require a form on $||v_t^\tau||_{L^2(p_t^\tau)}$.

Now we bound the maximal $E^\tau$:

$$|E^\tau|([0,1] \times \Omega) = \int_0^1 \int_\Omega |v_t^\tau| \, dp_t^\tau \, dt = \int_0^1 ||v_t^\tau||_{L^1(p_t^\tau)} \, dt \leq \int_0^1 ||v_t^\tau||_{L^2(p_t^\tau)} \, dt.$$

Using the Cauchy-Schwarz inequality

$$\int_0^1 ||v_t^\tau||_{L^2(p_t^\tau)} \, dt \leq \int_0^1 ||v_t^\tau||_{L^2(p_t^\tau)}^2 \, dt = \sum_k \tau \left( \frac{W_2(p_{k-1}^\tau, p_k^\tau)}{\tau} \right)^2 \leq C^2. \tag{15}$$

Moreover, we see for $0 < s < t < 1$,

$$W_2(p_t^\tau, p_s^\tau) \leq \int_s^t |(p^\tau)'|(r) \, dr \leq (t-s)^{1/2} \left( \int_s^t |(p^\tau)'|(r)^2 \, dr \right)^{1/2}$$

But since $|(p^\tau)'|(r)^2 = ||v_r^\tau||_{L^2(p_t^\tau)}^2$, we have

$$W_2(p_t^\tau, p_s^\tau) \leq C^2(t-s)^{1/2}. \tag{16}$$

This provides a uniform Hölder bound on $t \mapsto p_t^\tau$, which implies uniform boundedness and equicontinuity. Therefore, up to subsequences, both $E^\tau$ and $p^\tau$ converges to a limit as $\tau \to 0$: by Arzela-Ascoli, we have

$p^\tau \to p$ uniformly on $W_2$. Moreover, by boundedness of $|E^\tau|$, we have weak compactness on the space of measures, and therefore, there exists a subsequence s.t. $E^\tau \to E$ weakly.

Now, checking the distributional test, e.g. Santambrogio (2015, Definition 4.1), we have that $\partial_t p_t^\tau + \nabla \cdot E^\tau = 0$. Moreover, the distributional test passes to the limit as $v_t^\tau$ is continuous, $\int\int ||v_t^\tau|^2 dp_t^\tau dt \le C^2$, and $p^\tau \to p$ uniformly, and $E^\tau \to E$ weakly. In other words, we have $\partial_t p_t + \nabla \cdot E = 0$. We are left to compute $E$.

**Claim:** $E = -p\nabla\mathcal{L}$.

The trick here is to instead consider simpler curves than the interpolation we defined above. In particular, consider $\tilde{p}_t^\tau = p_{k+1}^\tau$ for $t \in (k\tau, (k+1)\tau)$ and $\tilde{v}_t^\tau = v_{k+1}^\tau$. Likewise, define $\tilde{E}^\tau = \tilde{p}^\tau \tilde{v}^\tau$. Since

$$\frac{W_2(p_{k+1}^\tau, p_k^\tau)}{\tau} = \frac{1}{\tau}\left(\int |\iota - T_{k+1}^\tau|^2 \, dp_{k+1}^\tau\right)^{1/2} = ||\tilde{v}_{k+1}^\tau||_{L^2(p_{k+1}^\tau)},$$

we have $||\tilde{v}_{k+1}^\tau||_{L^2(p_t^\tau)} = ||v_{k+1}^\tau||_{L^2(p_t^\tau)}$. This implies that $\tilde{E}^\tau$ has the same bound as in Eqn. 15. Moreover, by using Eqn. 16, we see $W_2(p_t^\tau, \tilde{p}_t^\tau) \le C^2\sqrt{\tau}$. Hence, we have the same convergence: $\tilde{p}^\tau \to p$ uniformly. One needs to be more careful with the momentum. Let $\tilde{E}^\tau \to \tilde{E}$ weakly, we look to prove $\int f \cdot d\tilde{E} = \int f \cdot dE$ for all $f : [0, 1] \times \Omega \to \mathbb{R}^p$ Lipschitz. We compute:

$$\left|\int f \cdot d\tilde{E}^\tau - \int f \cdot dE^\tau\right| \le \int_0^1 \int_\Omega |f \circ ((k\tau - t)v_{\kappa(t)}^\tau + \iota) - f| |v_{\kappa(t)}^\tau| \, dp^\tau \, dt \le Lip(f)\tau \int_0^1 \int_\Omega |\tilde{v}_t^\tau|^2 \, dp^\tau \, dt,$$

where $\kappa(t)$ returns the smallest multiple $k\tau \ge t$, and the RHS is bounded by $Lip(f)C^2\tau$. Thus, as $\tau \to 0$, we see that $E = \tilde{E}$.

But now, $\tilde{E}^\tau = \tilde{v}^\tau \tilde{p}^\tau = -\tilde{p}^\tau \nabla\mathcal{L}$ by Lemma 1. Therefore, for any $f \in C_c^1((0, 1) \times \Omega, \mathbb{R}^p)$, we test:

$$\int f \cdot d\tilde{E}^\tau = -\int f \cdot \nabla\mathcal{L}p^\tau.$$

As $\mathcal{L} \in C^1$, we know the integrand is continuous, so we can just pass the limit as $\tau \to 0$, meaning that $\tilde{E} = E = -p\nabla\mathcal{L}$ invoking the fact that $\tilde{p}^\tau \to p$ uniformly. □

## A.2 Proofs of Sec. 3.2.2

**Theorem 2:**
Suppose the true marginals evolve according to $\frac{d}{dt}p_t^* = -\nabla \cdot (p_t^* \nabla s_t^*)$ and $t \mapsto p_t^*$ is an absolutely continuous curve. Define $q(z)$ such that marginalizing $q$ with respect to all variables except $x_k, x_{k+1}$ yields the coupling $p_{t_k} \otimes (T_k^{k+1})_\# p_{t_k}$, where $T_k^{k+1}$ is the transport map from $p_{t_k}$ to $p_{t_{k+1}}$. Then,

$$\lim_{|t_k - t_{k+1}| \to 0} \int_0^1 \mathbb{E}_{p_t(x)} ||u_t(x) - \nabla s_t^*(x)||_2^2 \, dt = 0.$$

Replacing $u_t$ with $\frac{x_{t+1} - x_t}{\tau}$, this shows that $\nabla V$ (Eqn. 8) regresses to the reference action in the limit.

*Proof of Theorem 2.* By absolute continuity of the curve and Brenier's theorem, the Monge map between $p_s, p_t$ exists for $0 \le s < t \le 1$. By Ambrosio et al. (2006, Prop. 8.4.6), we have that

$$\nabla s_t^* = \lim_{h \to 0} \frac{1}{h}(T^*(p_t^*, p_{t+h}^*) - \mathrm{id}), \tag{17}$$

where $T^*(p_t^*, p_{t+h}^*)$ is the unique transport map between densities $p_t^*$ and $p_{t+h}^*$. Fixing a small $h > 0$, define

$$d_t(x) := \sum_{m=0}^{M-1} \mathbf{1}_{[hm, h(m+1))}(t)\frac{T_m^*(x) - x}{h}, \quad \text{where } T_m^* = T^*(p_{hm}^*, p_{h(m+1)}^*).$$

The idea of our next steps is to instead consider the normed difference between $u_t$ and $d_t$, and later conclude by the triangle inequality.

Recall that

$$u_t(x) = \mathbb{E}_{q(z)} \frac{u_t(x|z)p_t(x|z)}{p_t(x)} = \sum_{k=0}^{K-1} \mathbf{1}_{[t_k, t_{k+1})}(t) \int u_t(x|x_k, T_k^{k+1}(x_k)) \frac{p_t(x|x_k, T_k^{k+1}(x_k))}{p_t(x)} \, dp_{t_k}(x_k), \quad (18)$$

where we used the assumed disintegration of $q(z)$. We compute

$$\mathbb{E}_{p_t(x)} ||u_t(x) - d_t(x)||_2^2 = \int ||u_t(x) - d_t(x)||_2^2 \, dp_t(x) \tag{19}$$

$$= \int \left\| \int \left[ \frac{T_k^{k+1}(x_k) - x_k}{t_{k+1} - t_k} - \frac{T_m^*(x) - x}{h} \right] \frac{p_t(x|x_k, T_k^{k+1}(x_k))}{p_t(x)} \, dp_{t_k}(x_k) \right\|_2^2 \, dp_t(x) \tag{20}$$

$$\leq \int \int \left\| \frac{T_k^{k+1}(x_k) - x_k}{t_{k+1} - t_k} - \frac{T_m^*(x) - x}{h} \right\|_2^2 \frac{p_t(x|x_k, T_k^{k+1}(x_k))}{p_t(x)} \, dp_{t_k}(x_k) \, dp_t(x). \tag{21}$$

assuming that $t \in [t_k, t_{k+1})$ and using the fact

$$d_t(x) = \int d_t(x) \frac{p_t(x|x_k, T_k^{k+1}(x_k))}{p_t(x)} \, dp_t(x_k).$$

**Claim:** The reference $p_t$ is an absolutely continuous curve in $W_2$-space.

*Proof.* Let $0 \leq s < t \leq 1$ be given. It is known that affine Gaussian paths are absolutely continuous, in particular this means that there exists $g_k \in L^1([0,1])$ such that $W_2(p_{t_k}, p_{t_{k+1}}) \leq \int_{t_k}^{t_{k+1}} g_k(\tau) \, d\tau$. If $[s,t] \subset [t_k, t_{k+1}]$, then we conclude. Otherwise, consider $W_2(p_s, p_t) \leq W_2(p_s, p_{t_k}) + W_2(p_{t_k}, p_t)$. $\square$

The continuity claim allows us to take the limit: $W_2^2(p_{t_k}, p_t) \to 0$ as $|t_k - t| \to 0$. Moreover, by Santambrogio (2015, Theorem 5.10), we have $p_{t_k} \rightharpoonup p_t$, i.e. narrow convergence.

**Claim:** Suppose $p_{t_k} \rightharpoonup p_{hm}^*$ and $p_{t_{k+1}} \rightharpoonup p_{h(m+1)}^*$. We have $||T^*(p_{t_k}, p_{t_{k+1}}) - T^*(p_{hm}, p_{h(m+1)})||_{L^2(p_t)} \to 0$.

*Proof.* Let $t < \min(t_k, h(m+1))$. We first prove this claim for $T^*(p_t, p_{t_{k+1}}) \to T^*(p_t, p_{h(m+1)})$. Villani (2008, Cor. 5.23) gives convergence in measure (in this case, $p_t$ is the measure). Thus, the argument for $L^2(p_t)$ convergence follows from a typical analysis argument. To simplify notation, let $T \equiv T_t^{h(m+1)}$ and $(T_n)_{n \geq 1}$ the approaching sequence.

Suppose not, i.e. there exists a subsequence $(T_{n_i})$ such that $||T_{n_i} - T||_{L^2(p_t)} \geq \epsilon$ for some $\epsilon > 0$. By convergence in $p_t$-measure, there exists a further subsequence $(T_{n_{i_j}})$ that converges to $T$ pointwise $p_t$-a.e. Since $T_n, T \in L^2(p_t)$, by the Dominated Convergence Theorem $||T_{n_{i_j}} - T||_{L^2(p_t)} \to 0$, and hence $\int |T_{n_{i_j}}(x)| \, dp_t(x) \to \int |T(x)| \, dp_t(x)$. At this point, we have proven that for every subsequence (taken implicitly) of integral $\int |T_n(x)| \, dp_t(x)$, there exists a further subsequence that converges to the limit. It is a well-known result in analysis that this shows convergence of the original sequence of integrals, i.e. $||T_n - T||_{L^2(p_t)} \to 0$.

We now apply the previous result three times to the following and conclude:

$$||T^*(p_{t_k}, p_{t_{k+1}}) - T^*(p_{hm}, p_{h(m+1)})||_{L^2(p_t)} \leq ||T^*(p_{t_k}, p_{t_{k+1}}) - T^*(p_{t_k}, p_t)||_{L^2(p_t)}$$
$$+ ||T^*(p_{t_k}, p_t) - T^*(p_{hm}, p_t)||_{L^2(p_t)}$$
$$+ ||T^*(p_{hm}, p_t) - T^*(p_{hm}, p_{h(m+1)})||_{L^2(p_t)}$$

$\square$

To finish, we remind the reader that the densities $p_{t_k} = p_{t_k}^*$, therefore, the supposition of this claim is fulfilled by the observation of narrow convergence: $p_{t_k}^* \rightharpoonup p_{hm}^*$ and $p_{t_{k+1}}^* \rightharpoonup p_{h(m+1)}^*$. Therefore, combining the claim with Eqn. 21, we can make $\mathbb{E}_{p_t(x)}||u_t(x) - d_t(x)||_2^2 < \delta/2$ for some small $\delta > 0$. Then, if we had chosen a small enough $h$, we would have $\mathbb{E}_{p_t(x)}||\nabla s_t^*(x) - d_t(x)||_2^2 < \delta/2$. Combining these bounds, we conclude.

$\square$

## A.3   Extended Sec. 3.2.3

Given its importance as the reference flow we will match during FM training, we discuss how close of a *proxy curve* (generalization of $\mu_t$ in Eqn. 9) we can obtain within a family of paths that minimizes an energy functional. This will include methods such as Metric Flow Matching (Kapusniak et al., 2024), but also shares similarities with GSBM (Liu et al., 2024c). To motivate this, note that one could "sample" from $p_t$, by saving neural network parameters over the course of training on different initial samples $\theta_0 \sim p_0$. This approach can be used to build a dataset of intermediate weights $\mathcal{D} = \bigcup_{t \in [0,1]} \mathcal{D}_t$ of weights saved over the course of training. Following Kapusniak et al. (2024), if we define a data-dependent metric $g : \mathbb{R}^p \to \mathcal{S}_{++}(p)$, which is a smooth map parameterized by the dataset $\mathcal{D}$, we may compute a smooth energy-minimizing curve $\gamma_{\mathbf{x}_0,\mathbf{x}_1} := \arg\min_{\gamma_0=\mathbf{x}_0,\gamma_1=\mathbf{x}_1} \mathcal{E}_g(\gamma_t)$ between fixed points $(\mathbf{x}_0,\mathbf{x}_1) \sim \pi$ that can be shown to stay close to the data (Kapusniak et al., 2024, Proposition 1). Further, we may perform this minimization by training geodesic interpolants $\mathbf{x}_{t,\eta} \approx \gamma(t;\eta)$, see App. E for details. Following, we develop a framework using energy functionals, which are data-dependent in practice, motivated by the connection between the metric and potential approach (Kapusniak et al., 2024, App. C.1): $||\dot{x}_t||_{g(x_t)} = ||\dot{x}_t||_2^2 + V(x_t; x_0, x_1)$, where $V$ is a potential function depending on the boundary conditions. In this setting, we seek to characterize choices of energy $V$ to minimize the $W_2$ distance between the proxy probability path, which evolves by $v_\theta$, and the reference $p_t$ in Eqn. 3. We start by writing down a continuity equation for the proxy path (cf. Theorem 1).

**Theorem 3**: Suppose the Lagrangian $L(x_t, \dot{x}_t, t) = ||\dot{x}_t||_2^2 + V_t(x_t, \dot{x}_t)$ is Tonelli and strongly convex in velocity. The Lagrangian optimal transport map $T$ exists between $p_0$ and $p_1$. Moreover, there exists a locally Lipschitz, locally bounded vector field $w$ s.t.

$$\partial_t \hat{p}_t + \nabla \cdot (\hat{p}_t w_t) = 0$$

satisfies $\hat{p}_t = \text{Law}(\gamma_t)$ where $\gamma$ is a random, smooth Lagrangian-minimizing curve and $(\gamma_0, \gamma_1)$ is an optimal coupling of $p_0, p_1$.

*Proof of Theorem 3.* First, let us recall the definition of a *Tonelli* Lagrangian. Following (Schachter, 2017), it satisfies:

1. $L$ does not depend on time.

2. $L$ is $C^2$.

3. $L$ is strictly convex in velocity.

4. There exists a constant $c_0$ and a function $\theta : \mathbb{R}^p \to \mathbb{R}$ with superlinear growth, i.e. $\lim_{|v| \to \infty} \theta(v)/|v| = \infty$, with $\theta \geq 0$ s.t. $L(x, v) \geq c_0 + \theta(v)$.

Then, as noted in Schachter (2017, Ch. 3.3), the Lagrangian optimal transport problem has a solution, specifically a map $T : \mathbb{R}^p \to \mathbb{R}^p$ which pushforwards $p_0$ to $p_1$. For our purposes, we should also note that there exists a unique optimal trajectory $\sigma : [0,1] \times \mathbb{R}^p \to \mathbb{R}^p$ s.t.

$$\sigma = \arg\inf_{\sigma:[0,1]\times\mathbb{R}^p\to\mathbb{R}^p} \left\{ \int_0^1 \int_{\mathbb{R}^p} L(\sigma(t,x), \dot{\sigma}(t,x)) \, dp_0(x) \, dt : \sigma(1,\cdot)_\# p_0 = p_1 \right\}.$$

Using Schachter (2017, Prop. 3.4.4), we have a velocity field $w : [0,1] \times \mathbb{R}^p \to \mathbb{R}^p$ satisfying $\dot{\sigma}(t,x) = w_t(\sigma(t,x))$ which is locally Lipschitz and locally bounded. Then, by Schachter (2017, Prop. 3.4.3), the path defined by

$\hat{p}_t = (\sigma_t)_\# p_0$ and the velocity field $w_t$ satisfies the continuity equation

$$\partial_t \hat{p}_t + \nabla \cdot (\hat{p}_t w_t) = 0$$

in the sense of distributions. All that's left is to show that $\hat{p}_t = \mathrm{Law}(\gamma_t)$ where $\gamma$ is drawn from the Lagrangian-minimizing curves. However, this follows from Villani (2008, Thm. 7.21) as $(\hat{p}_t)_{t \in [0,1]}$ minimizes

$$\mathbb{A}(p) = \inf_\sigma \int_0^1 \int_{\mathbb{R}^p} L(\sigma(t,x), \dot{\sigma}(t,x)) \, dp_0(x) \, dt,$$

applying the equivalence between (iii) and (i) of Thm. 7.21. $\qquad \square$

The following discussion will focus on quantifying closeness of the reference $p$ and $\hat{p}$. Our objective is to characterize functionals that would induce good proxy trajectories which remain close to Eqn. 3. Hence, we ought to assume some regularity for $V$, in particular, we want the Lagrangian to be Tonelli. In Prop. 2 below, this definition of the learned path is used to find an expression for the $W_2$ distance that accounts for the closeness of $w_t$ to the loss gradient $-\nabla \mathcal{L}$.

**Proposition 2** (Adapted from Cor. 5.25 Santambrogio (2015)). *Suppose that $(p, \hat{p})$ resides in a compact domain $\Omega \subset \mathbb{R}^p$ and suppose that $p_t, \hat{p}_t$ are absolutely continuous w.r.t. Lebesgue measure for every $t$. Further, if we assume $p, \hat{p}$ are absolutely continuous curves in $W_2(\Omega)$, then*

$$W_2^2(p_t, \hat{p}_t) = 2 \int_0^1 \int_\Omega (x - T_t(x)) \cdot (\nabla \mathcal{L}(x) + w_t(T_t(x))) \, dt \, dp_t(x), \tag{22}$$

*where $T_t$ is the optimal transport map between $p_t$ and $\hat{p}_t$ for the cost $|x - y|^2/2$.*

*Proof of Prop. 2.* The assumption aligns with Santambrogio (2015, Cor. 5.25), which gives:

$$\frac{d}{dt} W_2^2(p_t, \hat{p}_t) = 2 \int_\Omega (x - T_t(x)) \cdot (\nabla \mathcal{L}(x) - w_t(T_t(x))) \, dp_t(x).$$

Integrating both sides and noting that $p_0 = \hat{p}_0$ yields the desired result.

$\qquad \square$

Following the action matching discussion, we define a *proxy action gap* for the data-dependent energy that will be used for a more intuitive and practical $W_2$-bound; see App. D for an analogue with entropy-regularization.

$$AG(p, \hat{p}) := \frac{1}{2} \int_0^1 \mathbb{E}_\gamma ||\nabla \mathcal{L}(\gamma_t) + \dot{\gamma}_t||^2 \, dt, \tag{23}$$

where the expectation is taken over all Lagrangian-minimizing curves s.t. $(\gamma_0, \gamma_1)$ is an optimal coupling under the Lagrangian in Theorem 3.

We note that this definition is unconventional as it uses curves obtained from $\hat{p}$ in Theorem 3. However, we believe this formulation to better match the interpretation of finding a proxy trajectory that reflects evolution via GD minimization. Intuitively, our Lagrangian-minimizing curves ought to have derivatives close to the gradient descent direction along its length, and the objective is to vary the potential $V$ to minimize $AG(p, \hat{p})$. Next, we adapt a result from Neklyudov et al. (2023); Albergo and Vanden-Eijnden (2023) to bound the Wasserstein distance in terms of the action gap (cf. Prop. 1)

**Proposition 3.** *Suppose that $\nabla \mathcal{L}$ is uniformly Lipschitz in $x$ with Lipschitz constant $K$. Then,*

$$W_2^2(p_t, \hat{p}_t) \le e^{(1+2K)t} \int_0^t \mathbb{E}_\gamma ||\nabla \mathcal{L}(\gamma_s) + \dot{\gamma}_s||^2 \, ds. \tag{24}$$

*where the expectation is taken over all Lagrangian-minimizing curves s.t. $(\gamma_0, \gamma_1)$ is an optimal coupling under the Lagrangian in Theorem 3.*

*Proof of Prop. 3.* First, note from Villani (2008, Thm. 7.21) that as $p_t, \hat{p}_t$ are continuous paths, there exists dynamical optimal couplings of $(p_0, p_1)$, as defined in Villani (2008, Def. 7.20), which we shall denote $\gamma_*, \gamma$ respectively. In particular, we use that $p_t = \text{Law}(\gamma_*(t))$ and $\hat{p}_t = \text{Law}(\gamma(t))$, and that both $(\gamma(0), \gamma(1))$ and $(\gamma_*(0), \gamma_*(1))$ are distributed according to $\pi$ the optimal coupling between $(p_0, p_1)$. Understanding this, we may define

$$Q_t := \mathbb{E}_{(\gamma(0), \gamma(1)) \sim \pi} ||\gamma_*(t) - \gamma(t)||^2 = \mathbb{E}_{(\gamma_*(0), \gamma_*(1)) \sim \pi} ||\gamma_*(t) - \gamma(t)||^2 \geq W_2^2(p_t, \hat{p}_t). \tag{25}$$

Furthermore, as the reference path $(p_t)_{t \in [0,1]}$ follows the continuity equation Eqn. 3, $\dot{\gamma}(t) = -\nabla\mathcal{L}(\gamma_*(t))$. Thus, considering the time-derivative, we have

$$\frac{\partial Q_t}{\partial t} = 2 \int \langle \gamma(t) - \gamma_*(t), -\nabla\mathcal{L}(\gamma_*(t)) - \dot{\gamma}(t)) \rangle \, d\pi(\gamma_0, \gamma_1)$$

$$= 2 \int \langle \gamma(t) - \gamma_*(t), -\nabla\mathcal{L}(\gamma_*(t)) + \nabla\mathcal{L}(\gamma(t)) \rangle \, d\pi(\gamma_0, \gamma_1)$$

$$+ 2 \int \langle \gamma(t) - \gamma_*(t), -\nabla\mathcal{L}(\gamma(t)) - \dot{\gamma}(t) \rangle \, d\pi(\gamma_0, \gamma_1)$$

The first term may be bounded by Lipschitzness of $\nabla\mathcal{L}$:

$$2\langle \gamma(t) - \gamma_*(t), -\nabla\mathcal{L}(\gamma_*(t)) + \nabla\mathcal{L}(\gamma(t)) \rangle \leq 2K ||\gamma(t) - \gamma_*(t)||^2.$$

The second term may be bounded by:

$$||\gamma(t) - \gamma_*(t)||^2 - 2\langle \gamma(t) - \gamma_*(t), -\nabla\mathcal{L}(\gamma(t)) - \dot{\gamma}(t) \rangle + ||\nabla\mathcal{L}(\gamma(t)) + \dot{\gamma}(t)||^2 \geq 0,$$
$$2\langle \gamma(t) - \gamma_*(t), -\nabla\mathcal{L}(\gamma(t)) - \dot{\gamma}(t) \rangle \leq ||\gamma(t) - \gamma_*(t)||^2 + ||\nabla\mathcal{L}(\gamma(t)) + \dot{\gamma}(t)||^2$$

In summary,

$$\frac{\partial Q_t}{\partial t} \leq (1 + 2K)Q_t + \int ||\nabla\mathcal{L}(\gamma(t)) + \dot{\gamma}(t)||^2 \, d\pi(\gamma_0, \gamma_1),$$

then by Gronwall's inequality:

$$Q_t \leq \exp(t(1 + 2K)) \int_0^t \int ||\nabla\mathcal{L}(\gamma(t)) + \dot{\gamma}(t)||^2 \, d\pi(\gamma_0, \gamma_1),$$

using the fact that $Q_0 = 0$, and now we conclude by the fact that $W_2^2(p_t, \hat{p}_t) \leq Q_t$. $\qquad\square$

As the loss is arbitrary, it is not guaranteed that the action gap vanishes. For one, the smoothness assumption on $\gamma$ means that eccentric losses cannot be fit exactly. However, as detailed in App. E, we can, in practice, weaken the smoothness assumption to match a learned minimizing curve or the simpler cubic splines (motivated by Rohbeck et al. (2025)). We also remark that optimizing this functional $V$ is challenging in practice, requiring learned interpolants as discussed in App. E, or modeling a drift potential directly as in JKOnet (Bunne et al., 2022; Terpin et al., 2024). To conclude, we make use of a natural smoothness assumption on the gradient descent path to prove a bound on $AG$.

**Theorem 4.** *Define $g(\gamma, t) := ||\nabla\mathcal{L}(\gamma_t) + \dot{\gamma}_t||^2$. Suppose for each $(x_0, x_1) \sim \pi$ there exists a smooth connecting curve $\tilde{\gamma}$ s.t. $\sup_{0 \leq t \leq 1} g(\tilde{\gamma}, t) \leq \delta$ for some $\delta > 0$ and length $\int_0^1 ||\dot{\gamma}_t||^2 \leq \Gamma$. If there exists $\eta > \delta > 0$ s.t.*

*1. $V(\gamma_t, \dot{\gamma}_t) \geq 2\Gamma g(\gamma, t)$ for $g(\gamma, t) \geq \eta$, and*

*2. $V(\gamma_t, \dot{\gamma}_t) \leq \min\{\Gamma/\delta, \Gamma\} g(\gamma, t)$ if $g(\gamma, t) \leq \delta$,*



then $\sup_{0 \le t \le 1} g(\gamma_*, t) \le 2\eta$, where $\gamma_*$ is the Lagrangian-minimizing curve.



*Proof of Theorem 4.* Suppose not, so that there exists $t_0 \in [0, 1]$ s.t. $g(\gamma_*, t_0) = ||\dot{\gamma}_*(t_0) + \nabla\mathcal{L}(\gamma_*(t_0))||^2 > 2\eta$. By continuity of $\gamma_*$ and $\nabla\mathcal{L}$ w.r.t. time, we have $0 \le t' \le t_0$ s.t. for any $t \in [t', t_0]$, $g(\gamma_*, t) \ge \eta$ and $t'$ is chosen s.t. $g(\gamma_*, t') = \eta$. By our assumption on $\eta$, it's natural that we look at the action under two different cases. First, if $g(\gamma_*, t) \ge \eta$, we have:

$$
\begin{aligned}
A(\gamma_*) &\ge \int_{t'}^{t_0} ||\dot{\gamma}_*(t)||_2^2 + V(\gamma_*(t), \dot{\gamma}_*(t)) \, dt \\
&\ge \int_{t'}^{t_0} ||\dot{\gamma}_*(t)||_2^2 + 2\Gamma ||\dot{\gamma}_*(t) + \nabla\mathcal{L}(\gamma_*(t))||_2^2 \, dt \\
&> ||\gamma_*(t') - \gamma_*(t_0)||^2 + 2\Gamma\eta \ge 2\Gamma\eta.
\end{aligned}
$$

Otherwise, we have by assumption that there exists a smooth connecting curve $\gamma$ s.t. $d_\nabla(\gamma) \le \delta < \eta$, hence

$$
\begin{aligned}
A(\gamma) &= \int_0^1 ||\dot{\gamma}(t)||_2^2 + V(\gamma(t), \dot{\gamma}(t)) \, dt \\
&\le \Gamma + \int_0^1 ||\dot{\gamma}_*(t) + \nabla\mathcal{L}(\gamma_*(t))||_2^2 \, dt \\
&\le \Gamma + \delta \cdot \Gamma/\delta = 2\Gamma.
\end{aligned}
$$

By minimality of the Lagrangian-minimizing curve $\gamma_*$, we have a contradiction. Hence, for all times, $g(\gamma_*, t) \le 2\eta$, implying the result. □

## B    Remark on weight initialization

By adapting a well-known result (Ambrosio et al., 2006, Prop. 9.3.2), we can quantify how the choice of weight initialization affects the distribution of converged training weights.



**Proposition 4.** *Further assume that $\mathcal{L}$ is $\lambda$-convex. Given two different weight initializations $p_0^{(0)}$ and $p_0^{(1)}$ on $\mathcal{P}(\Omega)$ that evolves according to Eqn. 3, we have $W_2(p_t^{(0)}, p_t^{(1)}) \le e^{-\lambda t} W_2(p_0^{(0)}, p_0^{(1)})$.*



*Proof of Prop. 4.* If $\mathcal{L}$ is $\lambda$-convex, then by Ambrosio et al. (2006, Prop. 9.3.2), we have that $L^\dagger(p) := \int_\Omega \mathcal{L}(x) \, dp(x)$ is $\lambda$-geodesically convex. Now, we see that Ambrosio et al. (2006, Thm. 11.1.4) applies, and we get the desired inequality. □

The primary hurdle to applying Prop. 4 is that $\mathcal{L}$ is rarely convex in the network parameters. For instance, optimization of a multi-layer perceptron (MLP) is highly non-convex due to non-linear activations and the product between hidden and outer layer weights. Interestingly, given a loss minimization problem on a MLP, we have a corresponding convex optimization problem (Pilanci and Ergen, 2020), i.e. the loss objective is convex in the network parameters and the two problems have identical optimal values. Therefore, with a modified loss function $\mathcal{L}'$, if $p_1'$ is distributed over its minimizers, e.g. gradient descent is used to solve the convex minimization problem until convergence, we can apply Prop. 4 with $\lambda = 0$ and $\mathcal{L}'(\theta_1) = \mathcal{L}(\theta_1)$ is minimal for $\theta_1 \sim p_1'$.

## C    Related works

**Conditional flow matching.** The CFM objective, where a conditional vector field is regressed to learn probability paths from a source to target distribution, was first introduced in Lipman et al. (2023). The CFM objective attempts to minimize the expected squared loss of a target conditional vector field (which is conditioned on training data and generates a desired probability path) and an unconditional neural network.

The authors showed that optimizing the CFM objective is equivalent to optimizing the unconditional FM objective. Moreover, the further work (Tong et al., 2024) highlighted that certain choices of parameters for the probability paths led to the optimal conditional flow being equivalent to the optimal transport path between the initial and target data distributions, thus resulting in shorter inference times. However, the original formulations of flow matching assumed that the initial distributions were Gaussian. Pooladian et al. (2023) extended the theory to arbitrary source distributions using minibatch sampling and proved a bound on the variance of the gradient of the objective. Tong et al. (2024) showed that using the 2-Wasserstein optimal transport map as the joint probability distribution of the initial and target data along with straight conditional probability paths results in a marginal vector field that solves the dynamical optimal transport problem between the initial and target distributions.

**Neural network parameter generation.** Due to the flexibility of neural network as function approximators, it is natural to think that they could be applied to neural network weights. Denil et al. (2014) paved the way for this exploration as their work provided evidence of the redundancy of most network parameterizations, hence showing that paramter generation is a feasible objective. Later, Ha et al. (2017) introduced Hypernetworks which use embeddings of weights of neural network layers to generate new weights and apply their approach to dynamic weight generation of RNNs and LSTMs. A significant portion of our paper's unconditional parameter generation section builds upon the ideas from Peebles et al. (2022); Wang et al. (2024) and the concurrent work of Soro et al. (2025) where the authors employ a latent diffusion model to generate new parameters for trained image classification networks. More direct auto-encoding methods have also seen success, for example Schürholt et al. (2024) and Wang et al. (2025).

**Weight generation for few-shot learning.** Weight space generation is commonly employed as a meta-learning algorithm. A prominent example in literature is for the task of few-shot learning. An early example is Ravi and Larochelle (2017) who designed a meta-learner based on the computations in an LSTM cell. Moreover, we may leverage the advancements in generative modeling for weight generation. Lee et al. (2023) used transformers for in-context reinforcement learning, but we also see the works of Zhmoginov et al. (2022); Hu et al. (2022); Kirsch et al. (2024); Fifty et al. (2024) use transformers and foundation models. More similar to our method is the body of work on using diffusion models for weight generation (Du et al., 2023; Zhang et al., 2024; Wang et al., 2024; Soro et al., 2025). These methods vary in their approach, some leveraging a relationship between the gradient descent algorithm and the denoising step in diffusion models to design their meta-learning algorithm. Others rely on the modeling capabilities of conditioned latent diffusion models to learn the target distribution of weights. Most evaluations conducted were in-distribution tasks, i.e. tasks sampled from the same data distribution as the training tasks, hence, there is room to explore adaptation for out-of-distribution tasks.

## D   Stochastic formulation of weight evolution

### D.1   Setup

The present formulation of gradient descent as gradient flow ignores the crucial role of noise within typical neural network optimization schemes. Stochastic differential equations (SDEs) provides a way to model SGD as a continuous-time process while taking into account the role of noise. Following Li et al. (2021), we write:

$$dX_t = -\nabla\mathcal{L}(X_t)dt + (\alpha\Sigma(X_t))^{1/2}dW_t \tag{26}$$

where $W_t$ is the Wiener process, $\alpha > 0$ is the learning rate, and $\Sigma(X) = \mathbb{E}[(\nabla\mathcal{L}_\xi(X) - \nabla\mathcal{L}(X))(\nabla\mathcal{L}_\xi(X) - \nabla\mathcal{L}(X))^\top]$; here, $\xi$ is a random variable denoting the random batch of training data in the context of SGD. We may then write the Fokker-Planck-Kolmogorov (FPK) equation

$$\partial_t p_t - \nabla \cdot (p_t \nabla\mathcal{L}) = \frac{\alpha}{2} \sum_{ij} \frac{\partial^2}{\partial x_i \partial x_j}([\Sigma^{1/2}\Sigma^{\top/2}]_{ij} p_t),$$

and the corresponding Schrödinger bridge (SB) problem

$$\mathbb{P}^* := \underset{\mathbb{P}_0=p_0, \mathbb{P}_1=p_1}{\arg\min} D_{KL}(\mathbb{P}|\mathbb{Q}) \tag{27}$$

where $\mathbb{Q} = \text{Law}(X)$ as governed by Eqn. 26.

## D.2 Solution with known SB matching methods

We run into the issue of well-posedness if the noise covariance is not known a priori, and also if it depends on the state $X_t$. For now, we are aware of the work by Berlinghieri et al. (2025), which only assumes standard SDE regularity conditions to optimize Eqn. 27. In particular, given a density over time $h(\cdot)$ and samples from a random time $t_i$, we may construct a state distribution at snapshots $\hat{f}_{t_i}(\cdot)$. For our approximation, we instead use an empirical time density $\hat{h}(\cdot)$ approximated from sampled timesteps, and a candidate SDE model parameterized by $\theta$, $f_{\theta,t}$. Training proceeds by using the maximum mean discrepancy (MMD) to quantify the discrepancy between $\hat{h} \circ f_{\theta,t}$ and $\hat{h} \circ \hat{f}_t$.

To simplify the analysis and align with most SB approximation methods, take $\epsilon_t = \alpha \mathbb{E}_{X_t}[\Sigma(X_t)]$ to form the approximation

$$dY_t = -\nabla \mathcal{L}(Y_t)dt + \sqrt{\epsilon_t}dW_t, \tag{28}$$

and obtain the FPK

$$\partial_t p_t - \nabla \cdot (p_t \nabla \mathcal{L}) = \frac{\epsilon_t}{2}\Delta p_t. \tag{29}$$

In this form, multi-marginal methods (Lavenant et al., 2024; Chen et al., 2023; Shen et al., 2025) may be employed with intermediate weight samples as reference data.

**Variational interpolants.** The recent work by Shen et al. (2025) sheds some light directly onto the problem of modeling gradient flows. This method allows us to specify a family of possible proxies e.g. those induced by the SDE

$$dX_t = \nabla \Psi_\alpha(X_t)dt + \gamma_\beta dW_t, \tag{30}$$

where $\alpha, \beta$ are learnable parameters. This method is explored in the context of multi-marginal Schrödinger bridges, but it is trivial to modify it for our purposes:

1. For $i = 0, \ldots, K-1$, consider data anchors $\{\theta_{t_i}^j\}_{j \in [N_i]}$ and $\{\theta_{t_{i+1}}^j\}_{j \in [N_{i+1}]}$.

   (a) Simulate forward to $t_{i+1}$ from $\{\theta_{t_i}^j\}_{j \in [N_i]}$ using Eqn. 30.
   (b) Simulate backward to $t_i$ from $\{\theta_{t_{i+1}}^j\}_{j \in [N_{i+1}]}$ using Eqn. 30.
   (c) Use simulated samples to estimate the drift between $t_i$ and $t_{i+1}$.

2. Concatenate the estimated drifts and use Stage 2 of Alg. 1 in Shen et al. (2025) to fit $\alpha$ according to the estimated drifts.

The diffusion parameter $\beta$ can be estimated in an outer loop of the above algorithm, as suggested by Shen et al. (2025), allowing us to match $\epsilon_t$. By following this procedure for flow model training, we can vary our interpolant within a natural family of path distributions, using data anchors to better inform training.

**Generalized Schrödinger Bridge Matching.** Due to the close relation with our analysis in Sec. 3.2, we further discuss GSBM (Liu et al., 2024c) which employs entropic action matching as the inner loop objective. In particular, given the reference process Eqn. 29, invoke Neklyudov et al. (2023, Prop. 3.1) to obtain a unique entropic action satisfying the FPK equation Eqn. 29. However, in this case, we have a two level optimization: **(1)** perform action matching to obtain the drift $u_t^\theta$ for a fixed reference path $(\hat{p}_t)_{t \in [0,1]}$, and **(2)** optimize the marginals $\hat{p}_t$ given the coupling $p_{0,1}^\theta$ evolved according to the learned drift $u_t^\theta$ in

$$dX_t = u_t^\theta(X_t)dt + \sqrt{\epsilon_t}dW_t. \tag{31}$$

Most relevant to us is the second stage which involves optimizing the marginal distributions. Given $x_0 \sim p_0$ and $x_1$ from Eqn. 31, we also obtain the intermediate states $\{X_{t_k}\}$ for $0 < t_1 < \cdots < t_K < 1$. To parameterize the marginals $p_t$, we assume a Gaussian path $p_t(X_t|x_0, x_1) = \mathcal{N}(\mu_t, \sigma_t^2 \boldsymbol{I})$, hence deferring our optimization to $\mu_t$ and $\sigma_t$. Liu et al. (2024c) uses 1-D splines with the control points $\{X_{t_k}\}$ to obtain $\mu_t$ and a uniform

sampling of $\sigma_t$ (with boundary conditions $\sigma_0 = \sigma_1 = 0$). Using this parameterization, we can compute the minimization objective

$$\mathcal{J} = \int_0^1 \mathbb{E}_{p_t(X_t|x_0, x_1)} \left[ \frac{1}{2} ||u_t^\theta(X_t)||_2^2 + V_t(X_t) \right] dt \tag{32}$$

to optimize the control points $X_{t_k}$ and $\sigma_{t_k}$, given some choice of $V_t(\cdot)$.

Returning to the marginals $p_t$, we wish to relate its evolution with Eqn. 28. Applying, for example, Du et al. (2024, Prop. 3), we can write down the FPK equation

$$\partial_t \hat{p}_t(x) = -\nabla \cdot (\hat{p}_t(x) u_t(x)) + \frac{\epsilon_t}{2} \Delta \hat{p}_t \quad \text{where} \quad u_t(x) = \dot{\mu}_t + \frac{1}{2} \left( \frac{\dot{\sigma}_t}{\sigma_t} - \frac{\epsilon_t}{\sigma_t} \right)(x - \mu_t). \tag{33}$$

Therefore, our analogy to Eqn. 23 would be to choose our energy functional $V$ to minimize the gap

$$\int_0^1 \mathbb{E}_{x \sim \hat{p}_t} ||\nabla \mathcal{L}(x) + u_t(x)||_2^2 \, dt.$$

# E  Interpolating paths

There are many variants of interpolating paths that have been used as reference in the flow matching literature. Typically, the conditional probability path is of the form

$$p_t(x|z) = \mathcal{N}(x; \mu_t(z), \sigma_t^2(z)\boldsymbol{I}) \tag{34}$$

with the consideration that the boundary conditions are satisfied, i.e. $\int p_0(x|z) \, dq(z) = p_0(x)$ and $\int p_1(x|z) \, dq(z) = p_1(x)$. The simplest case by Tong et al. (2024) considers a linear interpolant $\mu_t(x_0, x_1) = tx_1 + (1-t)x_0$ with a constant, small variance and $q(x_0, x_1) = p_0(x_0)p_1(x_1)$.

Since we are most concerned with the inducing vector field (see Eqn. 1), we would like a simple vector field that induce the desired conditional probability path. The Gaussian path Eqn. 34 is known (Rohbeck et al., 2025) to have flow

$$\phi_t(x|z) = \mu_t(z) + \sigma_t(z) \left( \frac{x - \mu_0(z)}{\sigma_0(z)} \right)$$

which, in fact, has a unique inducing vector field (Lipman et al., 2023)

$$u_t(x|z) = \frac{\sigma_t'(z)}{\sigma_t(z)}(x_t - \mu_t(z)) + \mu_t'(z). \tag{35}$$

The above suggests that if we have a desired interpolating path $\mu_t$, the vector field to match is known from Eqn. 35. Following, we discuss a few learned variants such as JKOnet (Bunne et al., 2022; Terpin et al., 2024), gWOT (Lavenant et al., 2024), and a varying proxy by Shen et al. (2025). We close by discussing the simple interpolants we use in our experiments.

**JKOnet.**  Deceptively, we first discuss a learned method which does not produce interpolants per se, but instead an energy functional $J_\xi$ used in a JKO scheme. Looking at the proof of Theorem 1 in App. A, we see the relation of Eqn. 3 to a JKO flow. Indeed, $J_\xi$ attempts to approximate $\int_\Omega \mathcal{L} \, dp$ from samples of the weight population by a neural network (Bunne et al., 2022; Terpin et al., 2024). As this is a learned method, we incur some computational cost, but also we require a dense sampling of intermediate weights to construct the necessary population samples. In return, we could even utilize this network as the meta-learner directly, or use it as a reference for flow model training.

**Lifted curves.**  In spline interpolation, it is best if data points correspond to different timesteps to better capture the trajectory over time. If instead we have a sampling of population over time, it is more natural to consider matching marginal path distributions. Lavenant et al. (2024) (gWOT) provides a framework for

exactly this. When we have data from $N_i$ samples at various time points $t_i$, $\{\theta_{t_i}^j\}_{i\in[K]}$, we may form the empirical distribution

$$\hat{\rho}_{t_i} = \frac{1}{N_i} \sum_{j=1}^{N_i} \delta_{\theta_{t_i}^j}.$$

To produce smoother interpolants, we also introduce a regularizer and Gaussian convolution with a kernel of width $h$ to obtain $\hat{\rho}_{t_i}^h$. Specifically, we minimize (by gradient descent) the convex functional

$$F_{K,\lambda,h}(\mathbf{R}) := \sigma^2 D_{KL}(\mathbf{R}|\mathbf{W}^\sigma) + \frac{1}{\lambda} \sum_{i=1}^{K} |t_{i+1} - t_i| D_{KL}(\hat{\rho}_{t_i}^h | \mathbf{R}_{t_i})$$

over a law on paths $\mathbf{R} \in \mathcal{P}(\Omega)$.

**Data-dependent geodesics.** Kapusniak et al. (2024) made use of a data-dependent metric to compute a geodesic. In practice, the interpolant is obtained through training a neural network to minimize

$$L_{\mathrm{mfm}}(\theta) = \mathbb{E}_{t\sim U[0,1],(\mathbf{x}_0,\mathbf{x}_1)\sim\pi} ||v_\theta(\mathbf{x}_{t,\eta^*}, t) - \dot{\mathbf{x}}_{t,\eta^*}||_{g(\mathbf{x}_{t,\eta^*})}^2. \tag{36}$$

Comparing with Eqn. 35, note that we are using a small, constant variance, so indeed we only match the interpolant derivative. Moreover, as $g$ is a data-dependent metric, its optimization towards a suitable proxy path reduces to learning a parameterization of $g$ w.r.t. the intermediate weights $\mathcal{D}$. Here, we note the MFM framework fits without issue into the proxy path framework of Sec. 3.2 primarily because geodesics are presumed smooth in time. Intuitively, due to randomness in the training process, some intermediate weights $\theta \in \mathcal{D}$ ought to be weighed less than others so that the induced geodesic $\gamma$ better matches the true loss gradient.

**Cubic splines.** Following the recent work by Rohbeck et al. (2025), we can fit a cubic spline to conditional data points and use this curve as the reference interpolant. Cubic splines are obtained by optimizing the variational objective

$$\mu_t(x_0, \ldots, x_k) = \arg \min_{\gamma \in \mathcal{H}^2([t_0, t_K])} \int_{t_0}^{t_K} ||\ddot{\gamma}(t)||_2^2 \, dt \tag{37}$$

where $x_k = \gamma(t_k)$ and $\mathcal{H}^2([t_0, t_K])$ denotes the class of functions that has absolutely continuous first derivative and weak second derivative on the interval $[t_0, t_K]$. The conditioning data may be sampled, say from our dataset $\mathcal{D}$. Moreover, Rohbeck et al. (2025) considered class-conditional trajectories. In our setting, this could mean weight trajectories from different training datasets. Thus, we sample intermediate weights $(x_0^c, \ldots, x_K^c)$, where $c$ denotes the training set, and we may use techniques such as classifier-free guidance on $v_\theta$ to improve training. Note that as the interpolant is entirely contingent on the intermediate distributions, and specifically the intermediate samples, interpolant optimization cannot be done within this approach. Instead, we rely on a faithful sampling $\mathcal{D}$ from the reference Eqn. 3 to provide a good proxy path.

**Piecewise linear interpolants.** We conclude by discussing the most straightforward method of incorporating conditional data. In particular, if we have points $z = (x_0, \ldots, x_K)$ at times $(t_0, \ldots, t_K)$, we have the conditional vector field

$$u_t(x|z) = \sum_{k=0}^{K-1} \frac{x_{k+1} - x_k}{t_{k+1} - t_k} \mathbf{1}_{[t_k, t_{k+1})}(t). \tag{38}$$

Within our conceptual framework, we may write

$$\frac{1}{2} \int_0^1 \mathbb{E}_\gamma ||\nabla\mathcal{L}(\gamma_t) + \dot{\gamma}_t||^2 \, dt = \frac{1}{2} \int_0^1 \sum_{k=0}^{K-1} \mathbf{1}_{[t_k, t_{k+1})}(t) \mathbb{E}_{(x_k, x_{k+1})\sim p_{t_k} \otimes p_{t_{k+1}}} ||\nabla\mathcal{L}(\gamma_t) + \frac{x_{k+1} - x_k}{t_{k+1} - t_k}||^2$$

as the action gap. As the interpolant is entirely contingent on the intermediate distributions, and specifically the intermediate samples, interpolant optimization cannot be done within this approach. However, this

approach has desirable limiting properties w.r.t. the sampling of $\mathcal{D}$. In particular, recalling Eqn. 3, if we let $\sigma : [0, 1] \times \mathbb{R}^p \to \mathbb{R}^p$ be the flow of the drift in Eqn. 3, the expression

$$\mathbb{E}_{(x_k, x_{k+1}) \sim p_{t_k} \otimes (\sigma_{t_{k+1} - t_k})_{\#} p_{tk}} ||\nabla \mathcal{L}(\gamma_t) + \frac{x_{k+1} - x_k}{t_{k+1} - t_k}||^2$$

goes to zero as $t_{k+1} - t_k \to 0$. In other words, if the intermediate samples are drawn from the same training trajectory, and are sampled with sufficient time-density, the gap can be made arbitrarily small.

## F  Implementation details

Here, we expound on the implementation of our approach. See Figure 8 for a schematic of the training and inference process.

### F.1  Pre-trained model acquisition

**Datasets and architectures.**  We conduct experiments on a wide range of datasets, including CIFAR-10/100 (Krizhevsky and Hinton, 2009), STL-10 (Coates et al., 2011), Fashion-MNIST (Xiao et al., 2017), CIFAR10.1 (Recht et al., 2019), and Camelyon17 (Veeling et al., 2018). To evaluate our meta-model's ability to generate new subsets of network parameters, we conduct experiments on ResNet-18 (He et al., 2015b), ViT-Base (Dosovitskiy et al., 2021), ConvNeXt-Tiny (Liu et al., 2022), the latter two are sourced from timm Wightman (2019). As we shall detail below, small CNN architectures from a model zoo (Schürholt et al., 2022) are also used for full-model generations.

**Model pre-training and checkpointing.**  For better control over the target distribution $p_1$, in experiments involving ResNet-18, ViT-Base, and ConvNeXt-Tiny, we pre-train these base models from scratch on their respective datasets. We follow Wang et al. (2024) and train the base models until their accuracy stabilizes (in practice, we train all base models for 100 epochs). Depending on the experiment, we save checkpoints differently. If only the converged weights are needed, we save 200 weights at every iteration past 100 epochs. Otherwise, if we require intermediate weights, then we specify the number of saving epochs and the number of weights to save in such an epoch. For instance, we may have 100 save epochs and 100 saves per epoch, meaning that we save at *every* training epoch and save weights in the first 100 iterations of each epoch.

**Model zoo.**  The model zoo used for meta-training in the model retrieval setting, was sourced from (Schürholt et al., 2022). As the base model, we employed their CNN-medium architecture, which consists of three convolutional layers and contains 10,000-11,000 parameters, depending on the number of input channels. We pre-trained these models as described above.

### F.2  Variational autoencoder

The variational autoencoder follows the implementation of Soro et al. (2025), which employs a UNet architecture for auto-encoding. In particular, given a set of model weights $\{\mathcal{M}_i\}_{i=1}^N$, we first flatten the weights to obtain vectors $\boldsymbol{w}_i \in \mathbb{R}^{d_i}$. For the sake of uniformity, we always zero-pad vectors to $d = \max_i d_i$. Alternatively, we allow for layer-wise vectorization: set a chunk size $\ell$ which corresponds to the weight dimension of a network layer. Then, zero-pad $\boldsymbol{w}_i$ to be a multiple of $\ell$, say $\tilde{d}$. This allows us to partition into $k$ equal length vectors $\boldsymbol{w}_{i,k} \in \mathbb{R}^{\tilde{d}/k}$. Typically, larger models benefit from layer-wise vectorization.

Subsequently, we train a VAE to obtain an embedding of such vectors by optimizing the objective:

$$L_{\text{VAE}}(\theta, \phi) := -\mathbb{E}_{\boldsymbol{z} \sim q(\boldsymbol{z}|\boldsymbol{w})}[\log p_\theta(\boldsymbol{w}|\boldsymbol{z}) + \beta D_{KL}(q_\phi(\boldsymbol{z}|\boldsymbol{w})||p(\boldsymbol{z}))] \tag{39}$$

where $\boldsymbol{w}$ is the vectorized weights, $\boldsymbol{z}$ is the embedding we are learning, and $p_\theta, q_\phi$ are the reconstruction and posterior distributions respectively. Moreover, we fix the prior $p(\boldsymbol{z})$ to be a $(0, 1)$-Gaussian and the weight is set to be $\beta = 10^{-5}$. For layer-wise vectorization, we simply change the input dimensions to match the chunk size. Upon decoding, we concatenate the chunks to re-form the weight vector.

As for training, the VAE was trained with the objective in equation 39. Moreover, following p-diff (Wang et al., 2024), we add Gaussian noise to the input and latent vector, i.e. given noise factors $\sigma_{in}$ and $\sigma_{lat}$ with encoder $f_\phi$ and decoder $f_\theta$, we have

$$\boldsymbol{z} = f_\phi(\boldsymbol{w} + \xi_{in}), \ \hat{\boldsymbol{w}} = f_\theta(\boldsymbol{z} + \xi_{lat}) \quad \text{where} \quad \xi_{in} \sim \mathcal{N}(0, \sigma_{in}^2 \boldsymbol{I}), \ \xi_{lat} \sim \mathcal{N}(0, \sigma_{lat}^2 \boldsymbol{I}).$$

A new VAE is trained at every instantiation of the N$\mathcal{M}$-CFM model as architectures often differ in their input dimension for different experiments. However, they are trained with different objectives: the VAE is trained to minimize reconstruction loss. In all experiments, we fix $\sigma_{in} = 0.001$ and $\sigma_{lat} = 0.01$.

---

**Algorithm 1** Sampling Trajectories from Weight Tensor

---

**Require:** Number of save epochs $N_{epochs}$, savepoints per epoch $S$, classifier size $D$, tensor of classifier weights
    $X \in \mathbb{R}^{N_{epochs} \times S \times D}$, number of time samples $K \leq N_{epochs}$.
**Ensure:** Sampled tensor $W \in \mathbb{R}^{K \times S \times D}$
1: Flatten $X$ to shape $[N_{epochs} \cdot S, D]$                 ▷ Assumes first dim. sorted by training iteration.
2: Sample $K \cdot S$ indices $I$ uniformly from $[0, N_{epochs}S)$
3: Extract $X[I]$ and reshape to $W \in \mathbb{R}^{K \times S \times D}$
4: **if** add_noise **then**
5:     $W \leftarrow W + \epsilon$                                            ▷ $\epsilon \sim \mathcal{N}(0, 10^{-3})$
6: **end if**
7: **return** $W$

---

### F.3 Generative meta-model

**Multi-marginal flow matching.** Multi-marginal flow matching (MMFM) proceeds in the same regression paradigm as flow matching models, with the difference being the regression target is now Eqn. 38 (piece-wise linear interpolation). In practice, $z = (x_0, \ldots, x_K)$, where we have $K = 3, 4$, or $5$, as evaluated in Table 1, and the elements of $z$ are sampled from $p_{t_0} \otimes \cdots \otimes p_{t_K}$ constituted by samples obtained from base model pre-training App. F.1. Typically, we save a lot more checkpoints than needed, and so we need to subsample them by Algorithm 1 to create the training dataset. For better training, we may also choose to sample weight initializations, i.e. $x_0 \in z$, for each batch. This can be done easily by using `torch.nn.init` to reset parameters of a module to the desired initialization, e.g. Kaiming uniform. In fact, sampling the weight initialization is all that is required for the validation step.

**JKOnet$^*$.** The dataloading aspect of JKOnet training is exactly the same as MMFM, so we focus our attention to the training regiment. Following Terpin et al. (2024), we may formally write our loss as

$$\sum_{k=0}^{K-1} \int_{\mathbb{R}^D \times \mathbb{R}^D} ||\nabla V_\theta(x_t, t) + (x_{t+1} - x_t)/\Delta t||^2 \, d\gamma(x_t, x_{t+1}). \tag{40}$$

Recalling the gradient descent formula, the time argument is not necessary for this loss, but we found it to improve empirical performance. Moreover, the $\gamma(\cdot, \cdot\cdot)$ distribution is traditionally an optimal coupling of consecutive marginals $p_{t_k}, p_{t_{k+1}}$, however, we have a more natural choice: $x_t$ and $x_{t+1}$ ought to be checkpoints saved consecutively during pre-training. Indeed, we found this to be the superior choice in terms of performance. Moreover, an important note is in order: since JKOnet$^*$ expects $x_t$ to evolve by a gradient flow, this imposes a condition on how the checkpoints are obtained. We found that meta-training **only works if pre-training is done using the SGD optimizer**. Modern optimizers such as Adam notably fails for JKOnet$^*$. We also connect this point to the footnote in Table 2: just because the weights evolve by a gradient flow does *not* mean that the encoded weights behave the same way. Indeed, model retrieval in latent space fails when we use JKOnet$^*$. This suggests an avenue for further research: construct an autoencoder that preserves gradient flow in latent space, where preferably the latent gradient flow can be deduced/estimated from the original gradient steps.

Table 6: Model architectures and hyperparameters. The JKOnet model uses the same UNet as N$\mathcal{M}$-CFM and N$\mathcal{M}$-MMFM, but with the up-sampling section replaced by pooling and linear layers.

| **Weight Encoder** (UNet) | |
| --- | --- |
| Architecture | VAE |
| Latent Space Size | $4 \times 4 \times 4$ |
| Upsampling/Downsampling Layers | 5 |
| Channel Multiplication (per Downsampling Layer) | (1, 1, 2, 2, 2) |
| ResNet Blocks (per Layer) | 2 |
| KL-Divergence Weight | 1e-5 |
| **N$\mathcal{M}$-CFM and N$\mathcal{M}$-MMFM Model** (UNet) | |
| Input Size w/ VAE | $4 \times 4 \times 4$ |
| Input Size w/o VAE | variable |
| Upsampling/Downsampling Layers | 4 |
| Channel Multiplication (per Downsampling Layer) | (2, 2, 2, 2) |
| ResNet Blocks (per Layer) | 2 |

**Architecture.** The neural network used for flow matching is the UNet from D2NWG (Soro et al., 2025). The specific hyperparameters used for the CFM model varies between experiments, so we leave this discussion to the next section. For experiments such as model retrieval where we require a conditioning vector, this is implemented by concatenating a context feature vector (e.g. images are passed into a CLIP encoder (Radford et al., 2021), and optionally an attention module if we have a set of context images) to the last dimension (same axis as the neural network parameters); of course, we ignore these extra features after the forward pass.

### F.4 Reward fine-tuning

**Adjoint matching with $\sigma_t = 10^{-3}$.** We start in Domingo-Enrich et al. (2025, App. C.5). As we are using N$\mathcal{M}$-CFM, let $Y_0 \sim p_0$ and $Y_1 \sim p_1$, then we may write

$$
\begin{aligned}
v(x, t) &= \mathbb{E}[\alpha_t Y_1 + \beta_t Y_0 \mid x = \alpha_t Y_1 + \beta_t Y_0] \\
&= \mathbb{E}\big[\frac{\dot{\alpha}_t (x - \beta_t Y_0)}{\alpha_t} + \dot{\beta}_t Y_0 \mid x = \alpha_t Y_1 + \beta_t Y_0\big] \\
&= \frac{\dot{\alpha}_t}{\alpha_t} x + (\dot{\beta}_t - \frac{\dot{\alpha}_t}{\alpha_t}\beta_t)\mathbb{E}[Y_0 \mid x = \alpha_t Y_1 + \beta_t Y_0],
\end{aligned}
\tag{41}
$$

using the fact that $Y_1 = (x - \beta_t Y_0)/\alpha_t$. On a practical note, we typically have $\alpha_t = t$, $\beta_t = 1 - t$. A central piece of the theory consists of relating this vector field $v$ with the score function $s(x, t)$. We note that once this relationship is established with $\sigma_t$, then the rest of the adjoint matching derivation follows. The crucial step is in writing the score function; following from Domingo-Enrich et al. (2025, Eqn. 92),

$$
s(x, t) = \frac{\mathbb{E}[p_{t|1}(x|Y_1)\nabla \log p_{t|1}(x|Y_1)]}{p_t(x)}, \quad \text{where} \quad p_{t|1}(x|Y_1) = \frac{\exp(-||x - \alpha_t Y_1||^2/2\beta_t^2)}{(2\pi\beta_t^2)^{D/2}}.
\tag{42}
$$

It suffices to change $p_{t|1}$:

$$
p_{t|1}(x|Y_1) = \frac{\exp(-||x - \alpha_t Y_1||^2/(2\sigma_t)^2)}{(2\pi\sigma_t^2)^{D/2}} \implies \nabla \log p_{t|1}(x|Y_1) = -\frac{x - \alpha_t Y_1}{\sigma_t^2}
\tag{43}
$$

and combine with Eqn. 42 to obtain $s(x, t) = -\beta_t \mathbb{E}[Y_0 \mid x = \beta_t Y_0 + \alpha_t Y_1]/\sigma_t^2$. Further combining with Eqn. 41, we have the correspondence

$$
v(x, t) = \frac{\dot{\alpha}_t}{\alpha_t} x + \left(\frac{\dot{\alpha}_t}{\alpha_t}\beta_t - \dot{\beta}_t\right)\frac{\sigma_t^2}{\beta_t} s(x, t) \iff s(x, t) = \frac{\beta_t \left(v(x, t) - \frac{\dot{\alpha}_t}{\alpha_t}x\right)}{\sigma_t^2 \left(\frac{\dot{\alpha}_t}{\alpha_t}\beta_t - \dot{\beta}_t\right)}.
\tag{44}
$$

Concluding, we may thus write the SDE that mirrors Domingo-Enrich et al. (2025, Eqn. 6):

$$
\begin{aligned}
dX_t &= b(X_t, t)\, dt + \epsilon(t)\, dB_t \\
&= \left[\frac{\dot{\alpha}_t}{\alpha_t}x + s(x, t)\left(\left(\frac{\dot{\alpha}_t}{\alpha_t}\beta_t - \dot{\beta}_t\right)\frac{\sigma_t^2}{\beta_t} + \epsilon(t)^2/2\right)\right] dt + \epsilon(t)\, dB_t.
\end{aligned}
\tag{45}
$$

---

**Algorithm 2** Deterministic adjoint matching for fine-tuning flow models

---

**Require:** Pre-trained FM velocity field $v^{base}$, step size $h$, number of fine-tuning iterations $N$, trajectory batch size $M$, dataset batch size $m$, initialized $v^{ft} = v^{base}$, cross-entropy loss $\mathcal{L}$.
**Ensure:** Reward fine-tuned FM velocity field $v^{ft}$.

1: **for** $n \in [0, \ldots, N-1]$ **do**
2:     Sample $M$ trajectories $\boldsymbol{X} = (X_t)_{0 \leq t \leq 1}$ with an Euler solver with step size $h$ and $X_0 \sim p_0$.
3:     $\{(x_i, y_i)\}_{i=1}^m \sim \mathcal{D}$                                                   ▷ Sample from classifier dataset
4:     For each of the $M$ trajectories, evaluate classifier on predicted weights
5:            $\ell(X_1) = \sum_{i=1}^m \mathcal{L}(\text{NNET}_{X_1}(x_i), y_i)$.                    ▷ $\ell(X_1)$ is a vector of size $M$
6:     $\tilde{a}_{t-h} = \tilde{a}_t + h\tilde{a}_t^\top \nabla_{X_t} v^{base}(X_t, t), \quad \tilde{a}_1 = \nabla_{X_1} \ell(X_1)$      ▷ Backward solve the lean adjoint ODE
7:     Detach from computation graphs: $X_t = \texttt{stopgrad}(X_t)$ and $\tilde{a}_t = \texttt{stopgrad}(\tilde{a}_t)$.
8:     $\mathcal{L}_{AM}(\theta) = \sum_{t \in \{0, \ldots, 1-h\}} ||v_\theta^{ft}(X_t, t) - (v^{base}(X_t, t) - \tilde{a}_t)||^2$.
9:     Compute $\nabla_\theta \mathcal{L}_{AM}$ and optimize as usual.
10: **end for**

---

**Deterministic in practice.** The key quantity in adjoint matching is the memory-less schedule given as $\epsilon(t) = \sqrt{2\eta_t}$, where reading from Eqn. 45, $\eta_t = \left(\frac{\dot{\alpha}_t}{\alpha_t}\beta_t - \dot{\beta}_t\right)\frac{\sigma_t^2}{\beta_t}$. Plugging in $\alpha_t = t$, $\beta_t = 1-t$, we have

$$\epsilon(t) = \sqrt{\frac{2\sigma_t^2}{1-t}((1-t)/t + 1)} = \sqrt{\frac{2\sigma_t^2}{t(1-t)}} = \sigma_t\sqrt{\frac{2}{t(1-t)}}.$$

Following the suggestions in Domingo-Enrich et al. (2025, App. H), we add a small value $h$ to both terms in the denominator. In practice, this means that $\sqrt{\frac{2}{t(1-t)}} \leq 10$, whereas $\sigma_t = 10^{-3}$. Looking at the adjoint matching algorithm, this clearly presents a problem of scale. We found in practice that a deterministic variant, i.e. $\epsilon(t) = 0$ works well in our evaluations. To formulate this modified algorithm, note that our primary objective is to optimize over the control $u$ in $dX_t = (b(X_t, t) + \epsilon(t)u(X_t, t))\, dt + \epsilon(t)\, dB_t$. The original formulation includes the correction term to the drift, and defines $v^{ft}(x, t)$ as

$$b(x, t) + \epsilon(t)u(x, t) = 2v^{ft}(x, t) - \frac{\dot{\alpha}_t}{\alpha_t}x.$$

We simplify by instead defining $b(x, t) = v^{base}(x, t)$ and $v^{ft}(x, t) = b(x, t) + u(x, t)$. Consequently, this simplifies the adjoint ODE and the regression target, leading to Algorithm 2.

**Multi-marginal adjoint matching.** Since this paper touches upon multi-marginal flow matching in its experiments, we think it natural to extend the adjoint matching computations to the multi-marginal setting. In this section, we provide the derivations and leave experimentation to future work. Suppose we are given $z = (Y_0, \ldots, Y_K)$ at times $(t_0, \ldots, t_K) \subset [0, 1]$, we first need to find an expression for a sampled point $x$ analogous to $x = \beta_t Y_0 + \alpha_t Y_1$. Motivated by Eqn. 38, define

$$x = \sum_{k=0}^{K-1} [(1 - s_t^{(k)})Y_k + s_t^{(k)}Y_{k+1}]\mathbf{1}_{[t_k, t_{k+1})}(t), \quad \text{where } s_t^{(k)} = \frac{t - t_k}{t_{k+1} - t_k}. \tag{46}$$

Differentiating w.r.t. $t$, we indeed get $u_t(x|z)$ in Eqn. 38; moreover, to simplify notatoin, set $r_t^{(k)} = 1 - s_t^{(k)}$. The key insight is that only one of the terms in Eqn. 46 is non-zero for any $t \in [0, 1]$, therefore

$$
\begin{aligned}
v(x, t) &= \sum_k \mathbf{1}_{[t_k, t_{k+1})}(t)\mathbb{E}[\dot{r}_t^{(k)}Y_k + \dot{s}_t^{(k)}Y_{k+1} \mid x] \\
&= \sum_k \mathbf{1}_{[t_k, t_{k+1})}(t)\mathbb{E}\left[\dot{r}_t^{(k)}Y_k + \dot{s}_t^{(k)}\frac{x - r_t^{(k)}Y_k}{s_t^{(k)}} \mid x\right] \\
&= \sum_k \mathbf{1}_{[t_k, t_{k+1})}(t)\left[\frac{\dot{s}_t^{(k)}}{s_t^{(k)}}x + \left(\dot{r}_t^{(k)} - \frac{\dot{s}_t^{(k)}}{s_t^{(k)}}r_t^{(k)}\right)\mathbb{E}[Y_k \mid x]\right].
\end{aligned}
\tag{47}
$$

Like before, the key step is to relate the velocity $v$ and the score function. Note that the backward conditional probabilty is now

$$p_{t|1}(x|z) = \sum_k \mathbf{1}_{[t_k, t_{k+1})}(t) p_{t|t+1}(x|Y_{k+1}),$$

(48)

where again we use $\sigma_t$:

$$p_{t|t+1}(x|Y_{k+1}) = \frac{\exp(-||x - s_t^{(k)}Y_{k+1}||^2/2\sigma_t^2)}{(2\pi\sigma_t^2)^{D/2}} \implies \nabla \log p_{t|t+1}(x|Y_{k+1}) = -\frac{x - s_t^{(k)}Y_{k+1}}{\sigma_t^2}.$$

(49)

Since $\mathbb{E}, \nabla$ are linear, we may move around the sum as we please. Therefore, analagous to the CFM case:

$$s(x,t) = -\sum_k \mathbf{1}_{[t_k, t_{k+1})}(t)\mathbb{E}\left[\frac{x - s_t^{(k)}Y_{k+1}}{\sigma_t^2}\bigg| x\right] = -\sum_k \mathbf{1}_{[t_k, t_{k+1})}(t)\frac{r_t^{(k)}}{\sigma_t^2}\mathbb{E}[Y_k \mid x].$$

(50)

At this point, it is a matter of using the same trick as in Eqn. 47: there must be at most one $k \in [0, \ldots, K-1]$ such that $\mathbb{E}[Y_k \mid x] = -\sigma_t^2 s(x,t)/r_t^{(k)}$. Therefore,

$$v(x,t) = \sum_k \mathbf{1}_{[t_k, t_{k+1})}(t)\left[\frac{\dot{s}_t^{(k)}}{s_t^{(k)}}x + \frac{\sigma_t^2}{r_t^{(k)}}\left(\frac{\dot{s}_t^{(k)}}{s_t^{(k)}}r_t^{(k)} - r_t^{(k)}\right)s(x,t)\right]$$

$$\iff s(x,t) = \sum_k \mathbf{1}_{[t_k, t_{k+1})}(t)\left[\frac{r_t^{(k)}}{\sigma_t^2\left(\frac{\dot{s}_t^{(k)}}{s_t^{(k)}}r_t^{(k)} - r_t^{(k)}\right)}\left(v(x,t) - \frac{\dot{s}_t^{(k)}}{s_t^{(k)}}x\right)\right]$$

(51)

Now recall that

$$\frac{dX_t}{dt} = v(X_t, t) \iff dX_t = \left(v(X_t, t) + \frac{\epsilon(t)^2}{2}s(x,t)\right)dt + \epsilon(t)\,dB_t$$

as given in Domingo-Enrich et al. (2025, Eqns. 3, 4). Let $b(X_t, t)$ denote the drift term in the SDE above. Plugging in $v$ and $s$ from Eqn. 51, we find

$$\eta_t = \sum_{k=0}^{K-1} \mathbf{1}_{[t_k, t_{k+1})}(t)\frac{\sigma_t^2\left(\frac{\dot{s}_t^{(k)}}{s_t^{(k)}}r_t^{(k)} - r_t^{(k)}\right)}{r_t^{(k)}}.$$

(52)

Therefore, the memory-less noise schedule $\epsilon(t) = \sqrt{2\eta_t}$ yields

$$b(X_t, t) = \sum_{k=0}^{K-1} \mathbf{1}_{[t_k, t_{k+1})}(t)\left[2v(X_t, t) - \frac{\dot{s}_t^{(k)}}{s_t^{(k)}}X_t\right]$$

and by analogy we define the $v^{ft}$ via a control function $u$:

$$b(x,t) + \epsilon(t)u(x,t) = \sum_{k=0}^{K-1} \mathbf{1}_{[t_k, t_{k+1})}(t)\left[2v^{ft}(x,t) - \frac{\dot{s}_t^{(k)}}{s_t^{(k)}}x\right].$$

(53)

## F.5 Practical considerations and scaling

Thus far, our method has been demonstrated on small CNN models (on the order of 10,000 parameters), and a natural direction for future work is extending it to larger network architectures. For instance, the collection of training trajectories, detailed in App. F.1, requires about 2 hours of base model training (NVIDIA A40 GPU) and 1 GB of storage per downstream dataset (e.g. CIFAR10). The meta-model itself comes to about 4M parameters with slight deviations caused by differences in base model size (e.g. MNIST classifiers are slightly smaller than CIFAR10 classifiers); more generally, it scales linearly with the number of parameters of

the base model. The modular nature of our framework, however, provides multiple avenues for improving scalability. One promising path is leveraging the growing number of publicly available model zoos for larger architectures (Falk et al., 2025; Schürholt et al., 2025), which would reduce both training time, and possibly storage requirements if models can be streamed on demand. Another direction is the incorporation of encoding schemes such as VAEs, which have proven critical in scaling weight-space learning to architectures like ResNet-18 and Vision Transformers (Schürholt et al., 2024; Wang et al., 2025). Furthermore, when only weight embeddings are needed for training, storage costs can be further reduced by pre-computing and saving embeddings directly, rather than full model weights.

## G  Further experiments

In this section, we explore in depth the design choices that were made concerning the generative model. In particular, we bring to light some upsides of flow matching over prior diffusion methods (Soro et al., 2025), evaluate each approach's ability to model weight trajectories, and examine the effects of weight space symmetry (Hecht-Nielsen, 1990; Chen et al., 1993).

### G.1  Parameter symmetries

In this section, we detail two methods for incorporating parameter symmetries into the modeling framework. As mentioned in the main text, the symmetries we investigate involve permutation (Hecht-Nielsen, 1990) and scaling (Chen et al., 1993) of neurons in a neural network. We proceed with two approaches: weight alignment and equivariant architectures.

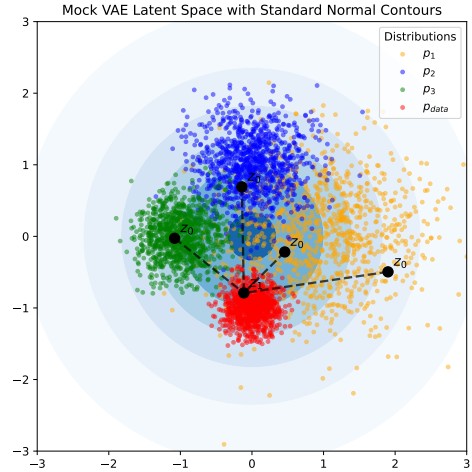

Figure 5: Mock visualization of VAE latent space in D2NWG. Although the interpolant (dotted line connecting $z_0$ to $z_1$) does not follow the reference trajectory $p_0 \rightarrow p_1 \rightarrow \cdots \rightarrow p_{data}$, the points on the line reside within the data manifold.

**Alignment to a reference.**   This approach involves choosing a reference base model and aligning the *permutation* of the layers of base models in the training and evaluation sets; we use Git Re-Basin (Ainsworth et al., 2023) for this task. We perform few-step inference and trajectory modeling with this setup, as shown in Table 7 and Figure 7.

**MonomialNFN as a meta-model.**   In this section, we substitute the UNet for the equivariant architecture MonomialNFN (Tran et al., 2024), which accounts for *both* permutation and scaling symmetries. The network design largely follows from the designs in Zhou et al. (2023); Tran et al. (2024). We start with a Gaussian Fourier Transformation with a mapping size of 16 and scale of 3. Then, we encode the features with a IOSinusoidalEncoding. The encoded features are passed into 4 MonomialNFN layers, employing a residual connection, of hidden dimension 32. Before every layer, we add a time-conditioned HyperNetwork (Sitzmann et al., 2020), similar to the time embedding layers of the UNet architecture. Finally, a last MonomialNFN layer is used to reduce the channel down to 1. We used a batch size of 32, and for optimization, we instantiate an AdamW optimizer with learning rate $10^{-3}$ and weight decay $10^{-2}$ for 1000 epochs. As we have explored few-step inference and trajectory modeling with layer alignment above, we focus on reward fine-tuning and evaluating the support of classifier weights as done in App. H.4, as shown in Table 8.

### G.2  Few-step inference

In this experiment, we wish to evaluate the inference capability of the generative meta-models under strained conditions, in this case, one-step and two-step inference of CNN3 base models. To further distinguish the flow matching framework, we also test N$\mathcal{M}$-CFM with a Gaussian source distribution (as opposed to Kaiming uniform). As comparison, we have the diffusion baseline from D2NWG (Soro et al., 2025).

Table 7: Mean validation accuracy over 10 runs of unconditional generation for CIFAR10 and MNIST using just one or two inference steps. We include results where the N$\mathcal{M}$-CFM source distribution is a standard Gaussian, as well as training runs where we aligned the layers to a reference model. Best mean accuracies are **bolded**, second-best underlined.

| | One-step | | Two-step | |
|---|---|---|---|---|
| | CIFAR10 | MNIST | CIFAR10 | MNIST |
| D2NWG | $51.36 \pm 4.64$ | $93.42 \pm 2.39$ | $50.54 \pm 4.75$ | $93.44 \pm 2.35$ |
| D2NWG-Aligned | $55.47 \pm 5.06$ | $95.94 \pm 0.80$ | $55.23 \pm 5.24$ | $95.96 \pm 0.81$ |
| N$\mathcal{M}$-CFM w/ Gauss | $26.72 \pm 5.81$ | $50.83 \pm 13.71$ | $47.46 \pm 10.99$ | $88.60 \pm 21.9$ |
| N$\mathcal{M}$-CFM-Aligned w/ Gauss | $25.63 \pm 4.04$ | $55.75 \pm 17.72$ | $49.15 \pm 5.18$ | $96.07 \pm 3.02$ |
| N$\mathcal{M}$-CFM | $49.75 \pm 3.07$ | $96.64 \pm 1.06$ | $62.98 \pm 0.47$ | $98.11 \pm 0.19$ |
| N$\mathcal{M}$-CFM-Aligned | $\mathbf{57.18 \pm 2.34}$ | $\mathbf{97.89 \pm 0.53}$ | $\mathbf{63.27 \pm 0.34}$ | $\mathbf{98.62 \pm 0.04}$ |

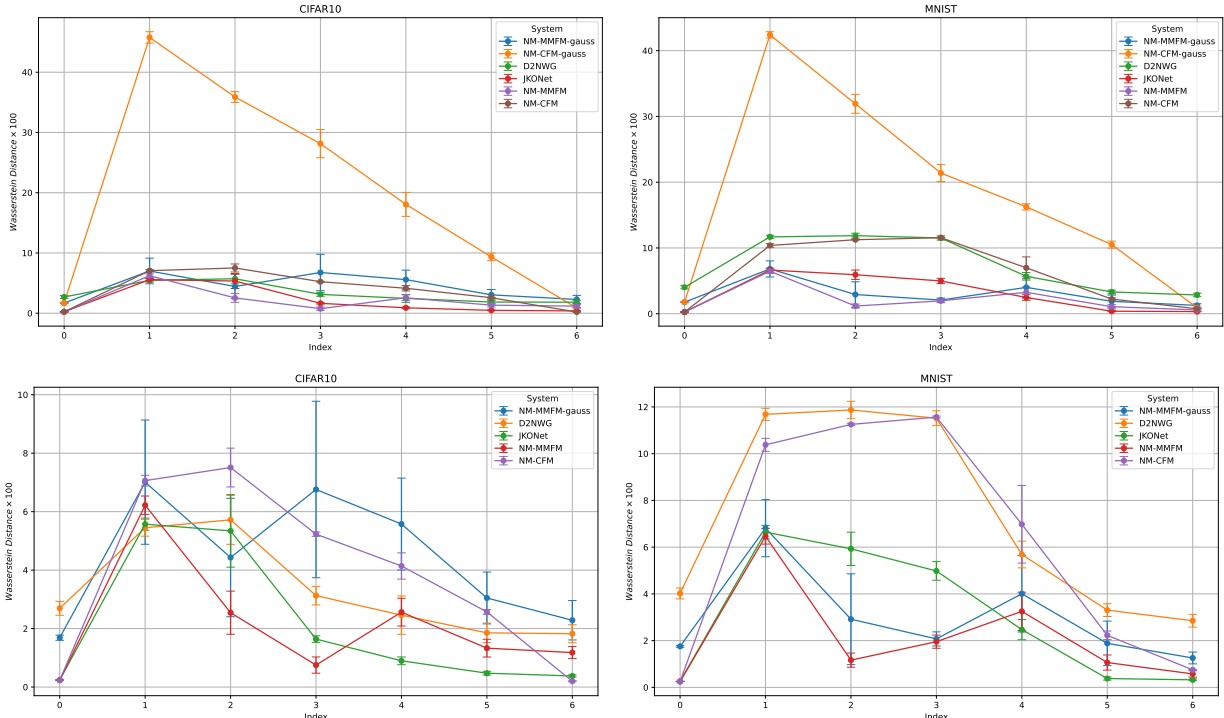

Figure 6: Mean Wasserstein-1 distance ($\times 100$) between reference and generated intermediate marginals over 5 seeds of unconditional generation. We also provide plots excluding N$\mathcal{M}$-CFM w/ Gauss due to its deviation from the rest.

Table 7 shows that N$\mathcal{M}$-CFM mostly outperforms D2NWG in the standard and aligned settings (see below for description). Moreover, we see a clear advantage when using Kaiming uniform (the weight initialization used during base model pre-training) over the standard Gaussian. Concerning parameter symmetries, Table 7 shows considerable improvements, especially on CIFAR10 tests, indicating that alignment helps in compute-constrained environments. The disparity between source distributions for the N$\mathcal{M}$-CFM approach can be explained by the distance between the Gaussian and trained weights distribution (see index 0 of the bottom row plots in Figure 6). In contrast, the VAE loss includes a KL divergence term which regularizes the latent space towards a standard Gaussian (Eqn. 39).

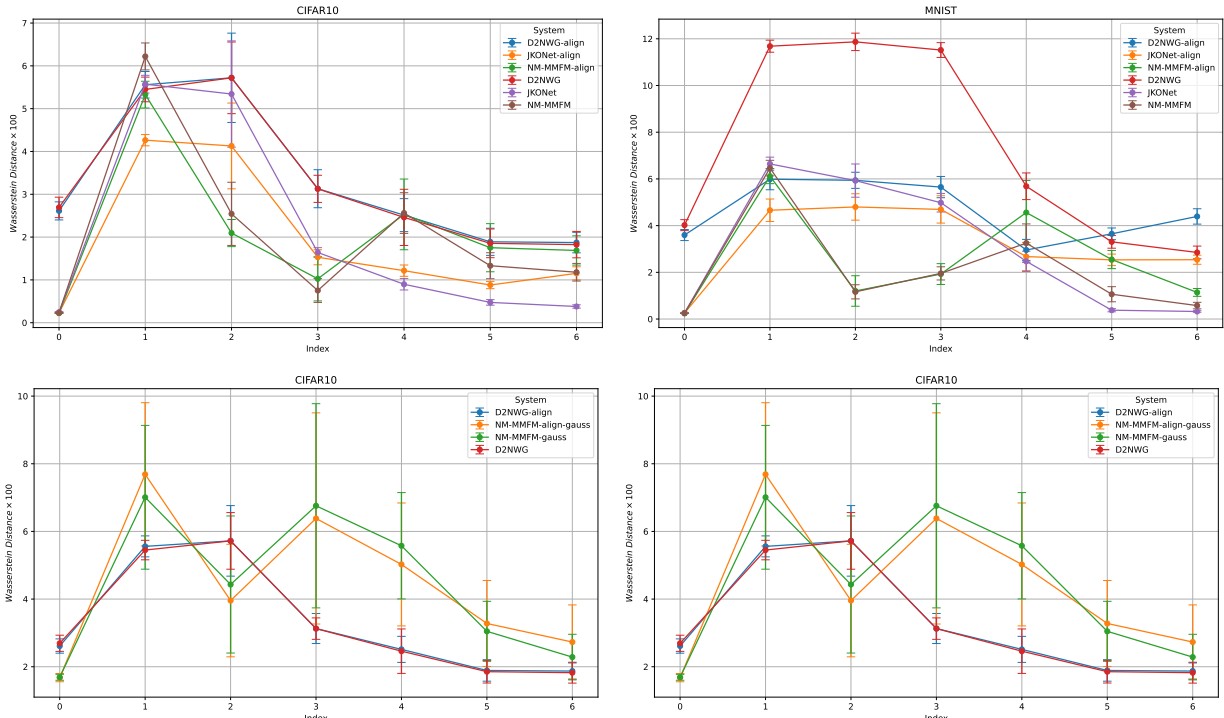

Figure 7: Mean Wasserstein-1 distance ($\times 100$) between reference and generated intermediate marginals over 5 seeds of unconditional generation, excluding N$\mathcal{M}$-CFM results, illustrating the effect of permutation alignment on the training and validation data. Due to the high $W_1$ distances resulting from N$\mathcal{M}$-CFM, we exclude them in the comparison.

## G.3  Trajectory modeling

In this experiment, we evaluate the ability of different approaches to model the weight trajectory. As decided in Section 5, we used N$\mathcal{M}$-MMFM(3) and N$\mathcal{M}$-JKO(4) as representatives for this method. Moreover, in the interest of fairness, we divide the trajectory into 5 buckets, and so the MMFM and JKO methods would need to interpolate between training distributions. Once again, our baseline is D2NWG where the VAE is trained on full trajectory weights at each batch iteration. See Figure 6 for results.

Interestingly, we find that the $W_1$-distance of the generated trajectory to be consistently lower in D2NWG vs. N$\mathcal{M}$-CFM, but both methods lag behind MMFM and JKO which explicitly models the weight trajectory. We suspect the D2NWG performance is because of the latent space which is trained to encode trajectory data. Thus, even if the interpolated weights do not follow the expected trajectory, it still lands on the data manifold; see Figure 5. Concerning parameter symmetries, Figure 7 shows a more modest improvement in the intermediate indices, e.g. indices 1-3, where D2NWG improves significantly, but happens to degrade in later indices, crucially where downstream evaluation takes place. We hypothesize that the improvement for D2NWG arises due to autoencoder underfitting. Indeed, the autoencoder only take parameters as input and is trained to reconstruct it, however, the N$\mathcal{M}$-CFM models take in an extra time parameter that helps distinguish between the different parameter distributions.

## G.4  Reward fine-tuning: support of classifier weights

We first note that for symmetry to hold, we had to remove the `MaxPool2d` layers that were previously included in the CNN architecture, hence the different validation accuracy. This can be seen as a downside of current equivariant architectures.

Intuitively, if we equate data in the same equivalence class, then the effective support of classifier weights ought to expand, thus improving generalization to prediction on corrupted data. This intuition turns out to be false, as seen in Table 8. It shows that the new architecture in fact degrades the validation accuracy, which is most evident in the Corruption Level 2 column. We hypothesize this is because the members of the equivalence class are separated by a considerable distance and they do not correspond to meaningful regions of weight space that influence generalization on corrupted datasets. Empirically, this reinforces the findings of a recent work (Zeng et al., 2025). Moreover, when using equivariant architectures, we forgo the basic Gaussian noise augmentation, which may be more beneficial for slight corruption in downstream task data.

Table 8: Mean validation accuracy over five generated classifiers after reward fine-tuning on increasingly corrupted datasets. The -MoNFN suffix indicates that network made use of the MonomialNFN (Tran et al., 2024) architecture instead of the UNet. The arrow '→' indicates the accuracy before (left) and after (right) reward fine-tuning.

| | CIFAR10 | | |
| --- | --- | --- | --- |
| Corruption Level | 0 | 1 | 2 |
| SGD fine-tuning | $55.68 \to 55.96$ | $52.47 \to 54.00$ | $19.79 \to 42.54$ |
| N$\mathcal{M}$-CFM-GaussAug | $54.52 \pm 0.56 \to 54.65 \pm 0.57$ | $51.62 \pm 0.56 \to 52.43 \pm 0.72$ | $19.45 \pm 1.06 \to 30.06 \pm 0.80$ |
| N$\mathcal{M}$-CFM-MoNFN | $54.09 \pm 0.73 \to 54.17 \pm 1.70$ | $50.94 \pm 2.02 \to 51.50 \pm 1.01$ | $19.76 \pm 1.62 \to 27.03 \pm 1.65$ |

| | MNIST | | |
| --- | --- | --- | --- |
| Corruption Level | 0 | 1 | 2 |
| SGD fine-tuning | $92.55 \to 93.65$ | $86.92 \to 91.19$ | $9.51 \to 87.64$ |
| N$\mathcal{M}$-CFM-GaussAug | $92.63 \pm 0.18 \to 92.65 \pm 0.18$ | $86.22 \pm 0.73 \to 88.74 \pm 0.29$ | $9.51 \pm 0.01 \to 30.36 \pm 5.78$ |
| N$\mathcal{M}$-CFM-MoNFN | $92.06 \pm 0.33 \to 92.23 \pm 3.48$ | $83.52 \pm 2.05 \to 85.61 \pm 5.58$ | $9.51 \pm 0.00 \to 18.09 \pm 2.52$ |

## G.5 Discussion

### G.5.1 Failure cases

In this section, we note specific failure cases and discuss possible explanations.

**Gaussian N$\mathcal{M}$-JKO.** In prior experiments, we also attempted to use the Gaussian distribution as the source distribution in the N$\mathcal{M}$-JKO approach, but we observed failure (i.e. validation accuracy no better than chance) in all trials. We suspect this has to do with JKOnet's sensitivity to changes in scale given that the model output is simply a scalar value. Indeed, since the standard deviation and norm of parameters distributed by a Gaussian is considerably higher (about $10 - 100\times$) than the initialization (Kaiming uniform) or the converged weights, JKOnet would need to effectuate a large gradient $\nabla_x V(x, t)$ at small times $t$, and suddenly transition to small adjustments after the first intermediate distribution (for N$\mathcal{M}$-JKO(4) this would be $t = 0.2$).

**Stochasticity levels.** When employing N$\mathcal{M}$-CFM on a latent space created by a VAE, we observe good performance with a standard deviation $\sigma = 0.1$ when sampling the interpolant $x_t \sim p_t(x_t|x_0, x_1)$. However, this fails completely when applied to the raw weight space, where we instead set $\sigma = 10^{-3}$. This clearly indicates that to generate performant base models, the parameters cannot deviate by much from the converged parameters (at least for the CNN architecture).

**Diffusion models on weight space.** We also attempted to use the diffusion model from D2NWG directly on weight space (i.e. D2NWG without the VAE component). We observe some decrease in the loss but found it to be at least an order of magnitude higher than the N$\mathcal{M}$-CFM loss and the validation accuracy did not exceed 20%. We hypothesize that this relates to the issue of stochasticity discussed above. With N$\mathcal{M}$-CFM, one has greater control over the level of stochasticity as opposed to a diffusion model. Indeed, the forward process of diffusion requires noising towards a Gaussian distribution and so there is a clear tradeoff when we decrease the beta noise schedule $(\beta_t)_{t \in [0,T]}$: if $\beta_t$ is made small for a greater number of timesteps, then

the forward process will not reach a proper Gaussian distribution. Consequently, this adversely affects the reverse (inference) process as the model will have a poor understanding of the source distribution.

### G.5.2 Conclusion

From these results, several key observations emerge. First, given that diffusion models fail when applied directly in weight space—and considering the results in Table 7—we see clear advantages in end-generation precision when using an FM model over prior diffusion-based approaches. In particular, the ability to control the level of stochasticity appears important for achieving high base-model validation accuracy (see failure cases above). Second, the flexibility in choosing the source distribution also substantially affects the accuracy with which weight-space trajectories are modeled, as illustrated in Figure 6. This further supports the use of (MM)FM, which can accurately model weight-space data without requiring an autoencoder. Lastly, we observe that although layer alignment of permutation states helps simplify the training distribution resulting in easier training (see Table 7 and Figure 7), in the usual case of inference, where we use 100 steps instead of 1 or 2, and when modeling weight trajectory, the improvements are quite modest as our current architecture has the capacity to fit the training data well. In fact, when we move on to using equivariant architectures, we find that in the case of reward fine-tuning, this degrades downstream performance.

Table 9: Best validation accuracy of unconditional $N\mathcal{M}$ generation for various datasets. *orig* denotes base models trained conventionally by SGD and *p-diff* those generated with p-diff (Wang et al., 2024). We focus on generating just the batch norm parameters.

| Base Models | CIFAR100 | | | CIFAR10 | | | MNIST | | | STL10 | | |
|---|---|---|---|---|---|---|---|---|---|---|---|---|
| | orig. | $N\mathcal{M}$ | p-diff. | orig. | $N\mathcal{M}$ | p-diff. | orig. | $N\mathcal{M}$ | p-diff | orig. | $N\mathcal{M}$ | p-diff |
| ResNet-18 | 71.45 | 71.42 | 71.40 | 94.54 | 94.36 | 94.36 | 99.68 | 99.65 | 99.65 | 62.00 | 62.00 | 62.24 |
| ViT-base | 85.95 | 85.86 | 85.85 | 98.20 | 98.11 | 98.12 | 99.41 | 99.38 | 99.36 | 96.15 | 95.77 | 95.80 |
| ConvNext-tiny | 85.06 | 85.12 | 85.17 | 98.03 | 97.89 | 97.90 | 99.42 | 99.41 | 99.40 | 95.95 | 95.63 | 95.63 |

## H  Experimental details and further results

In this section, we provide further experimental details such as hyperparameters and computation times, alongside some extra results.

### H.1  Unconditional generation

Unconditional generation involves two stages: first is the training of base models. We choose a Resnet18, ViT-B, ConvNext-tiny, and CNN3 for our base models and provide the training parameters in Table 10. Next, we train the generative meta-model; Table 11 lists the training settings. We found that in most cases, the autoencoder and $N\mathcal{M}$-CFM converge after 1000 epochs. One should note that when using Kaiming uniform for the source $p_0$, it is also necessary to train the autoencoder on this distribution. On a NVIDIA A40 card, we estimate training time to be around 2 hours when the VAE is used. Un-encoded training generally takes double that time. Inference requires less than 1 minute to complete for both encoded and un-encoded runs, and it takes a few seconds to go through the validation set to compute the classification accuracy. Notably, the JKO runs were about $2\times$ **as fast** since the model output is a scalar. This means we can compress inputs through the downsampling layers of the UNet, leading to a smaller parameter count. Extra results are presented in Table 9.

**Remark on generating batch norm parameters.**  In order to reduce the target parameter count, we restrict ourselves to the batch norm parameters for larger architectures as in Table 9. We train base models in the same way as App. F.1 suggests, but for training we only save `state_dict` tensors with `bn` in the key. The rest of the model is saved for evaluation: after generating batch norm parameters, we impute these parameters back into the trained base models and validate as usual.

Table 10: Task training settings. Note that only CNN3 was evaluated for MMFM and JKOnet experiments, so it's the only model type with non-zero *Num. of save epochs* and *Savepoints per epoch.*

| Model Type | ResNet18 | ViT & ConvNext | CNN |
|---|---|---|---|
| Optimizer | SGD | AdamW | SGD |
| Initial training LR | 0.1 | $1 \times 10^{-4}$ | 0.1 |
| Training scheduler | MultiStepLR | CosineAnnealingLR | MultiStepLR |
| Layer params. saved | Last 2 BN layers | Last 2 BN layers | All layers |
| Num. of save epochs | 0 | 0 | 100 |
| Savepoints per epoch | 0 | 0 | 100 |
| Num. final weights saved | 200 | 200 | 200 |
| Saved parameter count | 2048 | 3072 | [10565, 12743] |
| Training epochs | 100 | 100 | 100 |
| Batch Size | 64 | 128 | 128 |

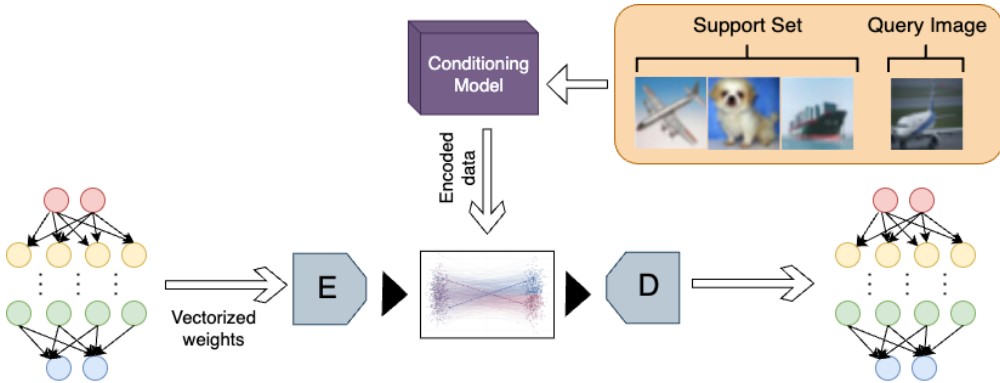

Figure 8: A schematic of the training process of N$\mathcal{M}$-CFM w/ VAE for conditional generation. Given a set of pre-trained target weights and a support set of images, we apply the conditioned flow model to pushforward a sample of the latent prior towards encoded target weights. The decoder is used during inference where we start from a sample of the latent prior and pushforward towards the target distribution with a trained vector field $v_\theta(\cdot, t; \boldsymbol{y})$ where $\boldsymbol{y}$ is the support set embedding.

## H.2  Model retrieval

For each dataset, we first sample 30 images as representatives, and then pre-compute its CLIP (Radford et al., 2021) embeddings. We then have a choice of how to aggregate the 30 embeddings. In our case, we use a light multi-head attention module and a linear layer to compress the context into a vector. Generally, we maintain the feature dimension—the version of CLIP we use returns a 768-dimensional vector. To condition the flow/diffusion model with this context, we concatenate to the channel dimension if we are using a VAE, or to the last dimension otherwise. One should note that the number of parameters differ between datasets; for instance, the three-channel datasets have slightly more parameters than their one-channel counterparts. Hence, we apply a simple zero-padding to standardize the input dimension. On a NVIDIA A40 card, we estimate a training time of 2 hours for the VAE and 3 hours for the CFM to achieve our level of accuracy (w/ VAE) and about 5 hours un-encoded. Inference times remain the same as in *unconditional generation.*

## H.3  Downstream initialization

In this evaluation, we obtain weights from model retrieval (albeit trained with weights before convergence), and fine-tune on their corresponding dataset, i.e. if we retrieve an MNIST classifier, we fine-tune on MNIST and $\widetilde{\text{MNIST}}$. Fine-tuning is done conventionally with the task training settings in Table 10. The corruption of datasets is done by applying the following transformations: random horizontal flip, random rotation (max 15 deg), color jittering, Gaussian blur (kernel size of 3, $\sigma \in [0.1, 2.0]$).

Table 11: Training settings for modules in the unconditional generation experiment. The number of JKOnet inference steps depends on the number of intermediate marginal distributions we are modeling.

| Model Type | Autoencoder | CFM w/ AE | MMFM | JKOnet |
|---|---|---|---|---|
| Optimizer | AdamW | AdamW | AdamW | AdamW |
| (LR, weight decay) | (1e-4, 2e-6) | (1e-4, 2e-6) | (3e-4, 2e-6) | (5e-3, 2e-6) |
| Num. inference steps | n/a | 100 | 100 | variable |
| Weight initialization | n/a | Kaiming uniform | Kaiming uniform | Kaiming uniform |
| Epochs | 500 | 1000 | 1000 | 1000 |
| Batch size | 64 | 64 | 64 | 64 |

## H.4   Reward fine-tuning

**Training settings.**   For the corresponding experiments, we used an AdamW optimizer with learning rate $2 \times 10^{-5}$ and weight decay $5 \times 10^{-4}$. We use a trajectory batch size of 8 (denoted $M$ in Algorithm 2) and a dataset batch size $m = 128/512$ depending on the dataset. We also clip gradients at norm 1.0 and set the number of fine-tuning iterations to 150. We use a cosine annealing scheduler with $\eta_{\min} = 10^{-6}$. The step size was set to $h = 0.025$, meaning our trajectory consists of 40 timesteps. As suggested in Domingo-Enrich et al. (2025, App. H), we only evaluated gradients at 20 out of 40 timesteps: 10 of the last timesteps and a uniform sample of 10 from the first 30 timesteps.

**Training tricks.**   We also introduce a few tricks. First, Algorithm 2 suggests that we must take a fine-tuning step every time we sample a dataset batch. We instead opt to average the starting lean adjoint $\tilde{a}_1$ over 3 batch samples; we found this to result in more stable training losses. In fact, this is all the iterations done per epoch, so if $N = 150$, we only have 450 batch iterations total. Moreover, as $\tilde{a}_1$ is the gradient of a classifier loss, we experimented with treating it like a stochastic gradient descent step, which means including a learning rate, momentum, and weight decay parameter. To clarify, this involves saving the $\tilde{a}_1$ values from previous training iterations. It does not seem to help much, other than the learning rate. We set the reward learning rate to 1.5 and momentum to 0.01.

**Padding regularization.**   Another trick specific to N$\mathcal{M}$-CFM-All is to use padding regularization. As mentioned in the model retrieval section, zero-padding is applied to standardize the input dimension given the variability between dataset classifiers. This trick indexes the padded elements in the input tensor and adds its $\ell_2$-norm to the loss, thus coercing it towards zero. Notably, this is an implicit method of conditioning on the dataset. For example, if we are regularizing more elements of the input, that suggests we are classifying on a one-channel dataset since these classifiers require more padding.

**Weight augmentation.**   In our experiments, we also tried augmenting the network weight data acquired from pre-training in an attempt to expand supp $\hat{p}_1$. This is done by simple Gaussian noise ($\sigma = 5 \times 10^{-3}$), dropout ($p = 0.02$), and mix-up. Recall that mix-up involves sampling a data pair $(w_1, w_2)$ and an interpolation parameter $\alpha \sim \text{Uniform}[0, 1]$, and returning an interpolation $(1 - \alpha)w_1 + \alpha w_2$.

**Corruption levels.**   As part of our experiments to get a sense of the width of classifier supports, we applied increasing corruption to the base datasets. The transformations are as follows:

1. Level 1: random horizontal flip, random rotation (max 15 deg).

2. Level 2: Level 1 + color jittering, and Gaussian blur (kernel size of 3 and $\sigma \in [0.1, 2.0]$).

3. Level 3: Level 2 + random erasing with $p = 0.5$, scale in $[0.2, 0.5]$, and ratio in $[0.3, 3.3]$.

**Support of classifier weights.**   To get a sense of how the weight distributions change as the dataset changes, see Table 12. In this experiment, we reward fine-tuned the N$\mathcal{M}$-CFM meta-model on increasingly corrupted versions of the base training dataset. The affect of the corruption is noticeable on the support as reward fine-tuning, which is constrained within the support set, fails to reach the accuracy of conventional

Table 12: Mean validation accuracy over five generated classifiers after reward fine-tuning on increasingly corrupted datasets. The -All suffix indicates that the CFM was trained on classifiers of CIFAR10, STL10, MNIST, and FMNIST, whereas +A indicates weight augmentation, and +P indicates regularization on padded values. The arrow '→' indicates the accuracy before (left) and after (right) reward fine-tuning.

| | CIFAR10 | | |
|---|---|---|---|
| Corruption Level | 0 | 1 | 2 |
| SGD fine-tuning | $63.38 \to 63.38$ | $59.93 \to 60.91$ | $24.18 \to 49.90$ |
| N$\mathcal{M}$-CFM | $62.53 \pm 0.02 \to 63.33 \pm 0.08$ | $58.65 \pm 0.22 \to 60.34 \pm 0.76$ | $24.84 \pm 0.93 \to 34.15 \pm 0.74$ |
| N$\mathcal{M}$-CFM-All | $28.92 \pm 11.19 \to 61.90 \pm 0.22$ | $23.57 \pm 16.43 \to 57.65 \pm 0.91$ | $22.72 \pm 1.56 \to 34.59 \pm 1.88$ |
| N$\mathcal{M}$-CFM-All+A | $45.07 \pm 15.64 \to 56.24 \pm 1.65$ | $37.53 \pm 15.16 \to 55.99 \pm 1.17$ | $19.89 \pm 4.11 \to 32.44 \pm 1.73$ |
| N$\mathcal{M}$-CFM-All+P | $50.18 \pm 18.49 \to 60.29 \pm 0.62$ | $51.36 \pm 14.51 \to 56.64 \pm 1.14$ | $21.84 \pm 3.69 \to 33.97 \pm 1.23$ |
| N$\mathcal{M}$-CFM-All+A+P | $39.25 \pm 17.68 \to 58.85 \pm 1.33$ | $41.59 \pm 15.05 \to 55.91 \pm 1.79$ | $19.21 \pm 3.30 \to 33.79 \pm 2.00$ |

| | MNIST | | |
|---|---|---|---|
| Corruption Level | 0 | 1 | 2 |
| SGD fine-tuning | $98.93 \to 98.93$ | $96.58 \to 97.78$ | $18.8 \to 97.55$ |
| N$\mathcal{M}$-CFM | $98.52 \pm 0.01 \to 98.79 \pm 0.04$ | $95.87 \pm 0.01 \to 97.01 \pm 2.27$ | $15.68 \pm 0.17 \to 91.21 \pm 3.05$ |
| N$\mathcal{M}$-CFM-All | $60.77 \pm 27.58 \to 97.56 \pm 1.20$ | $35.74 \pm 27.14 \to 59.32 \pm 27.88$ | $26.68 \pm 22.11 \to 65.40 \pm 34.17$ |
| N$\mathcal{M}$-CFM-All+A | $37.67 \pm 37.07 \to 95.84 \pm 0.53$ | $49.42 \pm 37.93 \to 92.50 \pm 8.99$ | $11.14 \pm 4.65 \to 35.32 \pm 28.25$ |
| N$\mathcal{M}$-CFM-All+P | $30.28 \pm 33.54 \to 95.72 \pm 1.03$ | $49.24 \pm 38.96 \to 94.65 \pm 0.41$ | $28.54 \pm 24.07 \to 78.87 \pm 21.20$ |
| N$\mathcal{M}$-CFM-All+A+P | $27.57 \pm 34.27 \to 95.88 \pm 0.35$ | $58.32 \pm 38.92 \to 94.59 \pm 0.30$ | $12.68 \pm 6.16 \to 88.64 \pm 3.92$ |

Table 13: Mean validation accuracy over five generated classifiers after reward fine-tuning on four datasets. The -All suffix indicates that the CFM was trained on classifiers of CIFAR10, STL10, MNIST, and FMNIST, whereas +A indicates weight augmentation, and +P indicates regularization on padded values. The arrow '→' indicates the accuracy before (left) and after (right) reward fine-tuning.

| | CIFAR10 | STL10 | MNIST | FMNIST |
|---|---|---|---|---|
| N$\mathcal{M}$-CFM-All | $28.92 \pm 11.19 \to 61.90 \pm 0.22$ | $42.21 \pm 11.68 \to 52.63 \pm 0.14$ | $60.77 \pm 27.58 \to 97.56 \pm 1.20$ | $43.12 \pm 34.22 \to 88.49 \pm 1.09$ |
| N$\mathcal{M}$-CFM-All+A | $45.07 \pm 15.64 \to 56.24 \pm 1.65$ | $27.21 \pm 11.44 \to 50.66 \pm 1.71$ | $37.67 \pm 37.07 \to 95.84 \pm 0.53$ | $55.49 \pm 33.85 \to 85.65 \pm 2.76$ |
| N$\mathcal{M}$-CFM-All+P | $50.18 \pm 18.49 \to 60.29 \pm 0.62$ | $22.05 \pm 6.32 \to 52.11 \pm 0.36$ | $30.28 \pm 33.54 \to 95.72 \pm 1.03$ | $32.07 \pm 29.62 \to 86.93 \pm 1.97$ |
| N$\mathcal{M}$-CFM-All+A+P | $39.25 \pm 17.68 \to 58.85 \pm 1.33$ | $21.27 \pm 5.06 \to 50.49 \pm 0.77$ | $27.57 \pm 34.27 \to 95.88 \pm 0.35$ | $46.20 \pm 34.08 \to 84.21 \pm 5.30$ |

fine-tuning. This holds true even for mild corruption schemes, suggesting the ideal classifier support on the corrupted set has little intersection with the original support, indicating narrowness of the set. To see that different classifiers have mostly disjoint supports, we try expanding supp $p_1^{\text{base}}$ by training on classifiers for different datasets. To verify this, we trained a new N$\mathcal{M}$-CFM model on classifiers for different datasets *without* any context conditioning. Since the fine-tuned target distribution ought to classify only one dataset, the hope is for fine-tuning to redirect the velocity field towards this one classifier distribution. This intuition turns out to be insufficient, as shown by the N$\mathcal{M}$-CFM-All rows in the table, further supporting our hypothesis. Moreover, this result holds with weight augmentations. The results also suggest that context conditioning is necessary for consistent validation accuracies, as convincingly shown in the MNIST case. Indeed, the padding regularization is an implicit form of context conditioning as the MNIST classifiers—expecting one-channel images—are slightly smaller than the 3-channel dataset classifiers. Further reinforcing our hypothesis, we also provide results of a N$\mathcal{M}$-CFM model trained on all four datasets fine-tuned to generate classifiers for each in Table 13.

**Computation times.** On a NVIDIA A40 GPU, one full training run takes about 2 and a half hours. During evaluation, we sample generated weights 5 times and validate on the test dataset; this test completes in under 5 minutes.

Table 14: True positive rate at the 5% significance level (TPR@5) and area under receiver operating characteristic curve (AUROC) for detection of harmful covariate shift on CIFAR10.1 and Camelyon17. We test on both the disagreement rate (DAR) and the entropy, setting $\lambda = \kappa/(|\mathbf{Q}| + 1)$. The best result for each column and our method are **bolded**.

| TPR@5 | CIFAR10 | | | Camelyon | | |
|---|---|---|---|---|---|---|
| $|\mathbf{Q}|$ | 10 | 20 | 50 | 10 | 20 | 50 |
| Detectron (DAR), $\kappa = 1$ | 0 | 0 | 0 | 0 | $.10 \pm .10$ | 0 |
| Detectron (DAR), $\kappa$ match | 0 | 0 | 0 | $.30 \pm .15$ | $.20 \pm .13$ | $\mathbf{.50 \pm .17}$ |
| **Meta-detectron (DAR)** | $.33 \pm .13$ | $.47 \pm .13$ | $.27 \pm .12$ | $\mathbf{.80 \pm .11}$ | $\mathbf{.40 \pm .13}$ | $.42 \pm .15$ |
| Detectron (Entropy), $\kappa = 1$ | $\mathbf{.60 \pm .17}$ | $.40 \pm .16$ | $.50 \pm .17$ | 0 | 0 | 0 |
| Detectron (Entropy), $\kappa$ match | $.50 \pm .17$ | $.10 \pm .10$ | $.20 \pm .13$ | $.10 \pm .10$ | 0 | 0 |
| **Meta-detectron (Entropy)** | $.27 \pm .12$ | $\mathbf{.93 \pm .07}$ | $\mathbf{.93 \pm .07}$ | 0 | 0 | $.27 \pm .12$ |

| AUROC | CIFAR10 | | | Camelyon | | |
|---|---|---|---|---|---|---|
| $|\mathbf{Q}|$ | 10 | 20 | 50 | 10 | 20 | 50 |
| Detectron (DAR), $\kappa = 1$ | 0.515 | 0.595 | 0.485 | 0.590 | 0.595 | 0.795 |
| Detectron (DAR), $\kappa$ match | 0.495 | 0.485 | 0.560 | 0.690 | 0.795 | **0.935** |
| **Meta-detectron (DAR)** | **0.849** | 0.838 | 0.938 | **0.900** | 0.760 | 0.806 |
| Detectron (Entropy), $\kappa = 1$ | 0.740 | 0.695 | 0.850 | 0.345 | 0.610 | 0.720 |
| Detectron (Entropy), $\kappa$ match | 0.735 | 0.730 | 0.820 | 0.510 | 0.455 | 0.600 |
| **Meta-detectron (Entropy)** | 0.716 | **0.987** | **0.996** | 0.747 | **0.836** | 0.847 |

Table 15: In-distribution validation accuracy before and after reward fine-tuning.

| | CIFAR10 | | |
|---|---|---|---|
| $|\mathbf{Q}|$ | 10 | 20 | 50 |
| Meta-detectron ($\mathbf{P}^*$) | $61.81 \pm 0.17 \rightarrow 61.01 \pm 0.14$ | $60.97 \pm 0.16 \rightarrow 60.49 \pm 0.25$ | $60.64 \pm 0.19 \rightarrow 60.32 \pm 0.30$ |
| Meta-detectron ($\mathbf{Q}$) | $60.84 \pm 0.27 \rightarrow 60.85 \pm 0.27$ | $61.61 \pm 0.30 \rightarrow 61.11 \pm 0.13$ | $60.90 \pm 0.15 \rightarrow 61.15 \pm 0.28$ |

| | Camelyon | | |
|---|---|---|---|
| $|\mathbf{Q}|$ | 10 | 20 | 50 |
| Meta-detectron ($\mathbf{P}^*$) | $92.75 \pm 0.24 \rightarrow 91.38 \pm 0.72$ | $92.84 \pm 0.21 \rightarrow 91.71 \pm 0.59$ | $92.63 \pm 0.15 \rightarrow 92.39 \pm 0.22$ |
| Meta-detectron ($\mathbf{Q}$) | $92.85 \pm 0.27 \rightarrow 92.47 \pm 0.21$ | $92.60 \pm 0.19 \rightarrow 92.44 \pm 0.34$ | $93.03 \pm 0.18 \rightarrow 92.70 \pm 0.42$ |

## H.5 Detecting harmful covariate shifts

**Training.** The only settings that were changed from reward fine-tuning is the learning rate, which is now $1.5 \times 10^{-5}$ and the number of fine-tuning epochs $N = 100$. The computation time varies between CIFAR10 and Camelyon17. The former completes in 2 hours, whereas the latter requires 3 and a half hours. The difference stems from the higher image resolution of Camelyon17, resulting in more parameters in the classifier.

**Choosing $\lambda$.** Following the exposition in Ginsberg et al. (2023), the choice of $\lambda$ can be motivated by a counting argument. We suppose that agreeing with the base classifier on a sample of $\mathbf{P}$ incurs a reward of 1 and disagreeing on a sample of $\mathbf{Q}$ incurs a reward of $\lambda$. Originally, to encourage agreement of $\mathbf{P}$ as the primary objective, $\lambda$ is set so that the reward obtained from disagreeing on *all* samples of $\mathbf{Q}$ is less than agreeing on just *one* sample of $\mathbf{P}$, i.e. $\lambda|\mathbf{Q}| < 1$, giving $\lambda = \frac{1}{|\mathbf{Q}|+1}$. However, this argument can be generalized slightly. As reward fine-tuning is a more conservative approach, we want to increase the reward for disagreeing on $\mathbf{Q}$. For instance, we may want the reward obtained from disagreeing on *all* samples of $\mathbf{Q}$ to be about the same as agreeing on $\kappa > 0$ samples of $\mathbf{P}$. This gives $\lambda = \kappa/(|\mathbf{Q}| + 1)$ as the $\ell_{dce}$ weight.

**Choosing $\kappa$.** We tried $\kappa = 1, 3, 6, 9, |\mathbf{Q}| + 1$ for Camelyon17 and $\kappa = 1, 32, 40, 50, |\mathbf{Q}| + 1$ for CIFAR. We ran two experiments with different weight settings. In our fixed run, seen in Tables 14 and 15, we used $\kappa = 4$ for Camelyon17 and $\kappa = 32$ for CIFAR10. The reason for the lower $\kappa$ values for Camelyon17 stems from the number of classes: recall that Camelyon17 is requires a binary classifier, whereas there are 10 classes

in the CIFAR10 dataset. The dataset batch size also matters: we used a batch size of 128 for Camelyon17 (as the images are larger) and 512 for CIFAR10. In the run where we varied $\kappa$ over the sample size $|\mathbf{Q}|$, we started at a reference point: $\kappa = 32$ for CIFAR10 at $|\mathbf{Q}| = 20$ and scaled naturally, i.e. $\kappa = 16$ when $|\mathbf{Q}| = 10$ and $\kappa = 80$ when $|\mathbf{Q}| = 50$. Likewise, we used the reference $\kappa = 4$ for Camelyon17 at $|\mathbf{Q}| = 20$ and scaled accordingly.

**Shift detection test.** We used a standard two-sample test identical to Ginsberg et al. (2023). Given the original $\mathbf{P}$ and the unknown $\mathbf{Q}$, we would like to rule out the null hypothesis $\mathbf{P} = \mathbf{Q}$ at the 5% significance level by comparing two statistics: *entropy* and the *disagreement rate*. The definition of entropy we use measures uncertainty over classes in the logits.

$$\text{Entropy}(x) = \sum_{c=1}^{N} \hat{p}_c \log \hat{p}_c \quad \text{where } \hat{p} = \frac{f(x) + g(x)}{2}, \tag{54}$$

where $f$ is our base classifier and $g$ is the generated classifier. In contrast to Detectron, we do not use CDC ensembles in our method. We use a Kolmogorov-Smirnov test to compute the p-value for covariate shift on the entropy distributions, comparing $g_{\mathbf{P}^*}$ and $g_{\mathbf{Q}}$. Intuitively, when $\mathbf{Q}$ is out-of-distribution, the generated classifier predicts with high entropy on $\mathbf{Q}$ and low on $\mathbf{P}^*$.

Regarding the disagreement rate, the null hypothesis is represented by $\mathbb{E}[\phi_{\mathbf{Q}}] \leq \mathbb{E}[\phi_{\mathbf{P}^*}]$ where $\phi$ is the disagreement rate and expectation is taken over trial seeds. This comes from the idea that its easier to learn to reject from a distribution that is not in the training set (since the base classifier $f$ will also be unsure). The test result is considered significant at $\alpha$% when $\phi_{\mathbf{Q}}$ is greater than the $(1 - \alpha)$ quantile of $\phi_{\mathbf{P}^*}$. In practice $\alpha = 5\%$.

**Further results.** See Tables 14 and 15 for the results when $\kappa$ is fixed. We also show more plots in Figures 9 and 10.

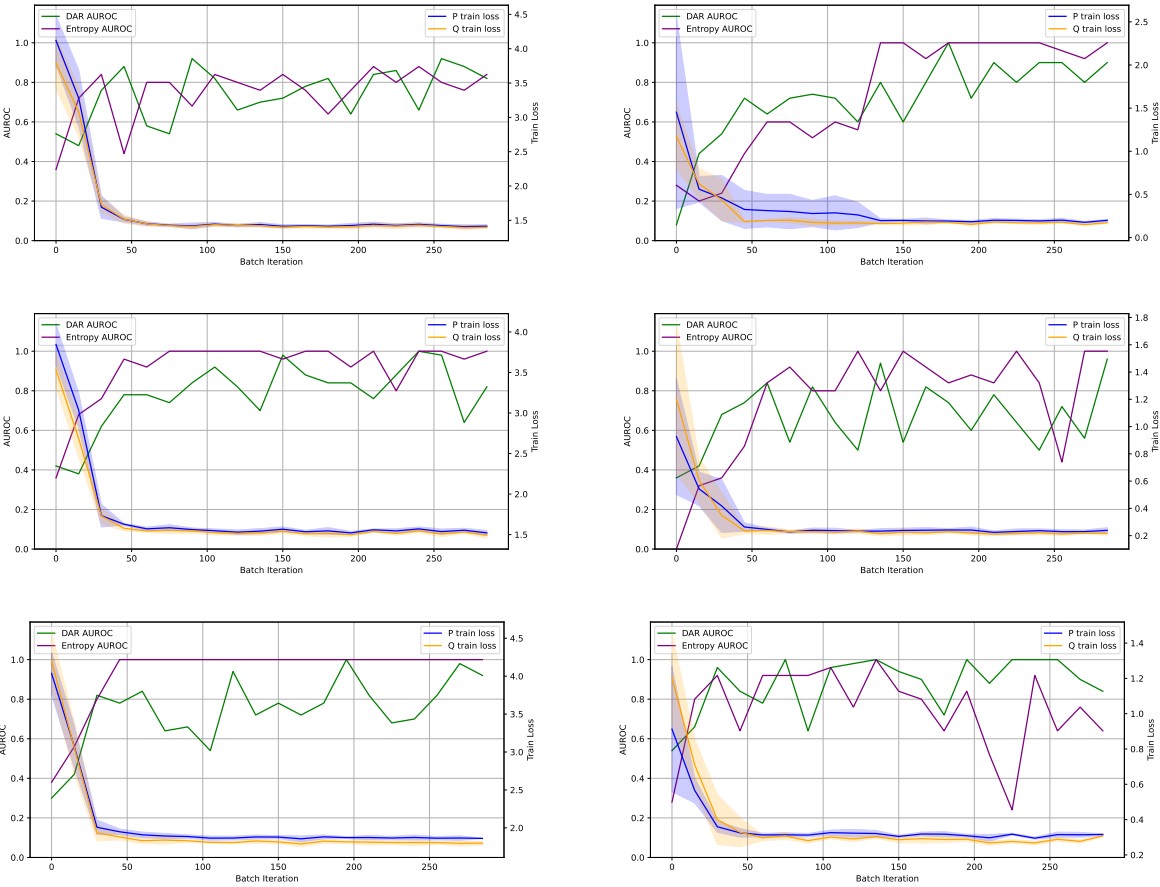

Figure 9: Plots illustrating how AUROC and $\ell_{cdc}$ evolves over reward fine-tuning iterations for CIFAR10 and Camelyon17 when $|\mathbf{Q}| = 10, 20, 50$ for *varying* $\kappa$ ordered from top to bottom.

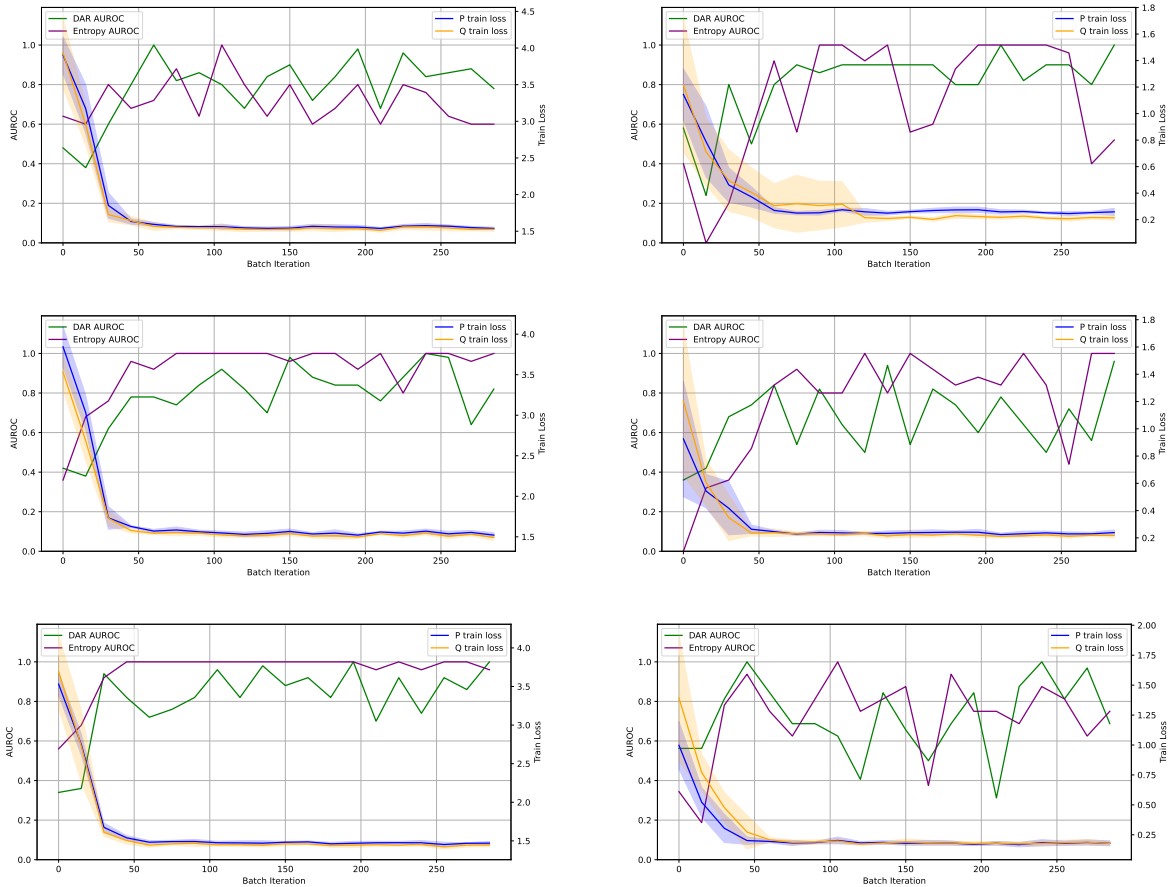

Figure 10: Plots illustrating how AUROC and $\ell_{cdc}$ evolves over reward fine-tuning iterations for CIFAR10 and Camelyon17 when $|\mathbf{Q}| = 10, 20, 50$ for *fixed* $\kappa$ ordered from top to bottom.

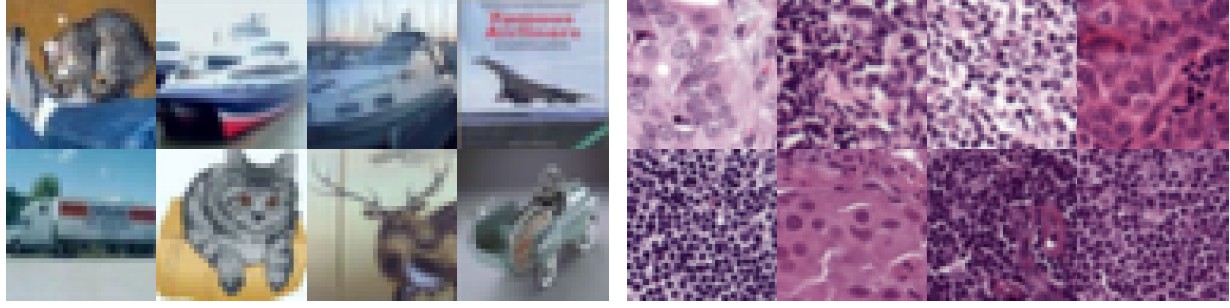

Figure 11: Samples of $\mathbf{P}^*$ (**top**) and $\mathbf{Q}$ (**bottom**) from the CIFAR10 (**left**) and Camelyon17 (**right**) evaluation of Meta-Detectron.

