# OpenReview forum: "Flows and Diffusions on the Neural Manifold"
_TMLR — Rejected by TMLR_

### Review · Reviewer_rwfy · 2025-07-30

**Summary Of Contributions:**

This paper explores using flow matching techniques over training
trajectories to generate neural network weights and detect covariate
shift.


**Strengths:** Reasonable technical foundations; tackles a relevant and
timely research problem; novel combination of known techniques.

**Weaknesses:** Design choices are not clearly motivated; unclear
advantages over existing work; relies heavily on access to full
optimization trajectories; architecture feels overly complex;
experiments are limited in depth; fails to address fundamental symmetries and invariances in neural network weight spaces.

**Additional Comments:**

The paper gives the impression that several complex tools were combined without clear reasoning for each. Instead of trying to do everything at once, consider narrowing the focus and digging deeper into fewer components. The covariate shift application feels tacked on: either integrate it more tightly into the core method or treat it as a separate contribution.

**Audience:**

Yes

**Audience Explanation:**

Even with its flaws, the paper addresses a topic that is of growing interest: learning distributions over weights instead of just working in data space. This touches on areas like Bayesian deep learning, ensembling, and OOD detection, which are relevant to many in the TMLR community.

**Claims And Evidence:**

No

**Claims Explanation:**

While the theory is generally sound, the paper does not clearly explain why flow matching is better than other options like diffusion models (e.g., see [https://openreview.net/pdf?id=j8WHjM9aMm](https://openreview.net/pdf?id=j8WHjM9aMm)). The experiments are limited and do not strongly back up the main claims. The overall presentation makes it hard to understand how the different components fit together. The flow matching, VAE, and reward fine-tuning parts feel more like separate modules than a coherent system.

**Requested Changes:**

**Critical:**

* Clearly explain why flow matching is used instead of alternatives like
  diffusion or simple trajectory matching (e.g. Diffusion-based Neural
  Network Weights Generation paper).
* Discuss the practicality of the method: how are these training trajectories collected, and is that realistic for larger models?
* Simplify and better integrate the model architecture. Right now, it feels overcomplicated without strong justification.
* Improve writing clarity: cut down on jargon and explain the motivation more clearly and directly.
* Expand experiments, especially by comparing to simpler baselines and prior work.
* Include an analysis of compute/storage requirements for collecting and using weight trajectories.
* Better justify specific architectural choices (e.g., why a VAE? why this reward tuning setup?).

**Strengthening:**

* Address how the approach handles neural network symmetries and invariances present in neural network architectures. Modern deep networks exhibit numerous invariances including permutation symmetries within layers, scaling invariances, and architectural constraints that define valid weight configurations. Without considering these invariances, the generative modeling problem over weights becomes unnecessarily high dimensional whereas the authors themselves rely on the manifold hypothesis to motivate their work but are contradicting this by ignoring the manifold structure imposed by these symmetries. This represents a fundamental concern that undermines the theoretical foundation of the approach. The authors should provide compelling reasoning for why their approach remains valid despite this limitation and appropriately scale back their claims about the effectiveness and generality of their method. At minimum, acknowledge this limitation explicitly and discuss how it affects the interpretation of their results.

---

> ### Author Response · Authors · 2025-08-29
> **Response to Reviewer rwfy**
>
> We thank the reviewer for their thoughtful feedback and especially for highlighting the issue of parameter symmetries, which helped us better situate our work within the broader context of weight-space generative modeling. Below, we address your points in turn, highlighting revisions in violet text.
>
> ## 1. Choice of Flow Matching vs. Alternatives (Critical Point 1)
> We have expanded our justification for using flow matching (FM) over diffusion models (see App. G for experiments) or simple trajectory matching. In particular:
> - Flexible choice of prior: As shown in new experiments (App. G.2-G.3), FM allows for flexible priors (e.g., Gaussian vs. Kaiming initialization), which significantly influences trajectory quality.
> - Better control over stochasticity: Diffusion-based approaches struggled to generate raw parameters, likely due to stochasticity and the prior. For instance, CFM degrades when $\sigma > 0.01$, motivating our formulation of a deterministic adjoint-matching method.
> - Handling multiple marginals: FM naturally supports multiple marginals (effectively multiple CFMs), which diffusion struggles with due to Gaussian prior constraints.
> - Comparison to other trajectory matching: Many alternatives are either not simulation-free or assume prior trajectory knowledge. FM, by contrast, is simulation-free and makes fewer assumptions, making it more practical in this setting.
> - Relation to prior work: While models such as D2NWG (VAE-based) mitigate some issues through KL regularization, our experiments show FM provides a more direct and versatile formulation.
>
> ## 2. Scaling and Practicality (Critical Points 2, 6)
> We added discussion in App. F.5 on scalability, with explicit remarks about how to scale with respect to weight trajectories. This includes considerations of memory, training time, and the potential role of more scalable encoders.
>
> ## 3. Writing and Presentation (Critical Points 3, 4, 7)
> We substantially revised the manuscript to improve clarity and organization:
>
> - Section 3 has been split into two parts. Section 3.2 now begins with the concept of the action gap, followed by a unified view of different objectives (MMFM and JKOnet) as minimizing this gap. This restructuring makes the conceptual through-line much clearer.
>   - Section 3.2.3, which contained less central theory, has been streamlined and moved to App. A.3. We retain an overview in the main text to maintain connections with related methods (e.g., Metric Flow Matching).
> - Section 4 now clearly frames our method as modular, aligning better with our intention to present this modular view of weight generative modeling. Unlike prior work that proposed a singular system for weight generation, we strike a connection with other more developed fields of generative modeling such as images: there is a central generative module, an encoder possibly for efficiency, and a post-hoc fine-tuning method. These modules are flexible and they may be used depending on the desired task, compute resources, or time requirements. To make this study feasible, we made simple, representative choices for each module, often inspired by prior work, but others could be made. Although, we emphasize that the meta-fine-tuning module has yet to be proposed in prior work on weight generation. We also provide some justification of each component. In brief:
>   - Encoder – a natural route to scale the method; we chose VAE due to its simplicity and its usage in the Diffusion-based Neural Network Weights Generation (D2NWG) paper; we suggest alternatives, some with symmetry-aware architectures.
>   - Generative meta-model – justified by experiments in App. G and the rationale in Point 1.
>   - Reward fine-tuning module – a novel and efficient approach via adjoint matching, competitive with or exceeding backpropagation-based fine-tuning while being less computationally expensive. We emphasize its flexibility, theoretical foundation (App. F.4), and applicability beyond our immediate experiments (e.g., handling covariate shift).
>
> This restructuring clarifies our intent: to propose a general framework for weight generative modeling that is modular, interpretable, and extensible, rather than prescribing a single rigid system.
>
> ## 4. Experiments (Critical Point 5)
> We included new experiments in App. G that:
>
> - Compare marginal distributions via $W_1$ trajectory distances,
> - Explore few-step inference, and
> - Investigate the role of parameter symmetry (see also Point 5 below).
>
> These additions strengthen both empirical coverage and the interpretation of our results.

---

> ### Author Response · Authors · 2025-08-29
> **Response to Reviewer rwfy Part 2**
>
> ## 5. Parameter Symmetries (Strengthening)
> We agree that parameter symmetries are important and made the following updates:
>
> - In our additional experiments, we touch on parameter symmetry by exploring two strategies: (i) aligning trajectories to a reference model and (ii) using an equivariant architecture.
> - Found that alignment improves underfitting cases but has limited effect in full inference (100 steps), where our methods already perform well. For instance, see the trajectory modeling experiments. Thus, we do not think that this effects the interpretation of results.
> - Acknowledged this gap explicitly: a footnote in the introduction, and we emphasize parameter symmetries as an important direction for future work.
>
> ## Walkthrough of claims (Section 1)
> > We unify and prove characterizations of various methods to approximate a
> gradient descent trajectory, enabling more accurate modeling of our priors.
> >
> We use the notion of the action gap and relate objectives of trajectory modeling methods to this notion in Section 3.
> > We incorporate theoretical considerations to design flow- and diffusion-based approaches for generating weights that match or exceed
> conventionally trained models on in-distribution tasks, provide better initializations for downstream training, and allows for conditioning on context data to retrieve pre-trained weights from a distribution pre-trained on various datasets.
> >
> See experiments on unconditional generation, downstream initialization, and model retrieval. Additionally, App. G provides further upsides of our approach vs. prior work.
> > We incorporate a fine-tuning mechanism, grounded in adjoint matching (Domingo-Enrich et al., 2025), to enhance performance.
> >
> Experiments in Section 5.2 and App. H.4.
> > We show how this can be used to detect harmful covariate
> shifts that outperforms the closest comparable baseline, supporting our motivation to reinterpret problems as
> questions on weight space.
> >
> Section 5.3.
>
> We hope these revisions and clarifications address the reviewer’s concerns and strengthen the paper’s contribution. Please let us know if there are further adjustments we can make to ensure the work is as clear and rigorous as possible.

---

> ### Author Response · Authors · 2025-09-06
> **Follow-up**
>
> Dear Reviewer rwfy,
>
> Thank you again for reviewing our work! As the discussion period ends shortly, we wanted to check if you have any further questions or found our responses helpful?
> Please let us know and thank you for your time!

---

> > ### Comment · Action_Editor_Syvs · 2025-09-14
> > **Call for Discussion**
> >
> > Hi Reviewer rwfy,
> >
> > Can you please respond to the author's reply, or at the very least indicate whether or not the author's reply addressed your concerns.
> >
> > Thanks

---

> > > ### Comment · Reviewer_rwfy · 2025-09-15
> > > **Response to authors**
> > >
> > > Thanks for your rebuttal. Here are my comments:
> > >
> > > * Regarding FM motivation, this response doesn't justify why flow matching is superior to diffusion models for neural network weight generation. The listed technical features lack connection to the specific challenges of modeling parameter spaces.
> > >
> > > * Regarding scalability, F.5 reveals severe limitations rather than solutions - only tested on 10K parameter models with 2 hours GPU training per dataset. Proposed workarounds are speculative, not demonstrated.
> > >
> > > * Regarding parameter symmetries, inadequately addressed. You acknowledge this "fundamental concern" but dismiss it with limited experiments claiming "limited effect." Relegating core theoretical gaps to "future work" doesn't resolve the undermined foundations of your approach.
> > >
> > > * Regarding integration, reframing components as "modular" doesn't fix the loose coupling between VAE, flow matching, and reward fine-tuning components.
> > >
> > > The response shows organizational improvements but fails to address the most substantive theoretical and methodological concerns raised in the original review.

---

> > > > ### Author Response · Authors · 2025-09-16
> > > > **Response to Comment**
> > > >
> > > > We thank the reviewer for their thoughtful comments. Before addressing them point-by-point, we would like to clarify the scope and objectives of the paper. As stated in paragraph 3 of the Introduction:
> > > > >In this paper, we introduce flow matching as a new class of methods for generating neural network weights, designed to incorporate structural priors such as training trajectories and source distributions.
> > > > >
> > > > And further, from the preceding paragraph:
> > > > >…we argue that learning to generate neural network weights opens a new perspective: it allows us to reinterpret diverse problems as questions on weight space. We illustrate this view in Section 5 through an application to detecting harmful covariate shift.
> > > > >
> > > > With this framing in mind, we respond to the reviewer’s specific concerns. If the reviewer finds this helpful, we will gladly *incorporate the response into the manuscript.*
> > > >
> > > > ## On FM motivation (vs. diffusion models):
> > > > - The relevance of FM is demonstrated in the experiments in Appendix G. For instance, diffusion models fail to directly generate neural network weights with the simple UNet architecture we used.
> > > >    - Direct approaches typically require tokenization or encoding methods [1], whereas FM can operate **directly** in parameter space.
> > > >    - We also observe that FM exhibits similar failures when the noise parameter $\sigma$ is set too high, suggesting that stochasticity itself is the limiting factor (and FM's greater control over this is beneficial).
> > > > - FM naturally generalizes to handling multiple marginals, effectively corresponding to consecutive CFMs [2, Prop. 2], whereas no analogue exists in diffusion models due to Gaussian prior constraints.
> > > >    - This is useful for modeling weight space trajectories as shown in Section 5.1 and App. G.3.
> > > > - FM further allows flexible prior choices, with clear effects shown in Figures 6 and 7.
> > > > - In addition, FM demonstrates superior performance in few-step regimes (Appendix G.2), which suggests that it may enable more efficient inference than diffusion models.
> > > > - These advantages align with observations in other domains such as biomolecule generation [3, 4], which is *similarly applicable for weight space modeling.*
> > > >
> > > > ## On scalability:
> > > > - Our experiments are conducted on models with ~10K parameters due to compute limitations on a shared GPU as we tested a broad set of methods beyond unconditional generation, and importantly, *we do not claim scalability as a primary contribution of this paper.*
> > > > - We included the scalability discussion because it is an important consideration. To this end, we used VAEs, showing in Tables 1–3 that they match direct methods in small-scale regimes. This empirical evidence *supports our speculation that VAEs may be a promising component for scaling to larger settings*.
> > > >
> > > > ## On parameter symmetries:
> > > > - The reviewer characterizes parameter symmetries as a “fundamental concern,” but we respectfully disagree that they *undermine the theoretical soundness* of the proposed methods. Rather, they present a challenge for model fitting.
> > > >    - Indeed, several prior works (e.g., D2NWG) do not account for such symmetries.
> > > > - To address this gap, we both acknowledged this caveat in the Introduction *and* we include modest experiments in App. G using two strategies that account for these symmetries. In the latter, we indeed observed limited effects and provide a discussion of potential reasons.
> > > > - More broadly, we emphasize that generative modeling on weight space is a large and diverse area. Our paper intentionally limits its scope to structural priors, training trajectories, and reward fine-tuning applications. Within this scope, parameter symmetries represent an important but secondary direction, which we therefore identify as future work.

---

> > > > ### Author Response · Authors · 2025-09-16
> > > > **Response to Comment Part 2**
> > > >
> > > > ## On integration of components:
> > > > - We respectfully disagree that the modularity of our components indicates loose coupling. Each component makes a distinct contribution to the pipeline: VAEs address efficiency, FM provides the generative modeling framework, and reward fine-tuning enables downstream adaptation. Their complementary roles justify their inclusion, even though they aren't necessary in all cases.
> > > >    - New to recent revisions, we also described an example pipeline in Figure 1.
> > > > - Following earlier reviewer feedback, we have strengthened the connection between theory (Section 3) and methodology (Section 4). The revised opening paragraph of Section 3.2 now explicitly motivates the generative model component, tying the theoretical framing more clearly to the methods that follow.
> > > >
> > > > ## Closing
> > > > We appreciate the reviewer’s concerns regarding scalability, parameter symmetries, and integration. However, we emphasize that our contributions are defined by the scope outlined in the Introduction (quoted above). Within this scope, we believe our theoretical discussion, empirical validation, and proposed extensions represent a coherent and meaningful advance. While we *agree* that scaling and parameter symmetries warrant deeper investigation, these constitute *separate strands of research*, and we therefore flag them as directions for future work rather than claiming to resolve them fully here. **If there are specific parts of the manuscript that give the impression we are overstating our claims, we would be grateful for guidance so that we can revise those passages more precisely.**
> > > >
> > > > [1] Peebles, W., Radosavovic, I., Brooks, T., Efros, A. A., & Malik, J. (2022). Learning to learn with generative models of neural network checkpoints. arXiv preprint arXiv:2209.12892.
> > > >
> > > > [2] Rohbeck, M., De Brouwer, E., Bunne, C., Huetter, J. C., Biton, A., Chen, K. Y., ... & Lopez, R. (2025). Modeling complex system dynamics with flow matching across time and conditions. In The Thirteenth International Conference on Learning Representations.
> > > >
> > > > [3] Yim, J., Campbell, A., Foong, A. Y., Gastegger, M., Jiménez-Luna, J., Lewis, S., ... & Noé, F. (2023). Fast protein backbone generation with se (3) flow matching. arXiv preprint arXiv:2310.05297.
> > > >
> > > > [4] Alex Morehead, Jianlin Cheng (2025). FlowDock: Geometric flow matching for generative protein–ligand docking and affinity prediction. Bioinformatics.
> > > >
> > > > [5] Schürholt, K., Mahoney, M. W., & Borth, D. (2024). Towards scalable and versatile weight space learning. arXiv preprint arXiv:2406.09997.
> > > >
> > > > [6] Wang, K., Tang, D., Zhao, W., Schürholt, K., Wang, Z., & You, Y. (2025). Recurrent diffusion for large-scale parameter generation. arXiv preprint arXiv:2501.11587

---

### Review · Reviewer_HWBG · 2025-08-07

**Summary Of Contributions:**

This paper tries to propose a framework for generating neural network weights using flow and diffusion-based generative models. The authors model the trajectory induced by gradient descent as a trajectory inference problem and aim to unify several trajectory inference techniques under a theoretical framework. The paper explores various architectural and algorithmic enhancements, such as reward fine-tuning, autoencoders for latent weight representation, and conditioning on task-specific context data. The proposed method is evaluated on tasks like generating in-distribution weights, improving initialization for downstream training, and detecting harmful covariate shifts.

**Additional Comments:**

- Tables 1 and 7 are confusing. For my understanding, the choices listed in the row are only valid for the proposed N$\mathcal{M}$. If I am correct, "orig." and "p-diff" should not have these rows in this table. Additionally, the use of || in this table is not common.
- Overall, all the tables and figures need more detailed explanations in their captions or body. Especially, the legends in some plots (e.g., Figure 2) are not sufficient.

**Audience:**

Yes

**Audience Explanation:**

Some readers may be interested in the applications of diffusion- and flow-based models to weight space learning.

**Broader Impact Concerns:**

I don't have particular concerns to report.

**Claims And Evidence:**

No

**Claims Explanation:**

### Overall Comments
This paper has major problems in its presentation. I believe this paper requires thorough revisions. It is unclear what the main technical contributions and benefits of the proposed methods are. More detailed comments can be found below.

### It is hard to understand the contributions of this paper
The major issue lies in the presentation. The paper introduces a set of concepts and theories without clear explanations on how they are used. In short, Section 3 is too hard to follow. Below, I list some items that need clarification:
- Action matching and JKOnet are introduced in Section 3.2.1; however, it is not clear how they are unified in this paper. I expected the unification of them to be discussed in this subsection or the next subsection. However, I could not find such a discussion. From my perspective, they are just presented as separate concepts.
- MMFM is introduced in Section 3.2.2. However, I faced difficulty in understanding how it will be used in the proposed method.
- It is not clear what the "proxy curve" mentioned in Section 3.2.3 is. They just mention it is a generalization of $\mu_t$, which is also unclear. Furthermore, it is also not clear in what sense "close" is used in the first sentence.
- They introduced the "action gap" in Section 3.2.3. I do not see how it is used in this paper and its connection with Proposition 2.
- I did not understand the main message from Theorem 4. This criticism applies to the other theoretical claims as well. They are presented without a clear description of how they are useful.
- For my understanding, Section 3.3 is the only section presenting their method. This section lists three components: weight encoder, generative meta-model, and reward fine-tuning. However, it is not clear how they are incorporated into the proposed model.
- The connection between what is presented in this section and the main problem, harmful covariate shifts detection, is not clear. Because this problem is the main scope of this paper, the application of the proposed methods to it should be placed in this section instead of in Section 4.

### It is hard to interpret the experimental results
This section shows some experimental results with few explanations and interpretations. In short, Section 4 is too hard to understand what the take-away messages are. Below, I list some items that need clarification:
- Tables 1 and 7 compare the proposed methods with conventional methods. However, there is no explanation about the results, making it difficult for readers to understand what the authors want to confirm with these results. The proposed methods achieve similar results to the conventional methods. Is this expected?
- In Section 4.1, they mention that they generate only batch norm parameters for ResNet-18, ViT-base, and ConvNext-tiny to reduce the size of the target parameters. However, it is not clear how they apply their method only to such a portion of network parameters.
- It is not clear what each curve indicates in Figure 2, mainly because there are no explanations about the legend. Which curves correspond to your proposed methods?
- At the beginning of Section 4, they mention that "we confirm various properties that are to be expected of weight generation models." However, even after reading Section 4.1, I did not understand what those properties were.
- The connection between each subsection is not clear. These subsections appear independent of each other due to the poorly organized structure of this section.

**Requested Changes:**

Please see my comments above.

---

> ### Author Response · Authors · 2025-08-29
> **Response to Reviewer HWBG**
>
> We thank the reviewer for their detailed comments and, in particular, for highlighting areas of presentation that required clarification. These critiques have helped us significantly improve the structure and readability of the paper. For instance, we have split up the theoretical discussion and the methods into Sections 3 and 4. Below, we address each point in turn, highlighting revisions in violet text.
>
> ## Concerning paper contributions
>
> 1. We agree that the connection between Action Matching and JKOnet was previously unclear. In the revised manuscript, Section 3.2 now begins with the notion of the action gap, and we explicitly show how both objectives (MMFM and JKOnet) can be recast as minimizing the action gap in the discretization limit. We have added some clarifying words in Section 3.2.2 in the JKOnet paragraph and the Theorem 2 statement to streamline the connection.
>
> 2. We now make clear that MMFM is one possible choice for the generative meta-model within our modular framework. Section 4 introduces the framework as comprising three interchangeable modules---encoder, generative meta-model, and reward fine-tuning---so MMFM can be substituted with alternatives such as JKOnet, Conditional Flow Matching, or diffusion. For implementation details, please see App. F.3.
>
> 3. We have significantly revised Section 3.2.3 in response to this concern:
> - Most technical details have been moved to App. A.3, as they are not directly used in our experiments.
> - In the main text, we now explain that proxy curves are a generalization of the linear interpolations in Eqns. 8 and 10. While those are straight-line interpolations, proxy curves allow for more general trajectories (e.g., optimized or learned curves from methods such as Metric Flow Matching or variational formulations of the action).
> - To avoid confusion, we considered the more conventional term “interpolant,” but since we already use that word in another context, we opted for “proxy curve.”
>
> 4. As mentioned in Point 1, we attempt to recast the objectives of other approaches (JKOnet and MMFM) as minimizing the action gap (at least in the discretization limit). When discussing proxy curves, we also define an analogue of the action gap, reinforcing continuity. Our intentions have been clarified by separating the action matching background into its own Section 3.2.1 and adding some preamble for Section 3.2 to overview the structure.
>
> 5. Theorem 4 has been moved to the appendix, with a clearer summary in the main text: it formalizes that if the gradient path is smooth enough, we can define a functional $V$ that ensures closeness to the loss gradient. This statement illustrates the type of guarantees motivating our framework, while technical details are left in the appendix.
> We hope that with the re-formatting of Section 3, other results, such as Theorem 2, becomes more clear.
>
> 6. We agree that our earlier organization blurred the distinction between theoretical exposition and the practical framework. To address this:
> - The proposed method now has its own Section 4, which clarifies our intention of proposing a **modular framework**, consisting of three components: encoder, generative meta-model, and reward fine-tuning.
> - We emphasize that these modules are exchangeable depending on the task, compute budget, or architectural considerations.
> - The role of reward fine-tuning is clarified: it operates on a pre-trained generative meta-model, and implementation details (architectural configuration, training settings) are now collected in Apps. F and H.4.
>
> 7. We agree that this application should be part of the methods section rather than deferred. Accordingly, Section 4.2 now directly situates our framework in the context of harmful covariate shift detection, with a few words added to the motivation to better align the proposed approach with the modular framework.

---

> ### Author Response · Authors · 2025-08-29
> **Response to Reviewer HWBG Part 2**
>
> ## Concerning results
>
> 1. We have included a brief discussion in the corresponding paragraph.
> 2. We remark briefly in App. H.1.
> 3. We have included some more details in the caption. In brief: MMFM_k refers to MMFM with k intermediate marginal distributions (that is, distributions in addition to $p_0$ and $p_1$) and likewise for JKO.
> 4. We agree that “properties” may not be the best term for this. Essentially, these are experiments that show basic tasks weight-space generative models are expected to fulfill, such as unconditional generation, retrieval given conditioning information, and a simple downstream task, all to show we are able to accurately model weight space and match/exceed baselines. In place, we went with “Generative modeling desiderata”, but we are open to suggestions.
> 5. To clarify our intention, the framework we wish to build is a modular one with three components: the first two (encoder and generative meta-model) is for conventional generation, whereas reward fine-tuning is an extra component that has not been explored at least for diffusion and flow-based meta-models. Therefore, we arrange the experiments to start with a familiar foundation of just using the encoder and generative meta-model. And then we expand to our new module of reward fine-tuning, alluding to some results in the appendix, and a claim about the support of classifier weights. Finally, we make use of this new module on a pertinent application.
>
> ## Additional Comments
> - Thank you for pointing this out. Originally this table had base model architectures along the rows, much like Table 9 (prev. Table 7). We have reformatted Table 1 to clarify and also removed the CNN3 section of Table 9 so that the old formatting makes sense.
> - We have specified the legend of some plots, including Figure 2 and Tables 2 and 3.
>
> ## Walkthrough of claims (Section 1)
> > We unify and prove characterizations of various methods to approximate a
> gradient descent trajectory, enabling more accurate modeling of our priors.
> >
> We use the notion of the action gap and relate objectives of trajectory modeling methods to this notion in Section 3.
> > We incorporate theoretical considerations to design flow- and diffusion-based approaches for generating weights that match or exceed
> conventionally trained models on in-distribution tasks, provide better initializations for downstream training, and allows for conditioning on context data to retrieve pre-trained weights from a distribution pre-trained on various datasets.
> >
> See experiments on unconditional generation, downstream initialization, and model retrieval. Additionally, App. G provides further upsides of our approach vs. prior work.
> > We incorporate a fine-tuning mechanism, grounded in adjoint matching (Domingo-Enrich et al., 2025), to enhance performance.
> >
> Experiments in Section 5.2 and App. H.4.
> > We show how this can be used to detect harmful covariate
> shifts that outperforms the closest comparable baseline, supporting our motivation to reinterpret problems as
> questions on weight space.
> >
> Section 5.3.
>
> We believe these revisions improve clarity, unify the theoretical and methodological contributions, and directly connect the framework to its motivating application. We hope this addresses the reviewer’s concerns, and we welcome any further suggestions that could strengthen the work.

---

> ### Author Response · Authors · 2025-09-02
> **Response Addendum**
>
> Following a comment from another reviewer, we misunderstood the point requiring clarification for *Contributions* (6) of your review. In our revision we had focused on clarifying the role of each component, but we now see that what was missing was an end-to-end picture of the pipeline itself.
>
> To address this, we have added a schematic (new Figure 1) that illustrates the pipeline and its stages. We reference this figure with additional notes in the opening of Section 4.1. While the figure provides one concrete example, we emphasize that our framework is modular and components may be swapped out. For instance, NM-CFM could be replaced by NM-JKO or NM-MMFM, with the corresponding change in the generative loss.
>
> We hope this resolves the earlier confusion and makes the pipeline clearer. Please let us know if any parts remain unclear, or if there are other points in the original response that would benefit from further clarification.

---

> ### Author Response · Authors · 2025-09-06
> **Follow-up**
>
> Dear Reviewer HWBG,
>
> Thank you again for reviewing our work! As the discussion period ends shortly, we wanted to check if you have any further questions or found our responses helpful?
> Please let us know and thank you for your time!

---

> > ### Comment · Action_Editor_Syvs · 2025-09-14
> > **Call for Response**
> >
> > Hi Reviewer HWBG,
> >
> > Can you please respond to the author's reply, or at the very least indicate whether or not the author's reply addressed your concerns.
> >
> > Thanks

---

> > > ### Comment · Reviewer_HWBG · 2025-09-14
> > >
> > > Thank you for your rebuttal and the revised manuscript. While your revisions partially address my initial concerns regarding the overall structure of the paper, I still find the connection between Sections 3 and 4 unclear. The contributions in each section appear independent, as it seems that the theoretical results presented in Section 3 are not directly leveraged in the proposed methodology. This separation makes it difficult to understand the **main** goal of the paper.
> > >
> > > Furthermore, it remains unclear how the theoretical developments in this paper are beneficial. For example, in the discussion surrounding Theorems 1 and 3, I do not see clear take-away messages or intuitive interpretations from the authors. Additionally, some paragraphs appear superfluous, such as the one immediately following Theorem 3.

---

> > > > ### Author Response · Authors · 2025-09-14
> > > > **Response to Comment**
> > > >
> > > > Hello, and thank you for the follow up. Below we attempt to make the connection between Sections 3 and 4 more explicit, and if this framing is satisfactory, we will integrate it directly into the paper.
> > > >
> > > > The overall goal of the paper, as stated in the Introduction (para. 3), is:
> > > >
> > > > >In this paper, we introduce flow matching as a new class of methods for generating neural network weights, designed to incorporate structural priors such as training trajectories and source distributions.
> > > > >
> > > > Considering prior work on trajectory modeling, we know of various methods in the space of flow-based models that can be used to model paths $(\theta_t)_{t \geq 0}$ of weights. The representative methods we present in Section 4: CFM, MMFM, and JKOnet* **all** attempt to model the drift of a continuity equation (CE) (Eqn. 3); see [1, Theorem 1], [2, Sec. 2.2]. This is the motivation for recalling Theorem 1 in Section 3.1 and positioning it as a unifying backdrop.  Furthermore, by introducing the notion of the action gap (Section 3.2.1), we show that these seemingly different objectives can be interpreted under a common lens, as formalized in Theorem 2. Since the methods mentioned all start from the CE, we focused instead on communicating the action gap lens.
> > > >
> > > > *In short, Section 3 provides the unifying perspective (via the action gap and CE) that underpins the **choice of methods (CFM, MMFM, JKO; and to an extent, the reward fine-tuning)** in Section 4.*
> > > >
> > > > Additionally, Section 3.2.3 discusses the action gap lens in a more general setting (more in App. A.3). This is highlighted in the paragraph after Theorem 3 (the *proxy action gap*) and likewise the purpose of Theorem 3 was to show that even under these generalizations, the resulting dynamics can still be written in CE form, tying back to Theorem 1. Since we move most of the discussion to App. A.3, we also used this paragraph to inform the reader of the results in the Appendix and provide an intuitive picture of Theorem 4.
> > > >
> > > > Please let us know if this revised framing addresses your concern; if so, we will incorporate these clarifications into the paper.
> > > >
> > > > [1] Yaron Lipman, Ricky T. Q. Chen, Heli Ben-Hamu, Maximilian Nickel, and Matt Le. Flow Matching for Generative
> > > > Modeling, February 2023.
> > > >
> > > > [2] Antonio Terpin, Nicolas Lanzetti, Martín Gadea, and Florian Dorfler. Learning diffusion at lightspeed.

---

> > > > > ### Comment · Reviewer_HWBG · 2025-09-14
> > > > >
> > > > > Thank you for your prompt response.
> > > > >
> > > > > I now understand the motivation behind Section 3 after reviewing your additional comments. However, the current manuscript does not clearly reflect this logical flow.
> > > > >
> > > > > Regarding the connection between Sections 3 and 4, I still find them largely independent. From my perspective, the method proposed in Sections 4 and 5 does not appear to be influenced by the content of Section 3 (e.g., in terms of method selection or the validity of relying on reward fine-tuning). If this is indeed the case, I suggest splitting the manuscript into two separate papers: one focusing on the theoretical aspects and the other on the methodology. Currently, many theoretical details are deferred to the appendix, which further separates the contributions. As it stands, combining these relatively unrelated contributions in a single paper makes the manuscript unnecessarily complex.

---

> > > > > > ### Author Response · Authors · 2025-09-15
> > > > > > **Response to Comment**
> > > > > >
> > > > > > Hello, and thank you again for the follow-up. Glad to hear there's an understanding! As promised in our prior response, we have included the exposition into the paper (mainly in the first para. of Sec. 3.2). We also take the new points you bring up, and we respond below.
> > > > > >
> > > > > > Our aim in this paper is to provide an overview of how flow-based frameworks can be used for neural network weight generation, and for this we believe it is necessary to include both the theoretical and empirical aspects. Crucially, much of what we present as theory is **not** intended to claim novel mathematical results but rather to apply and re-frame existing techniques in the neural-weight setting.
> > > > > >
> > > > > > Nevertheless, we see its value in providing a common lens for understanding the diverse methods we present in Section 4 (CFM, MMFM, JKOnet), as discussed in the previous response. For example, the connection between JKOnet and MMFM becomes clearer once viewed through the action gap formalism, and Theorem 1 underpins our choice to model trajectories via the continuity equation in the first place.
> > > > > >
> > > > > > Because our theoretical contributions primarily synthesize or modestly extend existing techniques, they are **not intended to stand as a separate line of results**. Rather, they serve to motivate and connect the methodological choices and empirical studies that follow. In this sense, the theoretical and empirical parts are tightly coupled, and separating them into two independent papers would fragment the narrative and obscure the intended purpose.
> > > > > >
> > > > > > Regarding the material in the appendix, we mainly push the details of Sec 3.2.3. This constitutes a slight generalization we do not empirically explore; we included them because they contextualize methods like Metric Flow Matching that attempt richer regression targets. As is conventional, proofs and technical derivations are placed in the appendix.
> > > > > >
> > > > > > In terms of revisions, we have taken concrete steps to clarify this flow:
> > > > > > - The title of Section 3.1 has been revised to *The continuity equation on neural network parameters*, allowing room for later specification of flow matching.
> > > > > > - At the start of Section 3.2, we now explain that Theorem 1 underpins the modeling framework and provides a **common lens** for the interpretation of methods.
> > > > > > - We explicitly state that JKOnet is included because of its connection to MMFM via the action gap (alongside a few other reasons).
> > > > > > - The last sentence in Section 4.1 and reward fine-tuning now ties back to Section 3.2.
> > > > > >
> > > > > > We hope these changes make the logical flow between Sections 3 and 4 clearer. Please let us know if there remain specific points of disconnect, and we would be glad to refine the exposition further.

---

### Review · Reviewer_A2CX · 2025-08-24

**Summary Of Contributions:**

The authors propose a novel framework for learning a flow-based generative model of neural network weights by treating the gradient descent optimization path as a trajectory to be learned. This work unifies several trajectory inference techniques, such as action matching and JKOnet, under a common framework to model this "gradient flow" on the neural manifold. By doing so, the optimization process itself becomes an inductive bias for a meta-learning model. The paper explores various architectural and algorithmic components, and presents empirical demonstrations showing that the proposed method can generate effective weight initializations, facilitate fine-tuning, and detect harmful covariate shifts.

**Audience:**

Yes

**Audience Explanation:**

This paper will likely be of high interest to researchers in generative modeling, meta-learning, and optimization. The prospect of learning priors over entire optimization trajectories is a significant and exciting research direction. There are also practical applications to covariate shift detection.

**Broader Impact Concerns:**

None.

**Claims And Evidence:**

Yes

**Claims Explanation:**

The methods rest on theoretical foundations and the authors provide experimental validation.

**Requested Changes:**

My primary concerns relate to the disorganization of the methodology section and the need for a clearer articulation of the conceptual advantages of the proposed approach.

**Major Revisions**

- The paper's primary weakness is its organization, particularly in Section 3. This section currently reads more like a literature review of different, seemingly disconnected techniques (e.g., action matching, JKOnet) rather than a coherent description of the proposed method. This makes it very difficult for the reader to understand the complete pipeline and how the different components fit together.
I strongly urge the authors to add a subsection at the very beginning of the methodology section that clearly lays out the entire proposed pipeline from start to finish. Following this overview, the subsequent subsections can then delve into the details of each component.
- When multiple methods are presented (e.g., action matching and JKOnet), it is unclear if both are used, if they are alternatives, or if one is presented for background. Please explicitly state their role in your final methodology. Why are both presented if only one is used?
- The paper needs to more clearly articulate why learning a generative model of the optimization trajectory is advantageous compared to simply running the optimization itself. Please add a discussion that addresses the question: "If the generative model learns to match the gradient flow trajectory, what are the benefits over executing the gradient flow directly?"
- The mathematical framework is built upon existing theory. While this is perfectly acceptable, the paper should be more transparent about this. The inclusion of standard proofs in the appendix feels unnecessary and could give a misleading impression of theoretical novelty.

**Minor Comments and Typos**
- Eq. (9): The notation $\mu_t$ is introduced without a clear definition. It appears to be the interpolation between the source and target distributions, but this should be stated explicitly, especially since $\mu$ was previously used to denote a measure.
- Eq. (6): The use of $\nabla^2$ for the Laplacian operator is non-standard in this context. It is more common to reserve $\nabla^2$ for the Hessian matrix. Please consider using the standard Laplacian symbol, $\Delta$, to avoid ambiguity.
- Eq. (14): There appears to be a missing norm inside the expectation.
- Pg. 2: $(\mathbf x_t)_{t \in [0,1]}$ should be called a “curve”, “path”, or “trajectory”, not a "sequence".

---

> ### Author Response · Authors · 2025-08-29
> **Response to Reviewer A2CX**
>
> We thank the reviewer for their detailed comments and, in particular, for highlighting areas of presentation that required clarification. Below, we address the points in turn, highlighting revisions in violet text.
>
> ## 1. Organization of Section 3 and Overall Pipeline
> We acknowledge that Section 3 previously read more like a literature survey than a coherent presentation of the proposed method. To address this:
>
> - We restructured Section 3 into two parts: the first (Sections 3.1-2) introduces the continuity equation and the theoretical lens of the action gap. The second shows how existing objectives (MMFM, JKOnet) can be cast as minimizing the gap (in the limit).
> - To separate theory from practice, the modular framework---comprising encoder, generative meta-model, and reward fine-tuning---is presented in Section 4, while Section 3 focuses on the theoretical discussion.
> - To clarify our intentions, Section 4 now clearly frames our method as modular, aligning better with our goal to present this modular view of weight generative modeling. Unlike prior work that proposed a singular system for weight generation, these modules are flexible and they may be used depending on the desired task, compute resources, or time requirements. To make this study feasible, we made simple, representative choices for each module, often inspired by prior work, but others could be made. Although, we emphasize that the meta-fine-tuning module has yet to be proposed in prior work on weight generation. We also provide some motivation for each component.
>
> ## 2. Role of Action Matching, JKOnet, and Other Methods
> The action gap is highlighted as the unifying theoretical concept (hopefully made clearer with the new formatting of Section 3.2), while in practice only MMFM, JKOnet, CFM, and diffusion are used as exchangeable instantiations for the generation module within our framework. We highlight this by placing the action matching background into its own subsection: Sec. 3.2.1.
>
> ## 3. Advantages of Learning Generative Models of Trajectories
> See the introduction, 2nd paragraph for a brief discussion of this concern, addressing why one might prefer generative modeling of weight trajectories over simply running optimization. In summary:
>
> Efficiency: Generative models can amortize the cost of optimization, producing few-step inference at test time. Evidence is given in new experiments in App. G.
>
> Recasting problems: Generative modeling allows problems traditionally framed in data space to be reinterpreted in weight space. Our covariate-shift detection example demonstrates this principle in practice, however, we envision other applications such as inverse image problems on INR weight space, or generating quantized networks (mentioned in the Conclusion section).
>
> We welcome feedback if more detailed discussion would be helpful in this paragraph.
>
> ## 4. Transparency About Theoretical Contributions
> We agree with the reviewer that the mathematical framework largely builds on existing theory. To avoid giving a misleading impression of novelty:
>
> - In Section 3.2 and its preambles, we now clearly state that the main contribution lies in recasting objectives through the action gap, not in developing entirely new theory.
> - Theorem 1 is acknowledged as stemming from existing work directly in the theorem box. We see the benefit of including the proof in relating this to the multi-marginal and JKO methods presented; to clarify this, we’ve added two remarks in the proof in App. A.1 to draw attention to this.
> - Section 3.2.3 has been trimmed and moved largely to App. A.3, with a clear statement that the goal is only a slight generalization (via proxy curves) rather than a novel theoretical advance.
>
> ## Minor comments and typos
> We have incorporated the suggestions into our revisions.
>
> We hope these revisions address the reviewer’s concerns about organization, clarity, and positioning of contributions. We believe the new structure makes the pipeline much easier to follow while presenting the theory in proper context. We welcome further suggestions to refine the exposition.

---

> > ### Comment · Reviewer_A2CX · 2025-08-31
> >
> > The revision did not address my concerns; I still do not understand what the proposed pipeline is. Considering that other reviewers also raised similar concerns, I think there should be more effort toward clarifying the method.

---

> > > ### Author Response · Authors · 2025-09-02
> > > **Response Addendum**
> > >
> > > Hello, and thank you for following up. Apologies for the earlier misunderstanding---in our revision we had focused on clarifying the role of each component, but we now see that what was missing was an end-to-end picture of the pipeline itself.
> > >
> > > To address this, we have added a schematic (new Figure 1) that illustrates the pipeline and its stages. We reference this figure with additional notes in the opening of Section 4.1. While the figure provides one concrete example, we emphasize that our framework is modular and components may be swapped out. For instance, NM-CFM could be replaced by NM-JKO or NM-MMFM, with the corresponding change in the generative loss.
> > >
> > > We hope this resolves the earlier confusion and makes the proposed method clearer. Please let us know if any parts of the pipeline remain unclear, or if there are other points in the original response that would benefit from further clarification.

---

> > > ### Author Response · Authors · 2025-09-16
> > > **Follow-up**
> > >
> > > Dear Reviewer A2CX,
> > >
> > > Thank you again for reviewing our work! As we are currently having a last minute discussion with other reviewers, we wanted to check if you have any further questions or found our responses helpful? Beyond the addendum above, relevant to your concerns, we have also renewed the opening paragraph of Section 3.2 to better connect the theory discussion of Section 3 and the methodology of Section 4.
> > >
> > > Please let us know and thank you for your time!

---

### Comment · Action_Editor_Syvs · 2025-08-09
**Review Process Extended**

Hi Everyone,

Owing to a reviewer dropping out of the process, the reviews for this work are lagging behind. I have assigned a third reviewer who should hopefully be able to review in the next two weeks. Apologies for the delays here.

---

### Author Response · Authors · 2025-08-29
**Meta Response**

Hi everyone,

We would like to thank the reviewers for their thoughtful and detailed comments, which have greatly improved the clarity and organization of our paper. The reviewing delay also gave us additional time to refine the work, and we hope these improvements make the wait worthwhile. Below, we summarize the major changes, highlighting changes in violet text.

## Writing and Presentation
### Motivation
We would like to draw attention to the 2nd paragraph of Sec. 1 which includes motivating remarks in addition to the abstract.
### Contributions
See the walkthrough of claims in the response to Reviewers HWBG and rwfy.
### Section 3

- Restructuring: Section 3 is now divided into two parts. Sections 3.1–3.2 introduce the continuity equation and the action gap as a unifying theoretical lens. Our intentions of using this as just a theoretical lens have been clarified by separating the action matching background into its own Section 3.2.1 and adding some preamble for Section 3.2 to overview the structure. The second shows how existing objectives (MMFM, JKOnet) can be cast as minimizing the gap (in the limit).
- Clarifying connections: The relationship between Action Matching and JKOnet is now explicitly stated in Section 3.2.2, including clarifications in the JKOnet paragraph and Theorem 2.
- Proxy curves: Section 3.2.3 has been trimmed and largely moved to App. A.3. In the main text, we clearly state that proxy curves are only a slight generalization of linear interpolations.
- Transparency of contributions: Since much of the framework builds on existing theory, we now explicitly state that our main contribution lies in recasting objectives through the action gap. Theorem 1 is acknowledged as stemming from prior work directly in its name. We retain the proof in App. A.1 but add two remarks clarifying its connection to multi-marginal and JKO methods.
### Section 4

- Method overview and modularity: The proposed method now has a dedicated Section 4, which clearly presents our framework as modular, with three components: encoder, generative meta-model, and reward fine-tuning. We emphasize that each module is exchangeable depending on the task, compute, or architectural considerations. The meta-fine-tuning component, in particular, is a novel contribution not seen in prior work. Section 4.1 now includes clearer motivation for each component and explains that our instantiations are representative rather than unique.
- Figures and tables: We replaced Figure 1 with a schematic of an end-to-end illustration of the pipeline. We also revised figure and table captions for clarity, and corrected formatting issues (notably the one in Table 1, flagged by Reviewer HWBG).
- Application connection: To better tie the framework to its application, we situated Meta-Detectron in the Methods (Sec. 4.2) and added a few words to the motivation.

## Further Experiments
In App. G, we added experiments that strengthen the case for using Flow Matching and support the modeling choices. These include:

- Flexible choice of prior: New experiments (App. G.2–G.3) show how priors (e.g., Gaussian vs. Kaiming) affect trajectory quality.
- Control over stochasticity: Diffusion-based approaches struggled to generate raw parameters, likely due to stochasticity and the prior. For instance, CFM degrades when $\sigma > 0.01$, motivating our formulation of a deterministic adjoint-matching method.
- Handling multiple marginals: FM naturally accommodates multiple marginals (multiple CFMs), unlike diffusion, which struggles due to Gaussian prior assumptions. This advantage is highlighted in new trajectory modeling experiments (Section 5.1, App. G.3).

## Parameter Symmetries
We recognize that our instantiations did not use symmetry-aware architectures, which is an important limitation. To address this:

- We added a footnote in the introduction and updated Section 4.1 to highlight symmetry-aware alternatives, noting this as a direction for future work in the Conclusion.
- To test whether results are significantly affected, we conducted additional experiments in App. G, incorporating two strategies: Alignment of weight data to a reference base model, and substitution of the UNet with a recent scale- and permutation-equivariant architecture (MonomialNFN).

These results suggest that while symmetry can provide benefits in underfitting regimes, our main conclusions remain robust under the full inference procedure.

We believe these revisions substantially improve the clarity, rigor, and positioning of the paper. The restructuring of Sections 3 and 4 makes the methodology far easier to follow, the added experiments strengthen our empirical case, and the discussion of symmetries provides a more balanced view of limitations and future directions. We hope these changes satisfactorily address the reviewers’ concerns and bring the paper closer to a publishable standard.

---

> ### Author Response · Authors · 2025-09-19
> **Meta Response Addendum**
>
> - Following the comments from Reviewer A2CX, we have added a diagram of an example pipeline to clarify how the different methods fit together.
> - We have also included clarifying notes to strengthen the connection between Sections 3 and 4, in particular by revising the opening paragraph of Section 3.2 to explicitly link the theoretical discussion with the methodological choices.
>
> ## On Scope
> We wish to emphasize that our paper does not aim to resolve every aspect of generative modeling on weight space. Rather, our contributions are focused on: **1)** leveraging structural priors (e.g., training trajectories and source distributions),  **2)** exploring informative priors, and **3)** incorporating reward fine-tuning for downstream tasks; **this was done by adopting the flow matching framework on weight space.** We have taken care to define this scope in the Introduction (last two paragraphs).
>
> ## On Permutation Symmetries
> We acknowledge that our earliest draft did not explicitly account for permutation symmetries. In the subsequent revisions, our challenge has been that this issue effectively amounts to a “negative open question”: one cannot conclusively prove the absence of an effect. The best we can do is present a series of experiments (App. G), each suggesting limited impact, and leave it to the reader’s discretion whether this cumulative evidence is convincing. For this reason, we are careful to describe our findings as modest, avoid overstating their significance, and clearly flag this as a limitation to be dealt with in future work.
>
> The reviewer’s concern was not directed at requesting specific additional experiments, but at the broader implications of how symmetries might affect our method. We agree this is an important question, but we emphasize that it falls beyond the immediate scope of this paper. **Further, similar published work such as D2NWG [1] does not account for this effect.** Our decision to acknowledge the issue and provide preliminary results was motivated by a desire not to overlook a concept that has received much attention in recent work, even if we could not provide a definitive resolution. We would also welcome guidance on which experiments might be most sensitive to permutation symmetry, so that future work can more directly address those points.
>
> ## On Scalability
> Regarding scalability, we emphasize that our base model definitions follow the model zoo introduced by Schürholt et al. [2], which is widely used in weight space learning studies. While we acknowledge that scaling to larger models remains an important challenge, we note our alignment with prior work and its scant connection with the paper's claims.
>
> ## Comparison with D2NWG
> Since D2NWG [1] has been mentioned repeatedly in this review, we add the following big-picture comparison to clarify how our contributions differ:
>
> - **Reward fine-tuning**: We present experiments on fine-tuning generative models in weight space with the recent adjoint matching framework
> - **Detecting covariate shift**: Using reward fine-tuning, we included a novel application to detecting covariate shift that better integrates *an understanding of the model* to determine out-of-distribution test data.
> - **Trajectory modeling**: We explore several methods for modeling training trajectories (enabled by switching to a flow matching framework), supported by theoretical discussion.
> - **Informative priors**: We investigate the role of informative priors and their effect on weight space generation (Sec 5, App. G).
>
> All of these elements align with our stated goal, as outlined in the discussion on Scope.
>
> ## Closing
>
> We would like to thank the reviewers for their many detailed and constructive comments, which have significantly improved the clarity and focus of the paper. We are also grateful to the Action Editor for permitting a more extended discussion period, which allowed us the opportunity to address concerns more thoroughly and to refine the manuscript in line with the feedback received.
>
> ### References
>
> 1. Bedionita, S., Andreis, B., Lee, H., Jeong, W., Chong, S., Hutter, F., & Hwang, S. J. (2025). Diffusion-based neural network weights generation. In 13th International Conference on Learning Representations, ICLR 2025.
>
> 2. Schürholt, K., Taskiran, D., Knyazev, B., Giró-i-Nieto, X., & Borth, D. (2022). Model zoos: A dataset of diverse populations of neural network models.

---

### Decision · Action_Editor_Syvs · 2025-09-29

**Recommendation:** Reject

**Additional Comments:**

Please consider the comments of the reviewers below and rewrite your manuscript to emphasize clarity, and more closely link the experimental results to the theoretical ones.

---

Note from EiCs: Prior to this decision being finalized, the EiCs became aware that a version of this paper was submitted to AISTATS, breaking the dual submission guidelines for both venues. The authors claim that they intended to withdraw the AISTATS submission prior to the date that reviewing was slated to begin, if a decision for the TMLR submission was not received by then. Nonetheless, since the full paper was submitted to AISTATS, this still constitutes a dual submission, and thus this should be considered a desk rejection. However, since the decision for TMLR was since finalized, we leave the text provided by the AE above for the sake of openness.

**Audience:**

Yes

**Audience Explanation:**

Flow matching and distributional weight space learning are both hot topics in machine learning.

**Claims And Evidence:**

No

**Claims Explanation:**

This paper really lacks on the clarity angle, which in turn makes it hard to be convinced by the claims. I appreciate that the authors have made some changes to the paper to address the issues of clarity, but there remains a majority opinion among the reviewers that the current manuscript still does not meet the standards of TMLR and requires major revision.

Furthermore, the study on parameter symmetry is inadequate. More work is required here to reconcile the core premise of manifold-based weight space learning with the neglect of the natural manifold on weight space induced by parameter symmetry.

Lastly, the complete disregard for scalability is concerning. Scalability is of fundamental importance in machine learning, despite the precedent set by related work.

I would like to thank the reviewers for engaging with discussion with the authors, and I would like to reiterate my appreciation for the authors' genuine attempts to improve their work. I find the work to be interesting but in need of serious polish, and it needs to give more attention to the fundamental concerns raised above.

**Resubmission Of Major Revision:**

The authors may consider submitting a major revision at a later time.